# Accelerated Distance-adaptive Methods for Hölder Smooth and Convex Optimization

**Yijin Ren**[*]      **Haifeng Xu**[*]

School of Information Management and Engineering
Shanghai University of Finance and Economics

**Qi Deng** [†]

Antai College of Economics and Management
Shanghai Jiao Tong University

`lanyjr@stu.sufe.edu.cn, xuhaifeng2004@stu.sufe.edu.cn, qdeng24@sjtu.edu.cn`

## Abstract

This paper introduces new parameter-free first-order methods for convex optimization problems in which the objective function exhibits Hölder smoothness. Inspired by the recently proposed distance-over-gradient (DOG) technique, we propose an accelerated distance-adaptive method which achieves optimal anytime convergence rates for Hölder smooth problems without requiring prior knowledge of smoothness parameters or explicit parameter tuning. Importantly, our parameter-free approach removes the necessity of specifying target accuracy in advance, addressing a significant limitation found in the universal fast gradient methods (Nesterov, 2015). We further present a parameter-free accelerated method that eliminates the need for line-search procedures and extend it to convex stochastic optimization. Preliminary experimental results highlight the effectiveness of our approach on convex nonsmooth problems and its advantages over existing parameter-free or accelerated methods.

## 1   Introduction

In this paper, we consider the following composite function

$$\min_{x \in \mathbb{R}^d} \ \psi(x) := f(x) + g(x), \tag{1}$$

where $f \colon \mathbb{R}^d \to \mathbb{R}$ is a continuous convex function, and $g \colon \mathbb{R}^d \to \mathbb{R} \cup \{+\infty\}$ is a simple proper lower semi-continuous (lsc) convex function of which the proximal operator is easy to evaluate. First-order algorithms—(sub)gradient descent and their accelerated variants—are the workhorses for solving (1), particularly in large-scale machine learning and AI applications. Their empirical success, however, hinges on judiciously chosen stepsizes; manual tuning quickly becomes the dominant practical bottleneck.

In standard analysis, the stepsize policy is often designed based on the smoothness level of the objective function. Specifically, subgradient methods for nonsmooth problems typically employ diminishing stepsizes, whereas gradient descent methods for smooth optimization often utilize a

---

[*]Equal contribution
[†]Corresponding author

39th Conference on Neural Information Processing Systems (NeurIPS 2025).

stepsize inversely proportional to the gradient Lipschitz parameter. In a seminal work, Nesterov [27] considered convex Hölder smooth problem where $f(x)$ satisfies the following condition:

$$\|\nabla f(x) - \nabla f(y)\|_* \leq L_\nu \|x - y\|^\nu, \forall x, y \in \mathbb{R}^d. \tag{2}$$

where $\nu \in [0, 1]$ continuously interpolates between the nonsmooth ($\nu = 0$) and smooth ($\nu = 1$) settings. Nesterov introduced accelerated gradient methods capable of universally achieving optimal convergence rates without prior knowledge of the smoothness parameters of the objective. Notably, the universal fast gradient method exhibits a convergence rate of

$$\psi(x^k) - \psi(x^*) \leq \mathcal{O}\left(\frac{L_\nu^{\frac{2}{1+\nu}} D_0^2}{\epsilon^{\frac{1-\nu}{1+\nu}} k^{\frac{1+3\nu}{1+\nu}}} + \epsilon\right). \tag{3}$$

where $D_0 := \|x^0 - x^*\|$ denotes the initial distance to the minimizer. An appealing property of this method is its independence from explicit knowledge of the smoothness level $\nu$ and the parameter $L_\nu$.

Despite these attractive features, recent studies have highlighted significant limitations of the universal gradient methods. One notable issue is that the universal gradient method relies on a line search subroutine to adapt to the unknown smoothness level, which makes it challenging to extend to stochastic optimization. More critically, the performance of universal methods is sensitive to the predetermined target accuracy level $\epsilon$. As $\epsilon$ must be set in advance, the method does not have an anytime convergence guarantee, which is crucial for practical implementations where iterations can be halted at an arbitrary stage. In addition, as pointed out by Orabona [30], an optimally set $\epsilon$ should depend on $D_0$, which is unfortunately unknown beforehand. Setting the value of $\epsilon$ without prior knowledge of the underlying problem structure often results in suboptimal trade-offs between the two error terms in (3). This turns out to be a critical problem in nonsmooth optimization and online learning [23, 3, 6, 5, 40]. Although [17] achieves a near-optimal rate in the smooth setting, it does not have a theoretical guarantee for nonsmooth or weakly smooth problems. A key strength of their work is providing guarantees for unbounded domains, albeit with a suboptimal rate in that setting. [1] investigates parameter-free methods and notes that most existing approaches still require certain problem-specific information, such as the Lipschitz constant and $D_0$. In contrast, our algorithms only require an estimate of a lower bound for $D_0$, and this incurs only a minor additional cost. Moreover, [15] further shows that it is impossible to develop a truly tuning-free algorithm for smooth or nonsmooth stochastic convex optimization when the domain is unbounded.

In this paper, we demonstrate that near-optimal parameter-free convergence rates can be achieved for convex Hölder smooth optimization, up to logarithmic factors. Specifically, instead of fixing the target $\epsilon$, we set variable target levels that dynamically change based on the optimization trajectory. As mentioned earlier, an optimal level shall depend on the distance to the minimizer and is difficult to compute. Motivated by the Distance over Gradients (DOG) style stepsize [3, 14], we approximate $\|x^0 - x^*\|$ by the maximum distance to the iterates and use this knowledge to choose stepsize in the accelerated gradient method. Leveraging the technique of distance adaptation, we are able to obtain a parameter-free and anytime convergence rate of

$$\mathcal{O}\left(\frac{D_0^{1+\nu} \hat{L}_\nu \log^2 e\frac{D_0}{\bar{r}}}{k^{\frac{1+3\nu}{2}}}\right), \tag{4}$$

where $\hat{L}_\nu$ is the locally Hölder smoothness constant and $\bar{r}$ is an initial guess for $4D_0$, without requiring any predefined accuracy level, knowing the smoothness level $\nu$ or the Hölder constant.

In addition, we propose a line-search-free accelerated method that achieves optimal convergence rates for both Hölder smooth and stochastic optimization problems. To eliminate the need for line search, we adopt a bounded domain assumption, as originally introduced by Rodomanov et al. [34]. Different from their approach, which explicitly requires the domain diameter $D$ to set stepsizes, our method exploits distance adaptation to approximate $D$. This can be particularly appealing as computing the diameter of a general convex set can be computationally intractable. Moreover, by estimating $D$ through the observed distance from the initial point, our method naturally adopts more adaptive stepsizes in large domains. Theorem 3 characterizes the convergence rate of this algorithm in the stochastic setting. Since we adopt the bounded domain assumption, the convergence rate remains the same as that in (4), with $D_0$ replaced by $D$. Although we cannot guarantee the potential gap between $D_0$ and $D$, experimental results support our theoretical insights and demonstrate the practical effectiveness of our approach.

## 1.1 Related work

The increasing computational cost associated with hyperparameter tuning has driven significant research interest in developing adaptive or parameter-free algorithms. The online learning community has extensively studied parameter-free optimization, particularly focusing on achieving nearly-optimal regret bounds without prior knowledge of domain boundedness or the distance to the minimizer. For example, see [23, 24, 29, 5, 40]. A recent breakthrough has been made by Carmon and Hinder [3], which moves beyond regret analysis and focuses directly on stochastic optimization. This algorithm appears to be conceptually simpler and motivates a few more practical SGD algorithms, such as Ivgi et al. [14], Defazio and Mishchenko [6], Moshtaghifar et al. [26]. A notable related study to our work is Moshtaghifar et al. [26], which applied the distance adaptation technique to Nesterov's dual averaging method. They have offered convergence guarantees across various problem classes, including nonsmooth, smooth, $(L_0, L_1)$-smooth functions, and many others. However, the convergence rate for the Hölder smooth problem remains suboptimal.

The concept of universal gradient methods adapting to Hölder smoothness was pioneered by Nesterov [27]. Subsequent works extended this approach to nonconvex optimization [11], stochastic settings [10] and stepsize adjustment [28]. It has been demonstrated that normalized gradient stepsizes [35] can automatically adapt to Hölder smoothness without requiring line search, as shown in Grimmer [12], Orabona [31]. Recently, Rodomanov et al. [34] proposed a line-search-free universally optimal method that is robust to stochastic noise in gradient estimations. However, this method requires the domain to be bounded and the diameter to be known. In parallel to universal methods, bundle-type methods [20, 2] have emerged as an effective approach for nonsmooth optimization, enabling self-adaptation to Hölder smoothness [18]. However, these methods often involve solving a complex cut-constrained subproblem and lack straightforward extensions to stochastic settings. The Polyak stepsize method [32] also exhibits self-adaptation to smoothness [13] and Lipschitz parameters [25]. It can be seen as a special case of the bundle-level method [8]. While imposing Nesterov's acceleration technique [7] in the Polyak stepsize can universally achieve the optimal rates for Hölder smooth problems, it typically requires knowledge of the optimal value.

Finally, it is important to emphasize that the parameter-free algorithms discussed in this paper differ from adaptive gradient methods [9], which have been substantially studied in the literature [16, 33, 36]. While adaptive gradient methods primarily focus on adjusting to Lipschitz constants or constructing a preconditioner to approximate the Hessian inverse [41, 38, 21], parameter-free algorithms in our context do not rely on such adaptation mechanisms.

## 2 Preliminaries

Let $\mathbb{R}^d$ denote the $d$-dimensional Euclidean space. Let $\|\cdot\| = \sqrt{\langle \cdot, \cdot \rangle}$ be the norm associated with inner product $\langle \cdot, \cdot \rangle$. Its dual norm is defined by $\|s\|_* = \max_{\|x\|=1} \langle s, x \rangle$, where $s \in \mathbb{R}^d$. For a convex function $f : \mathbb{R}^d \to \mathbb{R} \cup \{+\infty\}$, we use $\nabla f(x)$ to denote a (sub)gradient of $f(\cdot)$ at the point $x$. We use $\mathcal{B}_\delta(x) = \{y \in \mathbb{R}^d : \|y - x\| \leq \delta\}$ to denote the closed ball of radius $\delta$ centered at $x$. We define $I_\nu$ as $\left(\frac{1}{1-\nu}\right)^{\frac{1+\nu}{2}}$ when $\nu < 1$ and $1$ when $\nu = 1$ to simplify the expressions.

A convex function $f$ is said to be *locally Hölder smooth* at $z$ with radius $r$ if there exists a mapping $M_\nu : \mathbb{R}^d \times (0, +\infty) \to (0, +\infty)$ such that, for any $z \in \mathbb{R}^d$, for any $x, y \in \mathcal{B}_r(z)$ ($r > 0$), we have

$$\|\nabla f(x) - \nabla f(y)\|_* \leq M_\nu(z, r) \|x - y\|^\nu. \tag{5}$$

Nesterov [27] considered the *global Hölder smooth* functions (2) where the smoothness mapping $M_\nu(z, r)$ is reduced to a constant $L_\nu$. However, we will show that by incorporating both line search and distance adaptation, we can guarantee the boundedness of the iterates. Consequently, the complexity relies on the local Hölder smoothness, which is defined by

$$\widehat{M_\nu} := M_\nu(x^*, 3D_0) < +\infty.$$

# 3 The accelerated distance-adaptive method

**Motivation** Before describing the main algorithm, we first shed light on the intuition of Nesterov's universal gradient methods [27]. From the definition of global Hölder smoothness (2), we have the Hölder descent condition:

$$f(y) \leq f(x) + \langle \nabla f(x), y - x \rangle + \frac{L_\nu}{1 + \nu} \|x - y\|^{1+\nu}.$$

This inequality can be translated into an inexact variant of the usual Lipschitz smooth condition:

$$f(y) \leq f(x) + \langle \nabla f(x), y - x \rangle + \frac{\gamma(L_\nu, \delta)}{2} \|x - y\|^2 + \frac{\delta}{2}. \tag{6}$$

where $\gamma(L, \delta) := \left(\frac{1-\nu}{1+\nu} \frac{1}{\delta}\right)^{\frac{1-\nu}{1+\nu}} L^{\frac{2}{1+\nu}}$, and $\delta \in (0, \infty)$ is the trade-off parameter. Since the Hölder exponent may be unknown a priori, Nesterov proposed to use pre-specify $\delta = \epsilon$, where $\epsilon$ is the target accuracy, and then perform line search over $\gamma(L_\nu, \delta)$ to satisfy the inexact Lipschitz smoothness condition (6).

The primary challenge with this approach is selecting an optimal value for $\delta$. This parameter controls a fundamental trade-off: a smaller $\delta$ improves the approximation accuracy using a smooth surrogate, but it increases the effective smoothness constant $\gamma(L_\nu, \delta)$. However, choosing the optimal $\delta$ necessitates the knowledge of the distance to the minimizer, and cannot be done easily when the domain is unconstrained. Moreover, even if the domain is bounded with $D = \max_{x,y \in \text{dom } g} \|x - y\| < \infty$, this value can poorly overestimate $\|x - x^*\|$.

To address the limitation in the existing universal fast gradient method, we present the accelerated distance-adaptive method (AGDA) in Algorithm 1. Our algorithm can be viewed as a variant of the accelerated regularized dual averaging method [37, 27, 39] which involves the triplets $\{x^k, v^k, y^k\}$. The main difference from the prior work is the new stepsize and line search procedure to adapt to the distance to the minimizer.

Specifically, leveraging the distance adaptation technique [26, 14], we approximate $D_0$ by a sequence of values:

$$\bar{r}_k = \max\{\bar{r}_{k-1}, r_k\}, \quad \text{where} \quad r_k = \|x^0 - v^k\|, \; k \geq 0, \tag{7}$$

and then set the averaging sequence $a_k$ based on the distance estimation:

$$a_{k+1} = A_{k+1} - A_k, \quad \text{where} \quad A_{k+1} := \left(\sum_{i=0}^k \bar{r}_i^{\frac{1}{2}}\right)^2, \; k = 0, 1, 2, \dots \tag{8}$$

To deal with the unknown Hölder smooth parameter, we invoke a line search procedure to find the appropriate value of $\beta_{k+1}$, which satisfies

$$f(y^{k+1}) \leq f(x^{k+1}) + \langle \nabla f(x^{k+1}), y^{k+1} - x^{k+1} \rangle + \frac{\beta_{k+1}}{64\tau_k^2 A_{k+1}} \|y^{k+1} - x^{k+1}\|^2 + \frac{\tau_k \eta_k}{2}, \tag{9}$$

where $\eta_k$ measures the inexactness in Lipschitz smooth approximation and $\tau_k$ is introduced by Nesterov's momentum ($\tau_k = 1$ in the non-accelerated method). As mentioned earlier, $\eta_k$ is set to a fixed $\delta$ in the universal (fast) gradient method. Different from the earlier approach, we simply take $\eta_k = \frac{\beta_{k+1}\bar{r}_k^2 - \beta_k \bar{r}_{k-1}^2}{8a_{k+1}}$, which dynamically adjusts during the optimization process.

**Line search** We describe how to find a suitable $\beta_{k+1}$ that satisfies the descent property (9). Note that in this inequality $y^{k+1}$ is dependent on $\beta_{k+1}$, we can describe the searching process of $\beta_{k+1}$ by formulating it as finding a root of a continuous function. To formalize the idea, we define the auxiliary function $l_k(\beta)$ as follows:

$$l_k(\beta) := - f(y^{k+1}(\beta)) + f(x^{k+1}) + \langle \nabla f(x^{k+1}), y^{k+1}(\beta) - x^{k+1} \rangle$$
$$+ \frac{\beta}{64\tau_k^2 A_{k+1}} \|y^{k+1}(\beta) - x^{k+1}\|^2 + \frac{\beta \bar{r}_k^2 - \beta_k \bar{r}_{k-1}^2}{16 A_{k+1}}, \tag{10}$$

where $y^{k+1}(\beta) = \tau_k v^{k+1}(\beta) + (1 - \tau_k) y^k$ is the trial point, and $v^{k+1}(\beta) := \text{argmin}_x \sum_{i=1}^{k+1} a_i(\langle \nabla f(x^i), x \rangle + g(x)) + \frac{\beta}{2} \|x - x^0\|^2$. Our line search consists of two stages, each involving an iterative procedure. We assume the first stage and the second stage can be terminated in $i_k'$-th and $i_k^*$-th iterations, respectively. In the first stage, we find the smallest value $i \in \{1, 2, \dots, \}$ such that $l_k(2^{i-1}\beta_k)$ is nonnegative and set $i_k' = i$. Consequently, we will have two situations:

**Algorithm 1** Accelerated Gradient Method with Distance Adaption (AGDA)

---

**Input:** $x^0$ and $\bar{r}$;
1: Initialize $A_0 = 0$, $\bar{r}_{-1} = \bar{r}_0 = \bar{r}$ and $\beta_0$ be a small constant, like $10^{-3}$.
2: Set initial solution: $v^0 = y^0 = x^0$
3: **for** $k = 0, 1, ..., K - 1$ **do**
4:      Set $r_k$ and $\bar{r}_k$ according to (7);
5:      Update $a_{k+1}$ and $A_{k+1}$ by (8), and set $\tau_k = \frac{a_{k+1}}{A_{k+1}}$;
6:      Set $x^{k+1} = \tau_k v^k + (1 - \tau_k) y^k$;
7:      Apply the line search to find $\beta_{k+1}$ such that $l_k(\beta_{k+1}) \geq 0$;
8:      Compute $v^{k+1} = \operatorname{argmin}_y \left\{ \frac{\beta_{k+1}}{2} \|x^0 - y\|^2 + \sum_{i=1}^{k+1} a_i (\langle \nabla f(x^i), y - x^i \rangle + g(y)) \right\}$;
9:      Set $y^{k+1} = \tau_k v^{k+1} + (1 - \tau_k) y^k$;
10: **end for**
**Output:** $z^K = \arg\min_{y \in \{y^0, y^1, ..., y^K\}} \psi(y)$

---

1. $i'_k = 1$, i.e., $l_k(\beta_k) \geq 0$, then we set $i^*_k = 0$ and complete the line search;

2. $i'_k > 1$, then we perform a binary search to find an approximate root of $l_k(\cdot) = 0$ in the interval $[2^{i'_k - 2} \beta_k, 2^{i'_k - 1} \beta_k]$. The search stops when the interval width is no more than a tolerance level of $\epsilon^l_k = \frac{\beta_0}{2k^2}$. We set $\beta_{k+1}$ as the right endpoint of the final interval.

We now establish the correctness and computational efficiency of the line search procedure.

**Proposition 1.** *Suppose $f(\cdot)$ is locally Hölder smooth (5) in $\mathcal{B}_{3D_0}(x^*)$. In Algorithm 1, for any $k \geq 0$, at least one of the following two conditions holds:*

1. *$l_k(\beta_k) \geq 0$;*

2. *there exists $\beta^*_{k+1} > \beta_k$ such that $l_k(\beta^*_{k+1}) = 0$, and for all $\beta > \beta^*_{k+1}$, we have $l_k(\beta) > 0$.*

*Consequently, we have $\beta_{k+1} \leq \mathcal{O}(k^{\frac{3-3\nu}{2}})$. Moreover, the total number of iterations required by the line search in Algorithm 1 is $\sum_{k=0}^{K-1} (i'_k + i^*_k) = \mathcal{O}(K \log K)$.*

**Remark 1.** *Proposition 1 implies that our method requires an additional $\mathcal{O}(\log K)$ function evaluations compared to other algorithms [27]. However, it is worth emphasizing that our line search procedure only requires access to function values, whereas the line search in the universal fast gradient method involves both gradient and function value evaluations.*

Next, we provide two lemmas to obtain an important upper bound on the convergence rate and the guarantee of the boundedness of the optimization trajectory.

**Lemma 1.** *In Algorithm 1, suppose $\bar{r} \leq 4D_0$. If for $i = 0, 1, \ldots, k$, the line searches are successful and we have $l_i(\beta_{i+1}) \geq 0$, then we have the following convergence property:*

$$\psi(y^{k+1}) - \psi(x^*) \leq \frac{\beta_{k+1}(D_0^2 - D_{k+1}^2)}{2A_{k+1}} + \frac{\beta_{k+1} \bar{r}_{k+1}^2}{8A_{k+1}}. \tag{11}$$

**Lemma 2.** *In Algorithm 1, suppose that for all $i = 0, 1, \ldots, k$, the iterates $x^i, y^i$ and $v^i$ lie within the set $\mathcal{B}_{3D_0}(x^*)$. Then, the line search in the $k$-th iteration terminates in a finite number of steps, and $x^{k+1}, y^{k+1}, v^{k+1}$ remain within $\mathcal{B}_{3D_0}(x^*)$.*

One key insight of the distance adaptation is that the inequality (11) implies the boundedness of $r_k$. To avoid the first step of Algorithm 1 from searching too far and breaking the boundedness of $r_k$, we should adopt a conservative distance estimation such that $\bar{r} \leq 4D_0$. The smaller $\bar{r}$ is, the more likely it is that $\bar{r} \leq 4D_0$ holds. Therefore, by repeatedly applying these two lemmas, we derive an important upper bound on the convergence rate and establish the boundedness guarantee of the optimization trajectory throughout all iterations.

**Theorem 1.** *Suppose $f(\cdot)$ is locally Hölder smooth (5) in $\mathcal{B}_{3D_0}(x^*)$. For any $k > 0$, it holds that*

$$\psi(y^k) - \psi(x^*) \leq \frac{\beta_k(D_0^2 - D_k^2)}{2A_k} + \frac{\beta_k \bar{r}_k^2}{8A_k}, \tag{12}$$

*Furthermore, if $\bar{r} \leq 4D_0$, then it holds that $\|v^k - x^0\| \leq 4D_0$ and $\|v^k - x^*\| \leq 3D_0$, for all $k \geq 0$.*

Since both $x^i$ and $y^i$ are convex combination of $v^i$, we immediately have that all the generated points $\{x^i, y^i\}_{i \geq 0}$ are in $\mathcal{B}_{3D_0}(x^*)$.

Next, we further refine the upper bound in (12). Since $D_0^2 - D_k^2 \leq 2D_0 r_k$, $r_k \leq \bar{r}_k$ and $r_k \leq 4D_0$, the upper bound in Theorem 1 can be relaxed to

$$\psi(y^k) - \psi(x^*) \leq \frac{3\beta_k \bar{r}_k D_0}{2A_k}.$$

It remains to control the growth of $\frac{\bar{r}_k}{A_k}$. To this end, we invoke a useful logarithmic bound [14, 22] as follows.

**Lemma 3.** *Let $(d_i)_{i=0}^{\infty}$ be a positive nondecreasing sequence. Then for any $K \geq 1$,*

$$\min_{1 \leq k \leq K} \frac{d_k}{\sum_{i=0}^{k-1} d_i} \leq \frac{\left(\frac{d_K}{d_0}\right)^{\frac{1}{K}} \log \frac{e d_K}{d_0}}{K}. \tag{13}$$

To apply the above result, we simply take $d_k = \sqrt{\bar{r}_k}$. It shows there is always some $k < K$ where the error $\frac{\bar{r}_k}{A_k}$ is bounded by $\mathcal{O}\left(\frac{\log^2(\bar{r}_T/\bar{r})}{K^2}\right)$.

Next, we bring all the pieces together. As pointed out earlier, the line search ensures that $\beta_{k+1}$ is order up to $\mathcal{O}(k^{\frac{3-3\nu}{2}})$. Together with the bound over distance-adaptive term $\frac{\bar{r}_k}{A_k}$, we arrive at our final convergence rate in the following theorem.

**Theorem 2.** *Suppose all the assumptions of Theorem 1 hold. Then, Algorithm 1 exhibits a convergence rate that*

$$\psi(y^{k^*}) - \psi(x^*) \in \mathcal{O}\left(\frac{\widehat{M}_\nu D_0^{1+\nu} \left(\frac{4D_0}{\bar{r}}\right)^{1/k} \log^2 e \frac{D_0}{\bar{r}}}{K^{\frac{1+3\nu}{2}}}\right), \tag{14}$$

*where $k^* = \underset{0 \leq i \leq k}{\arg\min} \frac{\bar{r}_k^{1/2}}{\sum_{i=0}^{k-1} \bar{r}_i^{1/2}}$.*

**Remark 2.** *The term $\left(\frac{4D_0}{\bar{r}}\right)^{\frac{1}{k}}$ in the inequality (14) approaches 1 as $k$ increases and is bounded by a small constant under the mild condition $k \geq \Omega(\log(D_0/\bar{r}))$. If $\bar{r}$ is sufficiently large, this condition is much weaker than the polynomial dependency on $D_0$ typically required for nontrivial rate in gradient descent.*

**Remark 3.** *According to (14), to achieve an $\epsilon$-optimality gap, our method attains a near-optimal complexity bound of $\tilde{\mathcal{O}}\left(D_0^{\frac{2(1+\nu)}{1+3\nu}} \epsilon^{-\frac{2}{1+3\nu}}\right)$, where the $\tilde{\mathcal{O}}$ notation hides logarithmic factors arising from line search and distance adaptation, such as $\mathcal{O}(\log \frac{1}{\epsilon})$ and $\mathcal{O}(\log^2 \frac{D_0}{\bar{r}})$.*

**Remark 4.** *Theorems 1 and 2 require the initial guess $\bar{r}$ to lie within a reasonably large neighborhood, specifically $\bar{r} \leq 4D_0$. This condition is a key assumption underlying distance-adaptive methods [14]. For theoretical purposes, we provide an automatic initialization strategy for $\bar{r}$ in certain special cases (see Appendix). Empirically, we observe that the performance of the algorithm is largely insensitive to the specific choice of $\bar{r}$.*

## 4 Stochastic optimization

In this section, we focus on stochastic optimization of Hölder smooth functions, wherein problem (1), $f(x)$ exhibits the expectation form:

$$f(x) = \mathbb{E}_\xi[f(x, \xi)],$$

where $\xi$ is a random sample following from specific distribution. Due to the difficulty in exactly computing the gradient $\nabla f(x)$, it is challenging to perform line search. To bypass this issue, we present a new line-search-free and accelerated distance-adaptive method in Algorithm 2. At the cost of removing linesearch, we require an additional boundedness assumption.

**Assumption 1** (Boundedness of domain). *The set $\mathrm{dom}\, g$ is bounded, namely, $D = \sup_{x,y \in \mathrm{dom}\, g} \|x - y\| < +\infty$. We denote $\tilde{M}_\nu = M_\nu(x^*, D)$ for simplicity.*

Let us use $\nabla \tilde{f}(x, \xi)$ to represent a stochastic gradient, we further assume the stochastic gradient has a bounded variance: $\sigma^2 := \sup_{x \in \mathbb{R}^d} \mathbb{E}_\xi[\|\nabla \tilde{f}(x, \xi) - \nabla f(x)\|_*^2] < +\infty$. For the sake of notation, we denote $\tilde{\nabla} f(x^k) = \nabla \tilde{f}(x^k, \xi_k^x)$ and $\tilde{\nabla} f(y^k) = \nabla \tilde{f}(y^k, \xi_k^y)$ to present the stochastic gradient in the $k$-th iteration, where $\xi_k^x$ and $\xi_k^y$ are two i.i.d. samples.

---

**Algorithm 2** AGDA Line Search Free Modification (AGDA LSFM)

---

**Input:** $x^0, \bar{r}$;
 1: Initialize $A_0 = 0$, $\beta_0 = 0$, $\bar{r}_0 = \bar{r}$;
 2: Set initial solution: $v^0 = \hat{x}^0 = y^0 = x^0$;
 3: **for** $k = 0, 1, ..., K - 1$ **do**
 4:     Solve $v^k = \arg\min_x \sum_{i=0}^k a_i[f(x^i) + \langle \tilde{\nabla} f(x^i), x - x^i \rangle + g(x)] + \frac{\beta_k}{2}\|x^0 - x\|^2$;
 5:     Set $d_k = \|x^0 - \hat{x}^k\|$;
 6:     Update $\bar{r}_k$ and $A_{k+1}$ by (15) and (8)
 7:     Set $a_{k+1} = A_{k+1} - A_k$, $\tau_k = \frac{a_{k+1}}{A_{k+1}}$;
 8:     Set $x^{k+1} = \tau_k v^k + (1 - \tau_k)y^k$;
 9:     Compute $\hat{x}^{k+1} = \arg\min_y\{a_{k+1}[\langle \tilde{\nabla} f(x^{k+1}), y - x^{k+1} \rangle + g(y)] + \frac{\beta_k}{2}\|v^k - y\|^2\}$;
10:     Set $y^{k+1} = \tau_k \hat{x}^{k+1} + (1 - \tau_k)y^k$;
11:     Set $\eta_k = \frac{\beta_{k+1}\bar{r}_k^2 - \beta_k \bar{r}_k^2}{8a_{k+1}}$;
12:     Solve (16) to obtain the solution $\beta_{k+1}$;
13: **end for**

---

Algorithm 2 is equipped with the following rules:

$$\bar{r}_k = \max\{\bar{r}_{k-1}, r_k, d_k\}, \ k > 0, \tag{15}$$

where $d_k = \|\hat{x}^k - x^0\|$.

In stochastic settings, traditional line search methods cannot be used as they introduce bias. Therefore, it is necessary to develop an approach that does not rely on line search. Rather than performing line search to find the descent direction, Rodomanov et al. [34] proposes a nonlinear balance equation. The core idea is to bound the error term $f(y^{k+1}) - f(x^{k+1}) - \langle \nabla f(x^{k+1}), y^{k+1} - x^{k+1} \rangle - \frac{H_k}{2}\|y^{k+1} - x^{k+1}\|^2$ by constructing a balance equation incorporating $D$. We demonstrate that the term used to bound the error is effectively equivalent to line search, allowing us to use $\bar{r}_k$ to approximate $D$, which implies that $D$ is not essential. Subsequently, we will explain how to formulate the balance equation.

As we mentioned in Section 3, line search strategy aims to find $\beta_{k+1}$ such that $l_k(\beta_{k+1}) = 0$. The difficulty is that $y^{k+1}$ depends on $\beta_{k+1}$ and thus solving $l_k(\beta_{k+1}) = 0$ cannot be achieved by a closed-form solution. The motivation for applying the balance equation is to decouple the updating rule of $y^{k+1}$ from the $\beta_{k+1}$. Once $y^{k+1}$ has been updated, $l_k(\beta_{k+1}) = 0$ will degenerate to the following balance equation

$$\frac{\beta_{k+1} - \beta_k}{2A_{k+1}}\bar{r}_k^2 = [\langle \tilde{\nabla} f(y^{k+1}) - \tilde{\nabla} f(x^{k+1}), y^{k+1} - x^{k+1} \rangle - \frac{\beta_{k+1}}{64\tau_k^2 A_{k+1}}\|y^{k+1} - x^{k+1}\|^2]_+, \tag{16}$$

where $[\cdot]_+ = \max(0, \cdot)$. We use $\langle \tilde{\nabla} f(y^{k+1}), y^{k+1} - x^{k+1} \rangle$ to replace the $-f(y^{k+1}) + f(x^{k+1})$ since we cannot obtain the function value.

Since we decouple $y^{k+1}$ from $\beta_{k+1}$, equation (16) has a simple form that is easy to solve. Moreover, it has a unique closed-form solution given by

$$\beta_{k+1} = \beta_k + \frac{[64\tau_k^2 A_{k+1}\langle \tilde{\nabla} f(y^{k+1}) - \tilde{\nabla} f(x^{k+1}), y^{k+1} - x^{k+1} \rangle - \beta_k\|y^{k+1} - x^{k+1}\|^2]_+}{32\tau_k^2\bar{r}_k^2 + \|y^{k+1} - x^{k+1}\|^2}. \tag{17}$$

We leave the details about conducting the closed-form solution in the appendix.

We next conduct the convergence analysis of Algorithm 2. In order to use the unbiasedness of the inexact oracle, we adopt the balance equation to update the $\beta_{k+1}$. Moreover, we use $\bar{r}_k$ is a natural underestimation of $D$ and Lemma 3 ensures that the cost of underestimation can be reduced to $\mathcal{O}(\log \frac{D}{\bar{r}})$. We leave the proof in the appendix.

**Theorem 3.** *Suppose Assumption 1 holds. Algorithm 2 exhibits a convergence rate that*

$$\mathbb{E}[\psi(y^{k^*}) - \psi(x^*)] \in \mathcal{O}\left( \frac{\tilde{M}_\nu D^{1+\nu}}{K^{\frac{1+3\nu}{2}}} + \frac{\sigma D}{\sqrt{K}} \right). \tag{18}$$

*where $k^* = \arg\min_{1 \le k \le K} \left\{ \frac{\bar{r}_k}{A_k} \right\}$.*

## 5 Experiments

We evaluate the performance of our proposed method on a diverse set of convex optimization problems. The goal is to assess its efficiency and robustness across different application scenarios. Additional implementation details and extended results (more different problems and large scale experiments) are provided in the appendix.

### 5.1 Deterministic setting

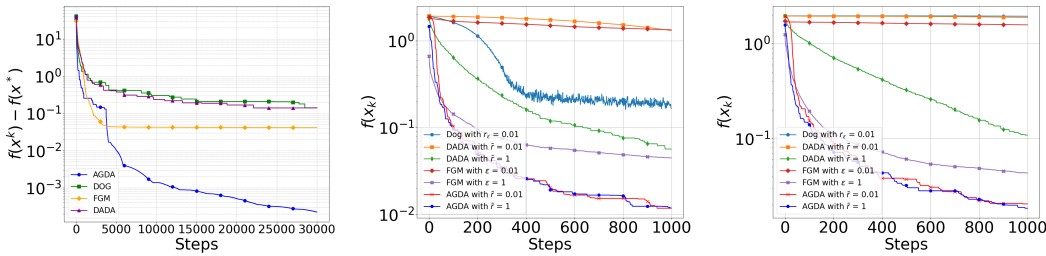

Figure 1: Performance of the compared algorithms. Left: softmax problem. Middle: Matrix game problem of size $(n, m) = (896, 128)$. Right: Matrix game of size $(n, m) = (448, 64)$.

**Softmax**  The first problem is optimizing the softmax function:

$$\min_{x \in \mathbb{R}^d} \ \mu \log \left[ \sum_{i=1}^n \exp\left( \frac{\langle a_i, x \rangle - b_i}{\mu} \right) \right], \tag{19}$$

where $a_i \in \mathbb{R}^d$, $b_i \in \mathbb{R}$ and $\mu$ is a given positive factor. This function can be viewed as a smooth approximation of the maximization function.

To facilitate a clear comparison, we design a simple baseline problem for evaluation. Specifically, we first generate i.i.d. vectors $\{\hat{a}_i\}$ and $b_i$ whose components are uniformly distributed in the interval $[-1, 1]$. Using these vectors, we define an initial softmax objective $\hat{f}(x)$ as in (19). We then shift the data by redefining $a_i = \hat{a}_i - \nabla \hat{f}(0_d)$, where $0_d$ is the $d$-dimensional zero vector, and set $f(x)$ according to (19), using the new $a_i$ and the original $b_i$. With this construction, $x = 0_d$ becomes the global minimizer of $f(x)$.

We employ various methods for comparison, specifically considering DOG [14], DADA [26], and the universal fast gradient method (FGM) [27] as benchmarks. For DOG, we set $r_\epsilon = 0.01$. Both DADA and AGDA are configured with $\bar{r} = 0.01$, while for FGM, we set $\epsilon = 0.01$. We set $n = 1000$, $d = 2000$ and $\mu = 0.005$ as the parameters of the problem. The results of our method are illustrated in the left part of Figure 1. As expected from complexity analysis, FGM, being an accelerated method, outperforms the non-accelerated baselines DOG and DADA. Notably, our proposed algorithm achieves the fastest convergence among all tested methods, which empirically confirms the advantage of our adaptive stepsize selection.

**Matrix game**  The second problem we experimented with is the matrix game problem [27]. We denote $\Delta_d$ as the standard simplex with dimension $d > 0$. Specifically, consider a payoff matrix $A \in \mathbb{R}^{n \times m}$, where two agents engage in a game by adopting mixed strategies $x \in \Delta_n$ and $y \in \Delta_m$ respectively to play a game without knowledge of each other's strategy. The gain of the first agent is

given by $\langle x, Ay \rangle$, which corresponds to the loss of the second agent. The Nash equilibrium of this game can be found by solving the saddle-point (min-max) problem:

$$\min_{x \in \Delta_n} \max_{y \in \Delta_m} \langle x, Ay \rangle. \tag{20}$$

This problem can be posed as a minimization problem: $\min_{x \in \Delta_n, y \in \Delta_m} \{\psi_{pd}(x, y) = \psi_p(x) - \psi_d(y)\} = 0$, where $\psi_p(x) = \max_{1 \le j \le m} \langle x, Ae_j \rangle$ and $\psi_d(y) = \min_{1 \le i \le n} \langle e_i, Ay \rangle$.

We generate the payoff matrix $A$ such that each entry is independently and uniformly distributed within the interval $[-1, 1]$. This problem is nonsmooth with Hölder smoothness parameter $\nu = 0$, making it a suitable test case for evaluating the robustness of optimization algorithms under minimal smoothness assumptions. We evaluate all methods on two problem sizes: $(n, m) = (896, 128)$ and $(n, m) = (448, 64)$. The performance of our method, along with the baselines, is shown in the right panels of Figure 1. The results demonstrate that our algorithm remains highly effective even in challenging nonsmooth settings, outperforming the alternatives in both cases.

## 5.2 Stochastic setting

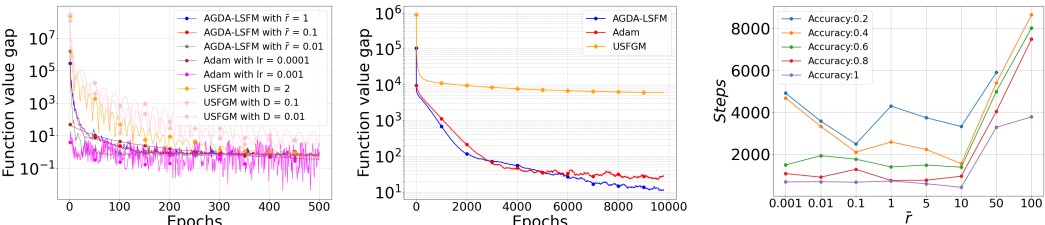

Figure 2: Performance of the compared algorithms. Left: robustness test on `diabetes` dataset. Right: Long-run test on `Boston housing` dataset. Right: robustness test on the softmax problem

**Least-squares** For the stochastic setting, we first consider the following problem:

$$\min_{x \in \mathbb{R}^d} f(x) = \frac{1}{2} \|Ax - b\|_2^2 \quad \text{s.t. } \|x\|_2 \le r. \tag{21}$$

We set the constraint radius $r = 10$ and conduct experiments using real-world datasets from LIBSVM[3]. For the first test, we use the diabetes dataset to examine robustness. For both USFGM and our AGDA-LSFM algorithm, we vary the initialization hyperparameter $D$ for USFGM and $\bar{r}$ for AGDA-LSFM—to evaluate sensitivity to stepsize-related inputs. As shown in the left panel of Figure 2, USFGM [34] and Adam exhibit unstable performance when hyperparameters are poorly tuned, while our algorithm maintains strong and consistent convergence across a wide range of settings.

To further validate algorithm efficiency in a practical regime, we repeat the experiment on the Boston housing dataset, tuning all methods with their best-performing hyperparameters. The middle panel of Figure 2 shows that our method achieves competitive long-term performance while preserving its robustness advantage. These results illustrate that our approach is not only stable but also effective in real-world stochastic optimization tasks.

## 5.3 Robustness

We conduct additional experiments to assess the robustness of our method with respect to the choice of the parameter $\bar{r}$, which reflects an estimate of the initial distance to the optimal solution, $D_0$. Our goal is to show that the performance of our algorithm remains stable across a wide range of $\bar{r}$ values, thereby reducing the sensitivity to inaccurate user-specified estimates.

To this end, we revisit the softmax minimization problem and vary $\bar{r}$ logarithmically from $10^{-4}$ to $10^4$. For each setting, we fix the target function value tolerance at $\epsilon \in \{0.2, 0.4, 0.6, 0.8, 1\}$ and record the number of iterations required to reach the specified accuracy. The results, shown in the

---
[3] `https://www.csie.ntu.edu.tw/cjlin/libsvm/`

third panel of Figure 2, reveal that our method is highly robust: the number of iterations remains nearly constant across several orders of magnitude of $\bar{r}$. This suggests that our approach can tolerate significant misspecification of $D_0$ without compromising convergence efficiency. In practice, users may either provide a rough estimate of the initial distance or simply default to a moderate value such as $\bar{r} = 10^{-3}$, which performs consistently well across our tests.

### 5.4 Non-convex neural network

To evaluate performance in non-convex optimization, we trained a ResNet18 model on the CIFAR-10 dataset. We compared the proposed AGDA algorithm against two established optimizers: AdamW and DoG. For AdamW, we set the learning rate to $10^{-3}$. The DoG algorithm was configured with $r_\epsilon = 10^{-3}$, which is consistent with the primary hyperparameter used in the AGDA implementation. The comparative results are presented below.

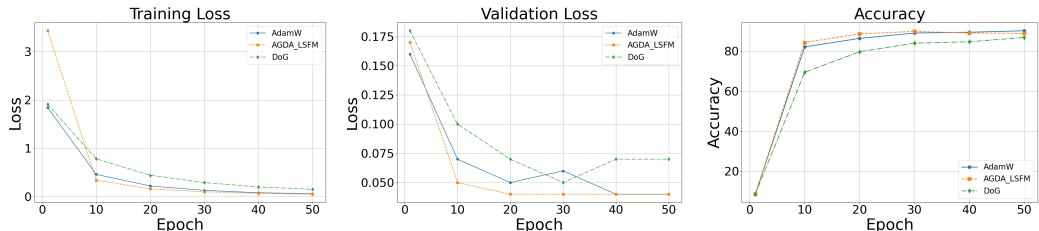

Figure 3: Performance of the compared algorithms in network training. Left: Training Loss. Right: Validation Loss. Right: Accuracy.

The results clearly show that AGDA-LSFM maintains performance on par with the AdamW baseline and achieves superior results compared to DoG. Crucially, these findings establish the competitiveness of our method against SOTA parameter-free approaches, suggesting its practical robustness extends beyond the theoretical assumptions (e.g., boundedness and convexity) that underpin its derivation, even within deep learning's highly non-convex landscape.

## 6 Conclusion

This paper introduces a novel parameter-free first-order method for solving composite convex optimization problems without requiring prior knowledge of the initial distance to the optimum ($D_0$) or the Hölder smoothness parameters. Our method achieves a near-optimal complexity bound for locally Hölder smooth functions in an anytime fashion, making it broadly applicable and practical. In the stochastic setting, we further develop a line-search-free accelerated method that eliminates the need for estimating the problem-dependent diameter D during stepsize selection. This enhances both theoretical generality and practical usability. Preliminary experiments demonstrate that our algorithms are competitive and often outperform existing universal methods for Hölder smooth optimization, particularly in terms of robustness and adaptivity. An important direction for future research is to improve the dependence on the diameter $D_0$ in the convergence complexity, and to further relax the boundedness assumptions typically required in the stochastic setting.

## 7 Acknowledgements

This research was supported in part by the National Natural Science Foundation of China [Grants 12571325, 72394364/72394360] and the Natural Science Foundation of Shanghai [Grant 24ZR1421300].

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

# Appendix

**Structure of the Appendix**    In Section A, we discuss the limitations of our algorithms. Section B presents the proofs of the lemmas for completeness. In Sections C and D, we provide detailed proofs of the main results discussed in Sections 3 and 4. In Section E, we introduce two methods for automatically setting the hyperparameters. Finally, Section F offers additional experiments to demonstrate the advantage of the proposed algorithms.

## A    Limitations

Algorithm 1 significantly reduces the multiplicative overhead of choosing a sufficiently small parameter $\bar{r}$ from a polynomial to a logarithmic factor, and lowers the average number of gradient evaluations to one per iteration—compared to four per iteration in the Universal Fast Gradient Method (FGM) [27]. However, this improvement comes at the cost of increased computational burden during the line search procedure. Specifically, to accurately adapt to the local Hölder smoothness, our method requires a more precise selection of the parameter $\beta_k$, leading to a total of $\mathcal{O}(k \log k)$ line search operations after $k$ iterations, whereas, in contrast, FGM only requires $\mathcal{O}(k)$.

## B    Auxiliary lemmas

### B.1    Proof of Lemma 3

This result was first established in Ivgi et al. [14, Lemma 3] and Liu and Zhou [22, Lemma 30]. We give a proof for completeness.

*Proof.* Let $R_k = \frac{r_k}{\sum_{i=0}^{k-1} r_i}$ and $R_0^{-1} = 0$, then for any $k \geq 0$

$$r_{k+1} R_{k+1}^{-1} = r_k R_k^{-1} + r_k \frac{r_k}{r_{k+1}} = R_{k+1}^{-1} - \frac{r_k}{r_{k+1}} R_k^{-1}.$$

Then

$$\sum_{i=0}^{k-1} \frac{r_i}{r_{i+1}} = \sum_{i=0}^{k-1} R_{i+1}^{-1} - \frac{r_i}{r_{i+1}} R_i^{-1} = \sum_{i=0}^{k-1} R_{i+1}^{-1} - R_i^{-1} + (1 - \frac{r_i}{r_{i+1}}) R_i^{-1}$$

$$= R_k^{-1} + \sum_{i=0}^{k-1} (1 - \frac{r_i}{r_{i+1}}) R_i^{-1} \leq R_{k^*}^{-1} (1 + k - \sum_{i=0}^{k-1} \frac{r_i}{r_{i+1}}),$$

where $k^* = \arg\min_{0 \leq i \leq k} R_i$. It then follows that

$$R_{k^*} \leq \frac{1 + k - \sum_{i=0}^{k-1} \frac{r_i}{r_{i+1}}}{\sum_{i=0}^{k-1} \frac{r_i}{r_{i+1}}} \leq \frac{1 + k - k(\prod_{i=0}^{k-1} \frac{r_i}{r_{i+1}})^{\frac{1}{k}}}{k(\prod_{i=0}^{k-1} \frac{r_i}{r_{i+1}})^{\frac{1}{k}}}$$

$$= \frac{1 + k - k(\frac{r_0}{r_k})^{\frac{1}{k}}}{k(\frac{r_0}{r_k})^{\frac{1}{k}}} \leq \frac{1 - k \log(\frac{r_0}{r_k})^{\frac{1}{k}}}{k(\frac{r_0}{r_k})^{\frac{1}{k}}}$$

$$= \frac{1 - \log(\frac{r_0}{r_k})}{k(\frac{r_0}{r_k})^{\frac{1}{k}}} = (\frac{r_k}{r_0})^{\frac{1}{k}} \frac{\log(e\frac{r_k}{r_0})}{k}.$$

$\square$

### B.2    Proof of auxiliary Lemmas

The following three-point Lemma is a well-known result. See also Chen and Teboulle [4, Lemma 3.2] and Lan et al. [19, Lemma 6]. We give a proof for the sake of completeness.

**Lemma 4.** *For any proper lsc convex function $\phi : \mathbb{R}^d \to \mathbb{R} \cup \{+\infty\}$, any $z \in \operatorname{dom} \phi$ and $\beta > 0$. Let $z_+ = \arg\min_{x \in \operatorname{dom} \phi}\{\phi(x) + \frac{\beta}{2}\|z - x\|^2\}$. Then, we have*

$$\phi(x) + \frac{\beta}{2}\|z - x\|^2 \geq \phi(z_+) + \frac{\beta}{2}\|z - z_+\|^2 + \frac{\beta}{2}\|z_+ - x\|^2, \forall x \in \mathbb{R}^d. \tag{22}$$

*Proof.*

$$\frac{1}{2}\|z_+ - x\|^2 + \frac{1}{2}\|z - z_+\|^2 - \frac{1}{2}\|z - x\|^2 = \frac{1}{2}\|x\|^2 - \langle z_+, x\rangle + \frac{1}{2}\|z_+\|^2$$
$$+ \frac{1}{2}\|z\|^2 - \langle z, z_+\rangle + \frac{1}{2}\|z_+\|^2$$
$$- \frac{1}{2}\|z\|^2 + \langle z, x\rangle - \frac{1}{2}\|x\|^2$$
$$= \langle z - z_+, x - z_+\rangle.$$

In view of the first-order optimal condition at $z_+$, we have

$$\langle \nabla\phi(z_+) + \beta(z_+ - z), x - z_+\rangle \geq 0.$$

Combining the two inequalities above, we have

$$\frac{\beta}{2}\|z_+ - x\|^2 + \frac{\beta}{2}\|z - z_+\|^2 - \frac{\beta}{2}\|z - x\|^2 = \beta\langle z - z_+, x - z_+\rangle$$
$$\leq \langle \nabla\phi(z_+), x - z_+\rangle$$
$$\leq \phi(x) - \phi(z_+),$$

where the last inequality uses the convexity of $\phi(\cdot)$. $\qquad\square$

**Lemma 5.** *For any $u \geq 0$, $k \geq 0$, there exits a positive constant $c_u$ such that:*

$$(k+1)^u - k^u \geq c_u(k+1)^{u-1}, \tag{23}$$

*where $c_u$ only depends on $u$.*

*Proof.* When $k = 0$, we have $(1)^u - 0^u = 1^u = 1^{u-1} = 1$. Now consider $k > 0$. We distinguish between two cases.

**Case 1:** If $u \geq 1$, we have

$$(k+1)^u - k^u = u\int_k^{k+1} x^{u-1}dx \geq uk^{u-1}$$

and hence

$$\left(\frac{k}{k+1}\right)^{u-1} \geq \left(\frac{1}{2}\right)^{u-1}.$$

Therefore,

$$(k+1)^u - k^u \geq u(\frac{1}{2})^{u-1}(k+1)^{u-1}.$$

**Case 2:** $0 \leq u < 1$,

$$(k+1)^u - k^u = u\int_k^{k+1} x^{u-1}dx \geq u(k+1)^{u-1}.$$

Therefore, we can set $c_u = u(\frac{1}{2})^{u-1}$. $\qquad\square$

## C   Missing details in Section 3

In this section, we provide a detailed convergence analysis of Algorithm 1. For the sake of simplicity, we define the following notations.

$$\phi_{k+1}(x) = \sum_{i=1}^{k+1} a_i[f(x^i) + \langle \nabla f(x^i), x - x^i\rangle + g(x)] + \frac{\beta_{k+1}}{2}\|x^0 - x\|^2,$$

$$\eta_k = \frac{\beta_{k+1}\bar{r}_k^2 - \beta_k\bar{r}_{k-1}^2}{8a_{k+1}},$$

Using this definition, it follows that $\phi_0(x) = \frac{\beta_0}{2}\|x^0 - x\|^2$.

### C.1 Proof of important lemmas

To begin our analysis, we first prove the key bound (6) used in universal gradient methods. The bound (6) ensures that these methods can be accelerated by line search without prior knowledge about $\nu$ and $L_\nu$.

The following result is from [[27], Lemma 2]. We give a proof for completeness.

**Lemma 6.** *Let* $\gamma(\widehat{M}_\nu, \delta) = \gamma_\nu(\widehat{M}_\nu, \delta) = (\frac{1-\nu}{1+\nu}\frac{1}{\delta})^{\frac{1-\nu}{1+\nu}}\widehat{M}_\nu^{\frac{2}{1+\nu}}$, *where* $\nu \in [0,1]$ *and* $\delta > 0$. *Here, we set* $(\frac{1-1}{1+1}\frac{1}{\delta})^{\frac{1-1}{1+1}} = 1$. *Suppose that for any* $x, y \in \mathcal{B}_{3D_0}(x^*)$, *we have*

$$\|\nabla f(x) - \nabla f(y)\|_* \leq \widehat{M}_\nu \|x - y\|^\nu. \tag{24}$$

*Then, for any* $x, y \in \mathcal{B}_{3D_0}(x^*)$ *we have*

$$f(y) \leq f(x) + \langle \nabla f(x), y - x \rangle + \frac{\gamma(\widehat{M}_\nu, \delta)}{2}\|x - y\|^2 + \frac{\delta}{2}. \tag{25}$$

*Proof.* Note that the condition (24) immediately implies

$$f(y) \leq f(x) + \langle \nabla f(x), y - x \rangle + \frac{\widehat{M}_\nu}{1+\nu}\|x - y\|^{1+\nu}$$

from basic convex analysis.

For any $a, b \in \mathbb{R}_+$ and $p, q \geq 1, \frac{1}{p} + \frac{1}{q} = 1$, applying Young's inequality we obtain

$$\frac{a^p}{p} + \frac{b^q}{q} \geq ab. \tag{26}$$

We choose $p = \frac{2}{1+\nu}, q = \frac{2}{1-\nu}, a = t^{1+\nu}$ and $b = (\frac{1+\nu}{1-\nu}\frac{\delta}{\widehat{M}_\nu})^{\frac{1-\nu}{1+\nu}}$ and have

$$\frac{(1+\nu)t^2}{2} + \frac{(1-\nu)(\frac{1+\nu}{1-\nu}\frac{\delta}{\widehat{M}_\nu})^{\frac{2}{1+\nu}}}{2} \geq t^{1+\nu}(\frac{1+\nu}{1-\nu}\frac{\delta}{\widehat{M}_\nu})^{\frac{1-\nu}{1+\nu}}$$

$$\frac{(1+\nu)t^2}{2}(\frac{1-\nu}{1+\nu}\frac{\widehat{M}_\nu}{\delta})^{\frac{1-\nu}{1+\nu}} + \frac{(1+\nu)\delta}{2\widehat{M}_\nu} \geq t^{1+\nu}$$

$$\frac{t^2}{2}(\frac{1-\nu}{1+\nu}\frac{1}{\delta})^{\frac{1-\nu}{1+\nu}}\widehat{M}_\nu^{\frac{2}{1+\nu}} + \frac{\delta}{2} \geq \frac{t^{1+\nu}}{1+\nu}\widehat{M}_\nu$$

$$\frac{t^2}{2}\gamma(\widehat{M}_\nu, \delta) + \frac{\delta}{2} \geq \frac{t^{1+\nu}}{1+\nu}\widehat{M}_\nu.$$

We set $t = \|x - y\|$ and obtain (25) directly. $\square$

To demonstrate the primary convergence results in Section 3, we first establish some useful lemmas regarding the well-definedness of line search and the boundedness of the iterates.

**Lemma 7.** *Let* $g : \mathbb{R}^d \to \mathbb{R} \cup \{+\infty\}$ *be a proper lsc convex function. For a given vector* $c \in \mathbb{R}^d$ *and point* $x^0 \in \mathbb{R}^d$, *define the function* $z(h)$ *for any* $h > 0$ *as:* $z(h) := \arg\min_{x \in \mathbb{R}^d}\{\langle c, x \rangle + g(x) + h\|x^0 - x\|^2\}$. *Then, the function* $\|x^0 - z(h)\|$ *is monotonically decreasing in* $h$ *and converges to* $0$ *as* $h \to +\infty$.

*Proof.* First, we prove $\|x^0 - z(h)\|$ is monotonically decreasing in $h$. For any $h_1, h_2$ such that $h_2 > h_1 > 0$, in view of the optimality of $z(h_1)$ and $z(h_2)$, we have

$$\langle c, z(h_1) \rangle + g(z(h_1)) + h_1\|x^0 - z(h_1)\|^2 \leq \langle c, z(h_2) \rangle + g(z(h_2)) + h_1\|x^0 - z(h_2)\|^2 \tag{27}$$

and

$$\langle c, z(h_1) \rangle + g(z(h_1)) + h_2\|x^0 - z(h_1)\|^2 \geq \langle c, z(h_2) \rangle + g(z(h_2)) + h_2\|x^0 - z(h_2)\|^2. \tag{28}$$

Combining (28) and (27) and noticing that $h_2 - h_1 > 0$, we have
$$(h_2 - h_1)\|x^0 - z(h_1)\|^2 \geq (h_2 - h_1)\|x^0 - z(h_2)\|^2,$$
which implies
$$\|x^0 - z(h_1)\| \geq \|x^0 - z(h_2)\|.$$

Next, we prove $\lim_{h \to +\infty} \|x^0 - z(h)\| = 0$ by contradiction. If there exists $\delta > 0$, for any $h \in \mathbb{R}_+, \|x^0 - z(h)\| \geq \delta$, then we have
$$\langle c, z(h) \rangle + g(z(h)) + h\|x^0 - z(h)\|^2 \geq \langle c, z(h) \rangle + g(z(h)) + h\delta^2.$$

Let us consider $h > h_0 > 0$. Uusing the optimality of $z(h_0)$, we obtain
$$\langle c, z(h) \rangle + g(z(h)) + h_0\|x^0 - z(h)\|^2 \geq \langle c, z(h_0) \rangle + g(z(h_0)) + h_0\|x^0 - z(h_0)\|^2,$$
Note that monotonicity proved above implies $\|x^0 - z(h)\| \leq \|x^0 - z(h_0)\|$, together with the above inequality, we have
$$\langle c, z(h) \rangle + g(z(h)) \geq \langle c, z(h_0) \rangle + g(z(h_0)).$$
Moreover, using the optimality at $z(h)$ and the lower boundedness $\|x^0 - z(h)\| \geq \delta$, we have
$$\langle c, x^0 \rangle + g(x^0) \geq \langle c, z(h) \rangle + g(z(h)) + h\|x^0 - z(h)\|^2 \geq \langle c, z(h_0) \rangle + g(z(h_0)) + h\delta^2. \quad (29)$$
Since $h$ can be arbitrarily large, this result is impossible unless $\delta = 0$. $\qquad\square$

### C.2 Convergence analysis of AGDA

**Outline**  The analysis of AGDA is slightly more involved than the standard complexity analysis for smooth problems, as we must simultaneously prove the boundedness of the iterates and establish the convergence rate. Our proof strategy is centered on an inductive argument. To proceed, we outline the structure of the analysis. Lemma 1 and Lemma 2 develop crucial results regarding one-step iteration, including the success of the line search, convergence error, and the boundedness of the iterates, assuming that all previous steps are well-defined. This serves as the foundational building block for our inductive analysis. Lemma 8 establishes growth bounds on $\beta_k$. Building on this, Proposition 1 addresses the complexity of the line search step. By employing mathematical induction, we conclude in Theorem 1 the boundedness property of all iterates and establish key convergence properties of $y^{k+1}$. Finally, utilizing the technique of distance-adaptive stepsizes, we derive the overall convergence rate of AGDA in Theorem 2.

Next, we establish an important property about the convergence of the algorithm.

**Proof of Lemma 1**

*Proof.* Since $l_k(\beta_{k+1}) \geq 0$, we have
$$l_k(\beta_{k+1}) = -f(\tau_k v^{k+1}(\beta_{k+1}) + (1 - \tau_k)y^k) + f(x^{k+1}) + \langle \nabla f(x^{k+1}), \tau_k v^{k+1}(\beta_{k+1}) $$
$$+(1 - \tau_k)y^k - x^{k+1} \rangle + \frac{\beta_{k+1}}{64\tau_k^2 A_{k+1}}\|\tau_k v^{k+1}(\beta) + (1 - \tau_k)y^k - x^{k+1}\|^2 + \frac{\beta \bar{r}_k^2 - \beta_k \bar{r}_{k-1}^2}{16 A_{k+1}} \geq 0,$$
$$\tag{30}$$
i.e.,
$$f(y^{k+1}) \leq f(x^{k+1}) + \langle \nabla f(x^{k+1}), y^{k+1} - x^{k+1} \rangle + \frac{\beta_{k+1}}{64\tau_k^2 A_{k+1}}\|y^{k+1} - x^{k+1}\|^2 + \frac{\tau_k \eta_k}{2}.$$

Because $x^{k+1} = \tau_k v^k + (1 - \tau_k)y^k$, $y^{k+1} = \tau_k v^{k+1} + (1 - \tau_k)y^k$ and $\eta_k = \frac{\beta_{k+1}\bar{r}_k^2 - \beta_k \bar{r}_{k-1}^2}{8a_{k+1}}$, it holds
$$f(y^{k+1}) \leq (1 - \tau_k)(f(x^{k+1}) + \langle \nabla f(x^{k+1}), y^k - x^{k+1} \rangle) + \tau_k(f(x^{k+1}) + \langle \nabla f(x^{k+1}), v^{k+1} - x^{k+1} \rangle)$$
$$+ \frac{\beta_{k+1}}{64\tau_k^2 A_{k+1}}\tau_k^2\|v^{k+1} - v^k\|^2 + \frac{\beta_{k+1}\bar{r}_k^2 - \beta_k \bar{r}_{k-1}^2}{16 A_{k+1}}$$
$$\leq (1 - \tau_k)f(y^k) + \tau_k(f(x^{k+1}) + \langle \nabla f(x^{k+1}), v^{k+1} - x^{k+1} \rangle)$$
$$+ \frac{\beta_{k+1}}{64 A_{k+1}}\|v^{k+1} - v^k\|^2 + \frac{\beta_{k+1}\bar{r}_k^2 - \beta_k \bar{r}_{k-1}^2}{16 A_{k+1}}.$$

Multiplying both sides by $A_{k+1}$, we have

$$A_{k+1}f(y^{k+1}) \leq A_k f(y^k) + a_{k+1}(f(x^{k+1}) + \langle \nabla f(x^{k+1}), v^{k+1} - x^{k+1}\rangle)$$
$$+ \frac{\beta_{k+1}}{64}\|v^{k+1} - v^k\|^2 + \frac{\beta_{k+1}\bar{r}_k^2 - \beta_k \bar{r}_{k-1}^2}{16}.$$

Note that

$$\|v^{k+1} - v^k\|^2 \leq (\|v^{k+1} - x^0\| + \|v^k - x^0\|)^2 \leq (2\max\{\|v^{k+1} - x^0\|, \|v^k - x^0\|\})^2 \leq 4\bar{r}_{k+1}^2.$$

It follows that

$$A_{k+1}f(y^{k+1}) \leq A_k f(y^k) + a_{k+1}(f(x^{k+1}) + \langle \nabla f(x^{k+1}), v^{k+1} - x^{k+1}\rangle)$$
$$+ \frac{\beta_k}{64}\|v^{k+1} - v^k\|^2 + \frac{\beta_{k+1} - \beta_k}{64}4\bar{r}_{k+1}^2 + \frac{\beta_{k+1}\bar{r}_k^2 - \beta_k\bar{r}_{k-1}^2}{16}$$
$$\leq A_k f(y^k) + a_{k+1}(f(x^{k+1}) + \langle \nabla f(x^{k+1}), v^{k+1} - x^{k+1}\rangle)$$
$$+ \frac{\beta_k}{2}\|v^{k+1} - v^k\|^2 + \frac{\beta_{k+1} - \beta_k}{16}\bar{r}_{k+1}^2 + \frac{\beta_{k+1}\bar{r}_k^2 - \beta_k\bar{r}_{k-1}^2}{16} \qquad (31)$$
$$\leq A_k f(y^k) + a_{k+1}(f(x^{k+1}) + \langle \nabla f(x^{k+1}), v^{k+1} - x^{k+1}\rangle)$$
$$+ \frac{\beta_k}{2}\|v^{k+1} - v^k\|^2 + \frac{\beta_{k+1}\bar{r}_{k+1}^2 - \beta_k\bar{r}_k^2}{16} + \frac{\beta_{k+1}\bar{r}_k^2 - \beta_k\bar{r}_{k-1}^2}{16},$$

where the last inequality uses $\bar{r}_{k+1} \geq \bar{r}_k$.

On the other hand, since $g(\cdot)$ is convex, we have

$$g(y^{k+1}) \leq (1 - \tau_k)g(y^k) + \tau_k g(v^{k+1}). \qquad (32)$$

Combining (31) and (32), we obtain

$$A_{k+1}\psi(y^{k+1}) \leq A_k\psi(y^k) + a_{k+1}(f(x^{k+1}) + \langle \nabla f(x^{k+1}), v^{k+1} - x^{k+1}\rangle + g(v^{k+1}))$$
$$+ \frac{\beta_k}{2}\|v^{k+1} - v^k\|^2 + \frac{\beta_{k+1}\bar{r}_{k+1}^2 - \beta_k\bar{r}_k^2}{16} + \frac{\beta_{k+1}\bar{r}_k^2 - \beta_k\bar{r}_{k-1}^2}{16},$$

For $\frac{\beta_k}{2}\|v^{k+1} - v^k\|^2$, we use Lemma 4, then

$$A_{k+1}\psi(y^{k+1}) \leq A_k\psi(y^k) + a_{k+1}(f(x^{k+1}) + \langle \nabla f(x^{k+1}), v^{k+1} - x^{k+1}\rangle + g(v^{k+1}))$$
$$+ \sum_{i=1}^{k} a_i(f(x^i) + \langle \nabla f(x^i), v^{k+1} - x^i\rangle + g(v^{k+1})) + \frac{\beta_k}{2}\|x^0 - v^{k+1}\|^2$$
$$- \sum_{i=1}^{k} a_i(f(x^i) + \langle \nabla f(x^i), v^k - x^i\rangle + g(v^k)) - \frac{\beta_k}{2}\|x^0 - v^k\|^2$$
$$+ \frac{\beta_{k+1}\bar{r}_{k+1}^2 - \beta_k\bar{r}_k^2}{16} + \frac{\beta_{k+1}\bar{r}_k^2 - \beta_k\bar{r}_{k-1}^2}{16}$$
$$\leq A_k\psi(y^k) + \sum_{i=1}^{k+1} a_i(f(x^i) + \langle \nabla f(x^i), v^{k+1} - x^i\rangle + g(v^{k+1})) + \frac{\beta_{k+1}}{2}\|x^0 - v^{k+1}\|^2$$
$$- \sum_{i=1}^{k} a_i(f(x^i) + \langle \nabla f(x^i), v^k - x^i\rangle + g(v^k)) - \frac{\beta_k}{2}\|x^0 - v^k\|^2$$
$$+ \frac{\beta_{k+1}\bar{r}_{k+1}^2 - \beta_k\bar{r}_k^2}{16} + \frac{\beta_{k+1}\bar{r}_k^2 - \beta_k\bar{r}_{k-1}^2}{16}.$$
$$(33)$$

We can shorten the inequality (33) by using the definition of $\phi_k(\cdot)$:

$$A_{k+1}\psi(y^{k+1}) \leq A_k\psi(y^k) + \phi_{k+1}(v^{k+1}) - \phi_k(v^k) + \frac{\beta_{k+1}\bar{r}_{k+1}^2 - \beta_k\bar{r}_k^2}{16} + \frac{\beta_{k+1}\bar{r}_k^2 - \beta_k\bar{r}_{k-1}^2}{16}.$$

Applying the upper inequality recursively, it holds

$$
\begin{aligned}
A_{k+1}\psi(y^{k+1}) \leq & \phi_{k+1}(v^{k+1}) - \phi_0(v^0) + \sum_{i=0}^{k}\frac{\beta_{i+1}\bar{r}_{i+1}^2 - \beta_i \bar{r}_i^2}{16} + \sum_{i=0}^{k}\frac{\beta_{i+1}\bar{r}_i^2 - \beta_i \bar{r}_{i-1}^2}{16} \\
\leq & \phi_{k+1}(v^{k+1}) + \frac{\beta_{k+1}}{16}\bar{r}_{k+1}^2 - \frac{\beta_0}{16}\bar{r}_0^2 + \frac{\beta_{k+1}}{16}\bar{r}_k^2 - \frac{\beta_0}{16}\bar{r}_{-1}^2 \\
\leq & \phi_{k+1}(v^{k+1}) + \frac{\beta_{k+1}}{16}\bar{r}_{k+1}^2 + \frac{\beta_{k+1}}{16}\bar{r}_{k+1}^2 \\
\leq & \phi_{k+1}(v^{k+1}) + \frac{\beta_{k+1}}{8}\bar{r}_{k+1}^2.
\end{aligned}
$$

where $\phi_0(v^0) = 0$ and $\beta_0 > 0$.

Since $v^{k+1} = \arg\min_x \phi_{k+1}(x)$, we use Lemma 4 again and obtain that:

$$
\begin{aligned}
A_{k+1}\psi(y^{k+1}) \leq & \phi_{k+1}(v^{k+1}) + \frac{\beta_{k+1}}{8}\bar{r}_{k+1}^2 \\
= & \sum_{i=1}^{k+1} a_i(f(x^i) + \langle \nabla f(x^i), v^k - x^i \rangle + g(v^k)) + \frac{\beta_{k+1}}{2}\|x^0 - v^{k+1}\|^2 + \frac{\beta_{k+1}}{8}\bar{r}_{k+1}^2 \\
\leq & \sum_{i=1}^{k+1} a_i(f(x^i) + \langle \nabla f(x^i), x^* - x^i \rangle + g(x^*)) + \frac{\beta_{k+1}}{2}\|x^0 - x^*\|^2 - \frac{\beta_{k+1}}{2}\|v^{k+1} - x^*\|^2 \\
& + \frac{\beta_{k+1}}{8}\bar{r}_{k+1}^2 \\
\leq & A_{k+1}\psi(x^*) + \frac{\beta_{k+1}}{2}\|x^0 - x^*\|^2 - \frac{\beta_{k+1}}{2}\|v^{k+1} - x^*\|^2 + \frac{\beta_{k+1}}{8}\bar{r}_{k+1}^2.
\end{aligned}
$$

Finally, we use $D_0$ and $D_{k+1}$ to replace $\|x^0 - x^*\|$ and $\|v^{k+1} - x^*\|$ and have

$$
\begin{aligned}
A_{k+1}\psi(y^{k+1}) & \leq A_{k+1}\psi(x^*) + \frac{\beta_{k+1}}{2}D_0^2 - \frac{\beta_{k+1}}{2}D_{k+1}^2 + \frac{\beta_{k+1}}{8}\bar{r}_{k+1}^2 \\
\psi(y^{k+1}) - \psi(x^*) & \leq \frac{\beta_k(D_0^2 - D_{k+1}^2)}{2A_{k+1}} + \frac{\beta_k \bar{r}_{k+1}^2}{8A_{k+1}}.
\end{aligned}
$$

$\square$

Note that the convergence result above is conditioned on the success of the line search, which further requires the boundedness of the iterates. We prove these important properties in the following lemma.

**Proof of Lemma 2**

*Proof.* For clarity, we divide the proof into the following parts.

**Part 1: Finite termination of the line search.**

Given the value of $x^k$, $y^k$, $A_{k+1}$, $\tau_k$, $\bar{r}_k$, $\bar{r}_{k-1}$, and $\beta_k$, $l_k(\beta)$ is defined by

$$
\begin{aligned}
l_k(\beta) := & -f(\tau_k v^{k+1}(\beta) + (1-\tau_k)y^k) + f(x^{k+1}) + \langle \nabla f(x^{k+1}), \tau_k v^{k+1}(\beta) \\
& + (1-\tau_k)y^k - x^{k+1} \rangle + \frac{\beta}{64\tau_k^2 A_{k+1}}\|\tau_k v^{k+1}(\beta) + (1-\tau_k)y^k - x^{k+1}\|^2 + \frac{\beta \bar{r}_k^2 - \beta_k \bar{r}_{k-1}^2}{16A_{k+1}},
\end{aligned}
\quad (34)
$$

where $v^{k+1}(\beta) := \arg\min_{x \in \mathbb{R}^d} \sum_{i=1}^{k+1} a_i(\langle \nabla f(x^i), x \rangle + g(x)) + \frac{\beta}{2}\|x - x^0\|^2, \beta \in \mathbb{R}_+$.

We analyze the function $v^{k+1}(\beta)$ and $l_k(\beta)$ first. $v^{k+1}(\beta) \in \mathrm{dom}\, g$ is well-defined and unique since $\sum_{i=1}^{k} a_i(\langle \nabla f(x^i), x \rangle + g(x)) + \frac{\beta}{2}\|x - x^0\|, \beta \in \mathbb{R}_+$ is strong convex and has a unique optimal solution. We claim that $g(x)$ restricted to $\mathrm{dom}\, g$ is continuous since it is convex and lsc. The convexity guarantees $g(x)$ is continuous at the interior point of $\mathrm{dom}\, g$, and lower semicontinuity guarantees

that it maintains the continuity on the remaining points of $\mathrm{dom}\, g$. Thus $v^k(\beta)$ is continuous. Since $l_k(\beta)$ is the composition of continuous functions, it is also continuous.

Next, we discuss the behavior of $l_k(\beta)$ when $\beta \to +\infty$. Recall that we assume $x^k, v^k, y^k \in \mathcal{B}_{3D_0}(x^*)$, we shall first prove that the line search for $y^{k+1}$ must be finitely terminated. Specifically, applying Lemma 7 with $c = \sum_{i=1}^{k+1} \nabla f(x^i)$ and $h = \frac{\beta}{2}$, we have that for a sufficiently large value $\hat{\beta}$, when $\beta \geq \hat{\beta}$, $\|x^0 - v^{k+1}(\beta)\| \leq 2D_0$, which further implies $\|x^* - v^{k+1}(\beta)\| \leq 3D_0$.

Let us consider $\beta \geq \beta_{k+1}^{\mathrm{TH}}$, $\delta > 0$, where

$$\beta_{k+1}^{\mathrm{TH}} := \max\left\{ \hat{\beta}, \frac{8A_{k+1}\delta + \beta_k \bar{r}_{k-1}^2}{\bar{r}_k^2}, 32\tau_k^2 A_{k+1}\gamma(\widehat{M_\nu}, \delta), \beta_k \right\}$$

we must have $v^{k+1}(\beta) \in \mathcal{B}_{3D_0}(x^*)$. Since $y^{k+1}(\beta)$ is a convex combination of $v^{k+1}(\beta)$ and $y^k$, we have $y^{k+1}(\beta) \in \mathcal{B}_{3D_0}(x^*)$. Similarly, we have $x^{k+1} \in \mathcal{B}_{3D_0}(x^*)$. Lemma 6 implies

$$f(y^{k+1}(\beta)) \leq f(x^{k+1}) + \langle \nabla f(x), y - x \rangle + \frac{\gamma(\widehat{M_\nu}, \delta)}{2}\|x - y\|^2 + \frac{\delta}{2}, \forall x, y \in \mathcal{B}_{3D_0}(x^*). \quad (35)$$

Moreover, due to the definition of $\beta_{k+1}^{\mathrm{TH}}$, we have

$$\frac{\gamma(\widehat{M_\nu}, \delta)}{2} \leq \frac{\beta}{64\tau_k^2 A_{k+1}}, \quad \text{and} \quad \frac{\delta}{2} \leq \frac{\beta \bar{r}_k^2 - \beta_k \bar{r}_{k-1}^2}{16A_{k+1}}.$$

Combining the above two results, we conclude that $l_k(\beta) \geq 0$. That is, $l_k(\beta)$ remains nonnegative when $\beta$ exceeds a certain threshold, and hence the search will terminate in finitely many steps.

Furthermore, we would like to point out that $2\beta_{k+1}^{\mathrm{TH}}$ is another threshold. For any $\beta \geq 2\beta_{k+1}^{\mathrm{TH}}$, we have $l_k(\beta) > 0$. The reason is that the following inequalities hold:

$$\frac{\gamma(\widehat{M_\nu}, \delta)}{2} \leq \frac{\beta}{64\tau_k^2 A_{k+1}} < \frac{2\beta}{64\tau_k^2 A_{k+1}}, \quad \text{and} \quad \frac{\delta}{2} \leq \frac{\beta \bar{r}_k^2 - \beta_k \bar{r}_{k-1}^2}{16A_{k+1}} < \frac{2\beta \bar{r}_k^2 - \beta_k \bar{r}_{k-1}^2}{16A_{k+1}}.$$

The second stage of the line search procedure also ends in finite steps as it employs a simple bisection method.

**Part 2: Boundedness of the $(k + 1)$-th iterates.**

First, we immediately have $x^{k+1} = \tau_k v^k + (1 - \tau_k)y^k \in \mathcal{B}_{3D_0}(x^*)$ by the assumption. Next, we prove $y^{k+1}, v^{k+1} \in \mathcal{B}_{3D_0}(x^*)$. Part 1 implies $l_i(\beta_{i+1}) \geq 0$ $(i = 0, 1, \ldots, k)$. Applying Lemma 1, we have

$$\psi(y^{k+1}) - \psi(x^*) \leq \frac{\beta_{k+1}(D_0^2 - D_{k+1}^2)}{2A_{k+1}} + \frac{\beta_{k+1}\bar{r}_{k+1}^2}{8A_{k+1}}. \quad (36)$$

We shall consider two cases.

**Case 1:** $\bar{r}_{k+1} = \bar{r}_k$, then $r_{k+1} \leq \bar{r}_k \leq 4D_0$;

**Case 2:** $\bar{r}_{k+1} = r_{k+1}$. Due to the non-negativity of the optimality gap and (36), we have

$$0 \leq \frac{\beta_{k+1}[D_0^2 - D_{k+1}^2]}{2A_{k+1}} + \frac{\beta_{k+1}r_{k+1}^2}{8A_{k+1}}.$$

By dividing both sides by $\frac{\beta_{k+1}}{2A_{k+1}}$, we obtain

$$0 \leq D_0^2 - D_{k+1}^2 + \frac{r_{k+1}^2}{4},$$

which implies:

$$D_{k+1} \leq \sqrt{D_0^2 + \frac{r_{k+1}^2}{4}} \leq D_0 + \frac{r_{k+1}}{2}.$$

By the triangle inequality, we have:

$$r_{k+1} \leq D_0 + D_{k+1} \leq D_0 + D_0 + \frac{r_{k+1}}{2}$$

$$r_{k+1} \leq 4D_0.$$

By repeatedly using the $D_k \leq D_0 + \frac{\bar{r}_k}{2}$, we have:

$$D_k \leq D_0 + \frac{r_k}{2} \leq D_0 + 2D_0 \leq 3D_0.$$

That is, $\|v^{k+1} - x^*\| \leq 3D_0$ and thus $v^{k+1} \in \mathcal{B}_{3D_0}(x^*)$. Thus, we have $y^{k+1} \in \mathcal{B}_{3D_0}(x^*)$ as well since $y^{k+1}$ is the convex combination of $v^{k+1}$ and $y^k$. $\qquad\square$

The following lemma develops an upper bound of $\beta_k$.

**Lemma 8.** *Suppose $f(\cdot)$ is locally Hölder smooth in $\mathcal{B}_{3D_0}(x^*)$ and $\beta_0 \leq 2^7 I_\nu \widehat{M}_\nu \bar{r}^\nu$. In Algorithm 1, for any $k \geq 0$, given $\epsilon^l_{k+1} > 0$, if at least one of the following two propositions holds:*

1. *$l_k(\beta_k) \geq 0$, in which case we set $\beta_{k+1} = \beta_k$;*

2. *there exists a root $\beta^*_{k+1}$ where $l_k(\beta^*_{k+1}) = 0$ and the line search returns a value satisfying $\beta_{k+1} \leq \beta^*_{k+1} + \epsilon^l_{k+1}$.*

*Then, we have*

$$\beta_k \leq 2^7 I_\nu \widehat{M}_\nu \bar{r}^\nu_{k-1} k^{\frac{3-3v}{2}} + \sum_{i=1}^{k} \epsilon^l_i. \tag{37}$$

*Proof.* First, we estimate the growth of $a_{k+1}$. We have

$$a_{k+1} = A_{k+1} - A_k = \left(A_{k+1}^{\frac{1}{2}} - A_k^{\frac{1}{2}}\right)\left(A_{k+1}^{\frac{1}{2}} + A_k^{\frac{1}{2}}\right) \leq 2\,\bar{r}_k^{\frac{1}{2}} A_{k+1}^{\frac{1}{2}},$$

which gives

$$a_{k+1}^2 \leq 4\bar{r}_k A_{k+1}.$$

Next, for $\beta^*_{k+1}$ that satisfies $l_k(\beta^*_{k+1}) = 0$, applying Lemma 1, we have

$$y^{k+1}(\beta^*_{k+1}) \in \mathcal{B}_{3D_0}(x^*), \tag{38}$$

where $y^{k+1}(\beta)$ is defined in the main text of the paper.

$l_k(\beta^*_{k+1}) = 0$ implies that

$$f(y^{k+1}(\beta^*_{k+1})) = f(x^{k+1}) + \langle \nabla f(x^{k+1}), y^{k+1}(\beta^*_{k+1}) - x^{k+1}\rangle$$
$$+ \frac{\beta^*_{k+1}}{64\tau_k^2 A_{k+1}}\|y^{k+1}(\beta^*_{k+1}) - x^{k+1}\|^2 + \frac{\beta^*_{k+1}\bar{r}_k^2 - \beta_k \bar{r}_{k-1}^2}{16A_{k+1}}. \tag{39}$$

Applying Lemma 6, we have

$$f(y^{k+1}(\beta^*_{k+1})) = f(x^{k+1}) + \langle \nabla f(x^{k+1}), y^{k+1}(\beta^*_{k+1}) - x^{k+1}\rangle$$
$$+ \frac{\gamma(\widehat{M}_\nu, \delta)}{2}\|y^{k+1}(\beta^*_{k+1}) - x^{k+1}\|^2 + \frac{\delta}{2}. \tag{40}$$

We take $\delta = \frac{\beta^*_{k+1}\bar{r}_k^2 - \beta_k \bar{r}_{k-1}^2}{8A_{k+1}}$, then combine (39) and (40), we have

$$\frac{\beta^*_{k+1}}{32\tau_k^2 A_{k+1}} \leq \gamma\left(\widehat{M}_\nu, \frac{\beta^*_{k+1}\bar{r}_k^2 - \beta_k \bar{r}_{k-1}^2}{8A_{k+1}}\right); \tag{41}$$

We prove this lemma by induction. By the assumption on $\beta_0$, it holds for $k = 0$. Next, we assume it is valid for some $k$.

**Case 1: The line search is satisfied by the previous step size ($\beta_{k+1} = \beta_k$).**

The inductive hypothesis is trivially satisfied for $k + 1$:

$$\beta_{k+1} = \beta_k \leq 2^7 I_\nu \widehat{M}_\nu \bar{r}_{k-1}^\nu k^{\frac{3-3v}{2}} + \sum_{i=1}^{k} \epsilon_i^l \leq 2^7 I_\nu \widehat{M}_\nu \bar{r}_k^\nu (k+1)^{\frac{3-3v}{2}} + \sum_{i=1}^{k+1} \epsilon_i^l,$$

where the final inequality holds because $\bar{r}_k$ is non-decreasing.

**Case 2: The line search requires a new step size ($\beta_{k+1} > \beta_k$).**

**Case 2.a:** $\beta_{k+1} \leq \beta_{k+1}^* + \epsilon_{k+1}^l$ and $\nu = 1$.

$$\frac{\beta_{k+1}^*}{32\tau_k^2 A_{k+1}} \leq \left(\frac{8A_{k+1}}{\beta_{k+1}^* \bar{r}_k^2 - \beta_k \bar{r}_{k-1}^2}\right)^0 \widehat{M}_\nu = \widehat{M}_\nu$$

$$\beta_{k+1}^* \leq 2^7 \widehat{M}_\nu \bar{r}_k = 2^7 I_\nu \widehat{M}_\nu \bar{r}_k$$

$$\beta_{k+1} \leq 2^7 I_\nu \widehat{M}_\nu \bar{r}_k + \epsilon_i^l \leq 2^7 I_\nu \widehat{M}_\nu \bar{r}_k + \sum_{i=1}^{k+1} \epsilon_i^l.$$

**Case 2.b:** $\beta_{k+1} \leq \beta_{k+1}^* + \epsilon_{k+1}^l$ and $\nu \neq 1$. First we analyze $\beta_{k+1}^*$. Since $\beta_{k+1}^*$ is the maximal zero point of $l_k(\cdot)$, applying Lemma 6, we have

$$\frac{\beta_{k+1}^*}{32\tau_k^2 A_{k+1}} \leq \gamma\left(\widehat{M}_\nu, \frac{\beta_{k+1}^* \bar{r}_k^2 - \beta_k \bar{r}_{k-1}^2}{8A_{k+1}}\right),$$

i.e.

$$\frac{\beta_{k+1}^*}{32\tau_k^2 A_{k+1}} \leq \left(\frac{1-\nu}{1+\nu} \frac{8A_{k+1}}{\beta_{k+1}^* \bar{r}_k^2 - \beta_k \bar{r}_{k-1}^2}\right)^{\frac{1-\nu}{1+\nu}} \widehat{M}_\nu^{\frac{2}{1+\nu}} \leq \left(\frac{8A_{k+1}}{(\beta_{k+1}^* - \beta_k)\bar{r}_k^2}\right)^{\frac{1-\nu}{1+\nu}} \widehat{M}_\nu^{\frac{2}{1+\nu}}.$$

It can be rewritten in the following form:

$$\beta_{k+1}^* (\beta_{k+1}^* - \beta_k)^{\frac{1-\nu}{1+\nu}} \leq 2^{\frac{10+4\nu}{1+\nu}} \bar{r}_k^{\frac{2\nu}{1+\nu}} (k+1)^{2\frac{1-\nu}{1+\nu}} \widehat{M}_\nu^{\frac{2}{1+\nu}}.$$

As $\beta_{k+1}$ increases with $\beta_{k+1} \geq \beta_k$, the left-hand side also increases. Thus, by identifying a value where the left-hand side is at most equal to the right-hand side, we can determine an upper bound for $\beta_{k+1}$.

Let $c_\nu = 2^7 \left(\frac{1}{1-\nu}\right)^{\frac{1+\nu}{2}} \widehat{M}_\nu$. For $c_\nu \bar{r}_k^\nu (k+1)^{\frac{3-3\nu}{2}} + \sum_{i=1}^{k} \epsilon_i^l$, we have

$$\left(c_\nu \bar{r}_k^\nu (k+1)^{\frac{3-3\nu}{2}} + \sum_{i=1}^{k} \epsilon_i^l\right)\left(c_\nu \bar{r}_k^\nu (k+1)^{\frac{3-3\nu}{2}} + \sum_{i=1}^{k} \epsilon_i^l - \beta_k\right)^{\frac{1-\nu}{1+\nu}}$$

$$\geq \left(c_\nu \bar{r}_k^\nu (k+1)^{\frac{3-3\nu}{2}} + \sum_{i=1}^{k} \epsilon_i^l\right)\left(c_\nu \bar{r}_k^\nu (k+1)^{\frac{3-3\nu}{2}} + \sum_{i=1}^{k} \epsilon_i^l - c_\nu \bar{r}_{k-1}^\nu k^{\frac{3-3\nu}{2}} - \sum_{i=1}^{k} \epsilon_i^l\right)^{\frac{1-\nu}{1+\nu}}$$

$$\geq \left(c_\nu \bar{r}_k^\nu (k+1)^{\frac{3-3\nu}{2}} + \sum_{i=1}^{k} \epsilon_i^l\right)\left(c_\nu \bar{r}_k^\nu (k+1)^{\frac{3-3\nu}{2}} - c_\nu \bar{r}_{k-1}^\nu k^{\frac{3-3\nu}{2}}\right)^{\frac{1-\nu}{1+\nu}} \tag{42}$$

$$\geq c_\nu \bar{r}_k^\nu (k+1)^{\frac{3-3\nu}{2}} \left(c_\nu \bar{r}_k^\nu (k+1)^{\frac{3-3\nu}{2}} - c_\nu \bar{r}_k^\nu k^{\frac{3-3\nu}{2}}\right)^{\frac{1-\nu}{1+\nu}}$$

$$\geq c_\nu^{\frac{2}{1+\nu}} \bar{r}_k^{\frac{2\nu}{1+\nu}} (k+1)^{\frac{3-3\nu}{2}} \left((k+1)^{\frac{3-3\nu}{2}} - k^{\frac{3-3\nu}{2}}\right)^{\frac{1-\nu}{1+\nu}}$$

Since $\frac{3-3v}{2} \in [0, \frac{3}{2}]$, and $\min_{1 \le u \le \frac{3}{2}} (\frac{1}{2})^{u-1} = 2^{-\frac{1}{2}} > \frac{1}{2}$, $\frac{3-3v}{2}(\frac{1}{2})^{\frac{3-3v}{2}-1} \ge \frac{3-3v}{4}$. Applying lemma 5, it holds that

$$
\begin{aligned}
& c_\nu^{\frac{2}{1+\nu}} \bar{r}_k^{\frac{2\nu}{1+\nu}} (k+1)^{\frac{3-3\nu}{2}} ((k+1)^{\frac{3-3\nu}{2}} - k^{\frac{3-3\nu}{2}})^{\frac{1-\nu}{1+\nu}} \\
\ge & \frac{3-3\nu}{4} c_\nu^{\frac{2}{1+\nu}} \bar{r}_k^{\frac{2\nu}{1+\nu}} (k+1)^{\frac{3-3\nu}{2}} ((k+1)^{\frac{1-3\nu}{2}})^{\frac{1-\nu}{1+\nu}} \\
\ge & \frac{3-3\nu}{4} (\frac{1}{1-\nu}) 2^{\frac{14}{1+\nu}} \widehat{M}_\nu^{\frac{2}{1+\nu}} \bar{r}_k^{\frac{2\nu}{1+\nu}} (k+1)^{\frac{3-3\nu}{2}} ((k+1)^{\frac{1-3\nu}{2}})^{\frac{1-\nu}{1+\nu}} \\
\ge & 2^{\frac{10+4\nu}{1+\nu}} \widehat{M}_\nu^{\frac{2\nu}{1+\nu}} \bar{r}_k^{\frac{2\nu}{1+\nu}} (k+1)^{2\frac{1-\nu}{1+\nu}}
\end{aligned}
\tag{43}
$$

This inequality implies that $\beta_{k+1}^* \le 2^7 I_\nu \widehat{M}_\nu \bar{r}_k^\nu (k+1)^{\frac{3-3\nu}{2}} + \sum_{i=1}^k \epsilon_i^l$ and $\beta_{k+1} \le \beta_{k+1}^* + \epsilon_{k+1}^l \le 2^7 I_\nu \widehat{M}_\nu \bar{r}_k^\nu (k+1)^{\frac{3-3\nu}{2}} + \sum_{i=1}^{k+1} \epsilon_i^l$. $\qquad\square$

Moreover, since we set $\epsilon_k^l = \frac{\beta_0}{2k^2}$, we can obtain an upper bound of $\beta_k$ as follows

$$
\beta_{k+1} \le 2^7 I_\nu \widehat{M}_\nu \bar{r}_k^\nu (k+1)^{\frac{3-3\nu}{2}} + \sum_{i=1}^{k+1} \frac{\beta_0}{2i^2} \le 2^7 I_\nu \widehat{M}_\nu \bar{r}_k^\nu (k+1)^{\frac{3-3\nu}{2}} + \beta_0 \le 2^8 I_\nu \widehat{M}_\nu \bar{r}_k^\nu (k+1)^{\frac{3-3\nu}{2}}.
$$

### C.3 Proof of Proposition Proposition 1

*Proof.* In the proof of Lemma 2, we have proved that the first stage of the line search terminates in a finite number of steps, and thus $l_k(2^{i_k'-1}\beta_k) \ge 0$. Moreover, we also proved that $l_k(\cdot)$ is continuous. Next, we show that at least one of the two propositions mentioned in Proposition 1 is correct.

When $i_k' = 1$, we have $l_k(2^{i_k'-1}\beta_k) = l_k(\beta_k) \ge 0$. Thus, the first proposition holds in this case.

When $i_k' > 1$, we have $l_k(2^{i_k'-1}\beta_k) \ge 0$ and $l_k(2^{i_k'-2}\beta_k) < 0$. Since $l_k(\cdot)$ is continuous, based on the intermediate value theorem, there exists at least one root in the interval $[2^{i_k'-2}\beta_k, 2^{i_k'-1}\beta_k]$. Moreover, as previously discussed in Lemma 2, there exists a threshold $2\beta^{\text{TH}}$ such that for any $\beta \ge 2\beta_{k+1}^{\text{TH}}$, $l_k(\beta) > 0$. Now, since $l_k(\beta)$ has at least one root and the set of roots has an upper bound, there exists a maximal root $\beta_{k+1}^*$ of the continuous function $l_k(\cdot)$, and therefore the second proposition holds.

It remains to estimate the upper bound of the amount of searching. Without loss of generality, we assume that $\beta_0 \le 2^7 I_\nu \widehat{M}_\nu \bar{r}^v$ in the Algorithm 1. This is a common assumption in the previous work, for example, see [27], and it is reasonable since the upper bound of the searched value increases polynomially in $k$. The initial value of the searched value is not very sensitive. Thus all the conditions of Lemma 8 are satisfied, we apply it to obtain that $\beta_{k+1} \le \mathcal{O}(k^{\frac{3-3\nu}{2}})$.

The first stage in the line search in the $k$-th iteration starts from $\beta_k$ to at most $2\beta_{k+1}$. So the length of the interval is at most $2\beta_{k+1} - \beta_k$ and this stage in line search procedure requires at most $i_k' \le 1 + \log(\frac{2\beta_{k+1}}{\beta_k})$ times in the $k$-th iteration. We sum them up and obtain that up to the $k$-th iteration, the total amount of line search operations in the first stage is at most

$$
\sum_{j=1}^k i_j' \le (1 + \log 2)k + \log(\frac{\beta_{k+1}}{\beta_0}) \le \mathcal{O}(k + \log k).
$$

The second step in line search process in the $k$-th iteration start from $2^{i_k-1}\beta_k$ to at most $2^{i_k}\beta_k$. Hence, the interval length is at most $2^{i_k}\beta_k - 2^{i_k-1}\beta_k = 2^{i_k-1}\beta_k \le \beta_{k+1}$ and this step of process requires at most $i_k^* - i_k' \le 1 + \log(\frac{\beta_{k+1}}{\epsilon_{k+1}^l})$ times in the $k$-th iteration. We sum them up and obtain that up to the $k$-th iteration, the total amount of line search operations in the first part is at most

$$
\sum_{j=1}^k i_j^* \le k + \sum_{i=1}^k \log\left(\frac{\beta_i}{\epsilon_i^l}\right) = k + \log\left(\prod_{i=1}^k \beta_i\right) - \log\left(\prod_{i=1}^k \epsilon_i^l\right) \le \mathcal{O}(k \log k),
$$

where we apply Lemma!8 to estimate $\beta_k$.

To summarize, the total amount of line search operations is $\mathcal{O}(k \log k)$. $\qquad\square$

## C.4 Proof of Theorem 1

*Proof.* We first prove the first part of this theorem. We apply Lemma 1 and 2 to prove it by induction. First, $x^0 = v^0 = y^0 \in \mathcal{B}_{3D_0}(x^*)$. Next, we assume it holds for some $k \geq 0$.

Since $x^k, v^k, y^k \in \mathcal{B}_{3D_0}(x^*)$, we have $l_k(\beta_{k+1}) \geq 0$ and $x^{k+1}, v^{k+1}, y^{k+1} \in \mathcal{B}_{3D_0}(x^*)$ by Lemma 2. Thus, applying Lemma 1, it holds that

$$\psi(y^{k+1}) - \psi(x^*) \leq \frac{\beta_{k+1}(D_0^2 - D_{k+1}^2)}{2A_{k+1}} + \frac{\beta_{k+1}\bar{r}_{k+1}^2}{8A_{k+1}}. \tag{44}$$

Therefore, for any $k > 0$, we have

$$\psi(y^k) - \psi(x^*) \leq \frac{\beta_k(D_0^2 - D_k^2)}{2A_k} + \frac{\beta_k\bar{r}_k^2}{8A_k}. \tag{45}$$

Then, we prove the second part of this Theorem. The proof is similar to that of Lemma 2. For completeness, we give a proof directly using the conclusion obtained in the first part and induction.

Since we assume $\bar{r} \leq 4D_0$, then $\bar{r}_0 = \bar{r} \leq 4D_0$. Next, we assume it holds for certain $k \geq 0$, then $x^{k+1} = \tau_k v^k + (1 - \tau_k)y^k$ lies in $\mathcal{B}_{3D_0}(x^*)$.

**Case 1:** $\bar{r}_{k+1} = \bar{r}_k$, then $r_{k+1} \leq \bar{r}_k \leq 4D_0$;

**Case 2:** $\bar{r}_{k+1} = r_{k+1}$. Applying the conclusion obtained in the first part, we have

$$0 \leq \psi(y^{k+1}) - \psi(x^*) \leq \frac{\beta_{k+1}(D_0^2 - D_{k+1}^2)}{2A_{k+1}} + \frac{\beta_{k+1}\bar{r}_{k+1}^2}{8A_{k+1}}. \tag{46}$$

Then it holds that

$$0 \leq D_0^2 - D_{k+1}^2 + \frac{r_{k+1}^2}{4}$$

$$D_{k+1}^2 \leq D_0^2 + \frac{r_{k+1}^2}{4}$$

$$D_{k+1} \leq \sqrt{D_0^2 + \frac{r_{k+1}^2}{4}} \leq D_0 + \frac{r_{k+1}}{2}.$$

By the triangle inequality, we have:

$$r_{k+1} \leq D_0 + D_{k+1} \leq D_0 + D_0 + \frac{r_{k+1}}{2}$$

$$r_{k+1} \leq 4D_0.$$

Repeat using the $D_k \leq D_0 + \frac{\bar{r}_k}{2}$, we have:

$$D_k \leq D_0 + \frac{r_k}{2} \leq D_0 + 2D_0 \leq 3D_0.$$

That is, $\|v^{k+1} - x^*\| \leq 3D_0$ and thus $v^{k+1} \in \mathcal{B}_{3D_0}(x^*)$. $y^{k+1} \in \mathcal{B}_{3D_0}(x^*)$ as well since $y^{k+1}$ is convex combination of $v^{k+1}$ and $y^k$.

In conclusion, we prove that for any $i \in \mathbb{N}$, $\|v^i - x^0\| \leq 4D_0$ and $\|v^i - x^*\| \leq 3D_0$. $\qquad\square$

The boundedness property is essential for us to remove the standard global Hölder smoothness assumption on the whole domain. Instead, we can safely use the local Hölder smoothness assumption as all the iterates remain in the ball $\mathcal{B}_{3D_0}(x^*)$.

Finally, all the preparatory work for Theorem 2 is now complete. Proposition 1 ensures the practical implementability of the algorithm, while the first part of Lemma 1 and Theorem 1 provides the tool needed to analyze the convergence rate. Furthermore, Theorem 1, Lemma 3, and Lemma 8 ensure that the convergence rate achieves the best-known rate.

## C.5 Proof of Theorem 2

*Proof.* Using the triangle inequality, it holds that

$$
\begin{aligned}
D_k &\geq |D_0 - r_k| \\
D_k^2 &\geq D_0^2 - 2D_0 r_k + r_k^2 \\
2D_0 r_k &\geq D_0^2 - D_k^2.
\end{aligned}
\tag{47}
$$

In view of Theorem 1, we have

$$
\psi(y^k) - \psi(x^*) \leq \frac{\beta_k [D_0^2 - D_k^2]}{2A_k} + \frac{\beta_k \bar{r}_k^2}{8A_k} \leq \frac{2\beta_k D_0 r_k}{2A_k} + \frac{\beta_k \bar{r}_k^2}{8A_k} \leq \frac{\beta_k D_0 \bar{r}_k}{A_k} + \frac{\beta_k \bar{r}_k^2}{8A_k},
\tag{48}
$$

where we use (47).

Applying Theorem 1, we can match the right hand side of inequality (48):

$$
\psi(y^k) - \psi(x^*) \leq \frac{\beta_k D_0 \bar{r}_k}{A_k} + \frac{4\beta_k D_0 \bar{r}_k}{8A_k} \leq \frac{3\beta_k D_0 \bar{r}_k}{2A_k},
\tag{49}
$$

Denote $k^* = \underset{0 \leq i \leq k}{\arg\min} \frac{\bar{r}_k^{\frac{1}{2}}}{\sum_{i=0}^{k-1} \bar{r}_i^{\frac{1}{2}}}$. Applying Lemma 3 gives

$$
\frac{\bar{r}_{k^*}^{\frac{1}{2}}}{\sum_{i=0}^{k^*-1} \bar{r}_i^{\frac{1}{2}}} \leq \frac{(\frac{\bar{r}_k}{\bar{r}_0})^{\frac{1}{2} \times \frac{1}{k}} \log e (\frac{\bar{r}_k}{\bar{r}_0})^{\frac{1}{2}}}{k} \leq \frac{(\frac{4D_0}{\bar{r}})^{\frac{1}{2k}} \log e \frac{4D_0}{\bar{r}}}{2k}.
\tag{50}
$$

Without loss of generality, we assume that $\beta_0 \leq 2^7 I_\nu \widehat{M}_\nu \bar{r}^\nu$ in Algorithm 1. This assumption is similar to the justification provided in the proof of Proposition 1.

Thus, for $k^*$, combining the inequality (50) and Lemma 8, it holds that

$$
\begin{aligned}
\min_{y \in \{y^0, y^1 \dots y^k\}} \psi(y) - \psi(x^*) &\leq \psi(y^{k^*}) - \psi(x^*) \\
&\leq \frac{3\beta_{k^*} D_0 \bar{r}_{k^*}}{2A_{k^*}} \\
&\leq \frac{3}{2} \beta_{k^*} D_0 \left( \frac{\bar{r}_{k^*}^{\frac{1}{2}}}{\sum_{i=0}^{k^*-1} \bar{r}_i^{\frac{1}{2}}} \right)^2 \\
&\leq \frac{3}{2} \times 2^8 I_\nu \widehat{M}_\nu D_0 \bar{r}_{k^*}^\nu k^{*\frac{3-3v}{2}} \left( \frac{(\frac{4D_0}{\bar{r}})^{\frac{1}{2k}} \log e \frac{4D_0}{\bar{r}}}{2k} \right)^2 \\
&\leq 384 I_\nu (\frac{4D_0}{\bar{r}})^{\frac{1}{k}} \log^2 e \frac{4D_0}{\bar{r}} \frac{\widehat{M}_\nu D_0 \bar{r}_{k^*}^\nu k^{*\frac{3-3v}{2}}}{k^2} \\
&\leq 384 I_\nu (\frac{4D_0}{\bar{r}})^{\frac{1}{k}} \log^2 e \frac{4D_0}{\bar{r}} \frac{\widehat{M}_\nu D_0^{1+\nu} k^{\frac{3-3v}{2}}}{k^2} \\
&\leq 384 I_\nu (\frac{4D_0}{\bar{r}})^{\frac{1}{k}} \log^2 e \frac{4D_0}{\bar{r}} \frac{\widehat{M}_\nu D_0^{1+\nu}}{k^{\frac{1+3v}{2}}},
\end{aligned}
\tag{51}
$$

where we use the fact $(k^*)^{\frac{3-3v}{2}} \leq k^{\frac{3-3v}{2}}$.

It remains to use that $\psi(z^k) = \min_{y \in \{y^0, y^1 \dots y^k\}} \psi(y)$ by definition. Since $(\frac{4D_0}{\bar{r}})^{\frac{1}{k}} \leq 2$ when $k \geq \log(\frac{4D_0}{\bar{r}})/\log 2$, the convergence rate of Algorithm 1 is

$$
\mathcal{O}\left( \frac{\widehat{M}_\nu D_0^{1+\nu} \log^2 e \frac{D_0}{\bar{r}}}{k^{\frac{1+3\nu}{2}}} \right).
\tag{52}
$$

$\square$

**Remark 5.** *In the proof of Theorem 2, we show that the convergence rate of Algorithm 1 has a multiplicative factor $I_\nu$. In fact, we can set $A_{k+1} = (\sum_{i=0}^{k} \bar{r}_i^{\frac{1}{n}})^n$, where $n \geq 2, n \in \mathbb{Z}_+$. By doing this, we can achieve a result similar to that of Theorem 2. If we choose $n \geq 3$, then the multiplicative factor $I_\nu$ will be replaced by another constant, which does not depend on $\nu$, as $I_\nu$ is generated by Lemma 5.*

## D   Missing details in Section 4

In this section, we provide the detailed proof of the results in Section 4 and conduct the convergence rate of Algorithm 2. For the stochastic case, we use the notation $-\xi_k$ to present the other random variables except $\xi_k$. The proofs are similar to the deterministic case, however, we do not need to prove the boundedness since we have assumed it under Assumption 1.

Lemma 9 and 10 are provided to analyze the balance equation to estimate of $\beta_k$.

**Lemma 9.** $\forall \alpha \geq 0, \beta > 0, \nu \in [0, 1)$, *we have*

$$\alpha r^{1+\nu} - \beta r^2 \leq \frac{1+\nu}{2}(\frac{1-\nu}{2})^{\frac{1+\nu}{1-\nu}}(\alpha^{\frac{2}{1-\nu}}/\beta^{\frac{1+\nu}{1-\nu}}), r \geq 0. \tag{53}$$

This auxiliary result has been used in [[34], Lemma E.3]. We give proof for completeness.

*Proof.* It is easy to see that $\alpha r^{1+\nu} - \beta r^2$ increases first and then decreases later as $r \geq 0$ increases. It achieves maximum on $[0, +\infty)$ iff its gradient equals to zero, i.e $r = (\frac{(1-\nu)\alpha}{2\beta})^{\frac{1}{1-\nu}}$.

Thus, for $r \geq 0$,

$$
\begin{aligned}
\alpha r^{1+\nu} - \beta r^2 &\leq \alpha((\frac{(1-\nu)\alpha}{2\beta})^{\frac{1}{1-\nu}})^{1+\nu} - \beta((\frac{(1-\nu)\alpha}{2\beta})^{\frac{1}{1-\nu}})^2 \\
&\leq \alpha(\frac{(1-\nu)\alpha}{2\beta})^{\frac{1+\nu}{1-\nu}} - \beta(\frac{(1-\nu)\alpha}{2\beta})^{\frac{2}{1-\nu}} \\
&\leq (\frac{1-\nu}{2})^{\frac{1+\nu}{1-\nu}} \frac{\alpha^{\frac{2}{1-\nu}}}{\beta^{\frac{1+\nu}{1-\nu}}} - (\frac{1-\nu}{2})^{\frac{2}{1-\nu}} \frac{\alpha^{\frac{2}{1-\nu}}}{\beta^{\frac{1+\nu}{1-\nu}}} \\
&\leq (\frac{1+\nu}{2})(\frac{1-\nu}{2})^{\frac{1+\nu}{1-\nu}} \frac{\alpha^{\frac{2}{1-\nu}}}{\beta^{\frac{1+\nu}{1-\nu}}}.
\end{aligned}
$$

$\square$

**Lemma 10.** *For nonnegative sequences $\{\alpha_i\}_{i\in\mathbb{N}}$ and $\{\gamma_i\}_{i\in\mathbb{N}}$, the sequence $\{h_i\}_{i\in\mathbb{N}}$ satisfies that*

$$h_{k+1} - h_k \leq \frac{(1-\nu)\alpha_{k+1}}{h_{k+1}^{\frac{\nu}{1-\nu}}} + \gamma_{k+1}, \tag{54}$$

*with $h_0 = 0$. Then for $k \geq 1$, we have*

$$h_k \leq (\sum_{i=1}^{k} \alpha_i)^{1-\nu} + \sum_{i=1}^{k} \gamma_i. \tag{55}$$

This auxiliary result has been used in [[34], Lemma E.9]. We give proof for completeness.

*Proof.* We prove it by induction.

Since $h_0 = 0 \leq 0$, we assume it is valid for some $k \geq 0$, then

$$
\begin{aligned}
h_{k+1} - \frac{(1-\nu)\alpha_{k+1}}{h_{k+1}^{\frac{\nu}{1-\nu}}} &\leq \gamma_{k+1} + h_k \\
&\leq \gamma_{k+1} + (\sum_{i=1}^{k} \alpha_i)^{1-\nu} + \sum_{i=1}^{k} \gamma_i \\
&\leq (\sum_{i=1}^{k} \alpha_i)^{1-\nu} + \sum_{i=1}^{k+1} \gamma_i.
\end{aligned}
$$

Define $\Gamma_{k+1}(x) := x - \frac{(1-\nu)\alpha_{k+1}}{x^{\frac{\nu}{1-\nu}}}$. It is easy to verify that $\Gamma_{k+1}(x)$ is increasing in $x$. Hence, it suffices to show

$$(\sum_{i=1}^{k} \alpha_i)^{1-\nu} + \sum_{i=1}^{k+1} \gamma_i \leq \Gamma_{k+1}((\sum_{i=1}^{k+1} \alpha_i)^{1-\nu} + \sum_{i=1}^{k+1} \gamma_i),$$

which means

$$(\sum_{i=1}^{k} \alpha_i)^{1-\nu} + \sum_{i=1}^{k+1} \gamma_i \leq (\sum_{i=1}^{k+1} \alpha_i)^{1-\nu} + \sum_{i=1}^{k+1} \gamma_i - (1-\nu)\alpha_{k+1}((\sum_{i=1}^{k+1} \alpha_i)^{1-\nu} + \sum_{i=1}^{k+1} \gamma_i)^{\frac{-\nu}{1-\nu}}. \quad (56)$$

Rearranging (56) gives us

$$(1-\nu)\alpha_{k+1}((\sum_{i=1}^{k+1} \alpha_i)^{1-\nu} + \sum_{i=1}^{k+1} \gamma_i)^{\frac{-\nu}{1-\nu}} \leq (\sum_{i=1}^{k+1} \alpha_i)^{1-\nu} - (\sum_{i=1}^{k} \alpha_i)^{1-\nu},$$

which is implied by

$$(1-\nu)\alpha_{k+1} \leq ((\sum_{i=1}^{k+1} \alpha_i)^{1-\nu} - (\sum_{i=1}^{k} \alpha_i)^{1-\nu})(\sum_{i=1}^{k+1} \alpha_i)^{\nu}. \quad (57)$$

The inequality (57) is valid since

$$((\sum_{i=1}^{k+1} \alpha_i)^{1-\nu} - (\sum_{i=1}^{k} \alpha_i)^{1-\nu})(\sum_{i=1}^{k+1} \alpha_i)^{\nu}$$

$$\geq ((\sum_{i=1}^{k+1} \alpha_i)^{1-\nu} - (\sum_{i=1}^{k+1} \alpha_i)^{1-\nu} + (1-\nu)(\sum_{i=1}^{k+1} \alpha_i)^{-\nu} a_{k+1})(\sum_{i=1}^{k+1} \alpha_i)^{\nu}$$

$$= ((1-\nu)(\sum_{i=1}^{k+1} \alpha_i)^{-\nu} a_{k+1})(\sum_{i=1}^{k+1} \alpha_i)^{\nu}$$

$$= (1-\nu)a_{k+1},$$

where the first inequality is due to the first order condition of the concave function $(\cdot)^{1-\nu}, \nu \in [0,1]$. $\qquad \square$

Next, we provide the upper bound of the expectation of $\beta_k$.

**Lemma 11.** *Suppose the Assumption 1 holds and $f(\cdot)$ is locally Hölder smooth in $\mathcal{B}_{3D_0}(x^*)$. In Algorithm 2, it holds that*

$$\mathbb{E}[\beta_k] \leq 2^{\frac{9+9\nu}{2}} \tilde{M}_\nu D^\nu k^{\frac{3-3\nu}{2}} + 2^5 k^{\frac{3}{2}} \sigma. \quad (58)$$

*Proof.* We denote $\|\nabla f(x^{k+1}) - \tilde{\nabla} f(x^{k+1})\|_* = \Delta_{k+1}^x$ and $\|\nabla f(y^{k+1}) - \tilde{\nabla} f(y^{k+1})\|_* = \Delta_{k+1}^y$.

The expectation of $(\Delta_k^x)^2$ and $(\Delta_k^y)^2$ satisfies

$$\begin{aligned}
\mathbb{E}[(\Delta_k^x)^2] &= \mathbb{E}[\|\nabla f(x^{k+1}) - \tilde{\nabla} f(x^{k+1})\|_*^2] \leq \sigma^2, \\
\mathbb{E}[(\Delta_k^y)^2] &= \mathbb{E}[\|\nabla f(y^{k+1}) - \tilde{\nabla} f(y^{k+1})\|_*^2] \leq \sigma^2.
\end{aligned} \quad (59)$$

From the balance equation, we have

$$\begin{aligned}
\frac{\tau_k \eta_k}{2} =& [\langle \tilde{\nabla} f(y^{k+1}) - \tilde{\nabla} f(x^{k+1}), y^{k+1} - x^{k+1}\rangle - \frac{\beta_{k+1}}{64\tau_k^2 A_{k+1}}\|y^{k+1} - x^{k+1}\|^2]_+ \\
\leq& [\langle \nabla f(y^{k+1}) - \nabla f(x^{k+1}), y^{k+1} - x^{k+1}\rangle + \langle \nabla f(x^{k+1}) - \tilde{\nabla} f(x^{k+1}), y^{k+1} - x^{k+1}\rangle \\
& + \langle \tilde{\nabla} f(y^{k+1}) - \nabla f(y^{k+1}), y^{k+1} - x^{k+1}\rangle - \frac{\beta_{k+1}}{64\tau_k^2 A_{k+1}}\|y^{k+1} - x^{k+1}\|^2]_+, \\
\leq& [\tilde{M}_\nu \|y^{k+1} - x^{k+1}\|^{1+\nu} + (\Delta_{k+1}^y + \Delta_{k+1}^x)\|y^{k+1} - x^{k+1}\| - \frac{\beta_{k+1}}{64\tau_k^2 A_{k+1}}\|y^{k+1} - x^{k+1}\|^2]_+.
\end{aligned}$$

Note that $[\cdot]_+$ is a monotonically increasing function. The first inequality uses the convexity of $f(\cdot)$ and the second inequality applies the locally Hölder smoothness and Cauchy-Schwarz inequality.

**Case 1:** $\nu = 1$

$$\frac{\beta_{k+1}\bar{r}_k^2 - \beta_k \bar{r}_{k-1}^2}{2A_{k+1}} \leq [(\tilde{M}_\nu - \frac{\beta_{k+1}}{64\tau_k^2 A_{k+1}})\|y^{k+1} - x^{k+1}\|^2 + (\Delta_{k+1}^y + \Delta_{k+1}^x)\|y^{k+1} - x^{k+1}\|]_+$$

$$\beta_{k+1}\bar{r}_k^2 - \beta_k \bar{r}_{k-1}^2 \leq 2A_{k+1}[(\tilde{M}_\nu - \frac{\beta_{k+1}}{64\tau_k^2 A_{k+1}})\|y^{k+1} - x^{k+1}\|^2 + (\Delta_{k+1}^y + \Delta_{k+1}^x)\|y^{k+1} - x^{k+1}\|]_+$$

$$\beta_{k+1}\bar{r}_k^2 - \beta_k \bar{r}_{k-1}^2 \leq 2A_{k+1}[(\tilde{M}_\nu - \frac{\beta_{k+1}}{2^8\bar{r}_k})\|y^{k+1} - x^{k+1}\|^2 + (\Delta_{k+1}^y + \Delta_{k+1}^x)\|y^{k+1} - x^{k+1}\|]_+$$

$$\beta_{k+1}\bar{r}_k^2 - \beta_k \bar{r}_{k-1}^2 \leq 2(k+1)^2[\bar{r}_k(\tilde{M}_\nu - \frac{\beta_{k+1}}{2^8\bar{r}_k})\|y^{k+1} - x^{k+1}\|^2 + \bar{r}_k(\Delta_{k+1}^y + \Delta_{k+1}^x)\|y^{k+1} - x^{k+1}\|]_+$$

$$\beta_{k+1} - \beta_k \leq \frac{2(k+1)^2}{\bar{r}_k}[(\tilde{M}_\nu - \frac{\beta_{k+1}}{2^8\bar{r}_k})\|y^{k+1} - x^{k+1}\|^2 + (\Delta_{k+1}^y + \Delta_{k+1}^x)\|y^{k+1} - x^{k+1}\|]_+,$$

where we use $A_{k+1} \leq (k+1)^2 \bar{r}_k$ and $D \geq \bar{r}_{k+1} \geq \bar{r}_k$.

We prove that

$$\beta_k^2 \leq \sum_{i=1}^{k} 2^{10} i^2 ((\Delta_i^x)^2 + (\Delta_i^y)^2) + 2^{18}\tilde{M}_\nu^2 D^2. \tag{60}$$

Since $\beta_0^2 = 0 < 2^{18}\tilde{M}_\nu^2 D^2$, we prove it by induction. Define $k^* = \max\{i|\beta_i \leq 2^9 \tilde{M}_\nu D\} \geq 0$, so $\forall k \leq k^*$ satisfies the inequality 60. We assume it is valid for certain $k \geq k^*$, then

$$\beta_{k+1} - \beta_k \leq \frac{1}{\bar{r}_k}[2(k+1)^2(\frac{\beta_{k+1}}{2^9 D} - \frac{\beta_{k+1}}{2^8\bar{r}_k})\|y^{k+1} - x^{k+1}\|^2 + 2(k+1)^2(\Delta_{k+1}^x + \Delta_{k+1}^y)\|y^{k+1} - x^{k+1}\|]_+$$

$$\leq \frac{1}{\bar{r}_k}[-2(k+1)^2\frac{\beta_{k+1}}{2^9\bar{r}_k}\|y^{k+1} - x^{k+1}\|^2 + 2(k+1)^2(\Delta_{k+1}^x + \Delta_{k+1}^y)\|y^{k+1} - x^{k+1}\|]_+$$

$$\leq \frac{1}{\bar{r}_k}2(k+1)^2\frac{2^9\bar{r}_k(\Delta_{k+1}^x)^2}{2\beta_{k+1}} + \frac{1}{\bar{r}_k}2(k+1)^2\frac{2^9\bar{r}_k(\Delta_{k+1}^y)^2}{2\beta_{k+1}}$$

$$= 2^9(k+1)^2\frac{(\Delta_{k+1}^x)^2 + (\Delta_{k+1}^y)^2}{\beta_{k+1}}.$$

Then

$$\frac{1}{2}(\beta_{k+1}^2 - \beta_k^2) \leq \beta_{k+1}(\beta_{k+1} - \beta_k) \leq 2^9(k+1)^2(\Delta_{k+1}^y + \Delta_{k+1}^x)^2$$

$$\beta_{k+1}^2 \leq 2^{10}(k+1)^2(\Delta_{k+1}^y + \Delta_{k+1}^x)^2 + \beta_k^2$$

$$\beta_{k+1}^2 \leq 2^{10}(k+1)^2(\Delta_{k+1}^y + \Delta_{k+1}^x)^2 + \beta_k^2$$

$$\beta_{k+1}^2 \leq 2^{10}(k+1)^2(\Delta_{k+1}^y + \Delta_{k+1}^x)^2 + \sum_{i=1}^{k} 2^{10} i^2 (\Delta_i^y + \Delta_i^x)^2 + 2^{18}\tilde{M}_\nu^2 D^2$$

$$\beta_{k+1}^2 \leq \sum_{i=1}^{k+1} 2^{10} i^2 (\Delta_i^y + \Delta_i^x)^2 + 2^{18}\tilde{M}_\nu^2 D^2,$$

where we use $\beta_k^2 \leq \sum_{i=1}^{k} 2^{10} i^2 \Delta_i^2 + 2^{18}\tilde{M}_\nu^2 D^2$ by induction. Applying the inequality $a^2 + b^2 \leq (a+b)^2$ where $a, b \geq 0$, we obtain

$$\beta_{k+1} \leq 2^5(\sum_{i=1}^{k+1} i^2(\Delta_i^x + \Delta_i^y)^2)^{\frac{1}{2}} + 2^9 \tilde{M}_\nu D.$$

We take the expectation of $\beta_k$:

$$\mathbb{E}[\beta_k] \leq \mathbb{E}[2^5(\sum_{i=1}^{k} i^2(\Delta_i^x + \Delta_i^y)^2))^{\frac{1}{2}} + 2^9 \tilde{M}_\nu D]$$

$$\leq 2^5(\sum_{i=1}^{k} i^2(\mathbb{E}[(\Delta_i^x)^2] + \mathbb{E}[(\Delta_i^y)^2])^{\frac{1}{2}} + 2^9 \tilde{M}_\nu D$$

$$\leq 2^5 \sum_{i=1}^{k} \sqrt{2} i^2 \sigma + 2^9 \tilde{M}_\nu D$$

$$\leq 2^{\frac{11}{2}} k^{\frac{3}{2}} \sigma + 2^9 \tilde{M}_\nu D,$$

where we use the Jensen's inequality that $\mathbb{E}[X^{\frac{1}{2}}] \leq (\mathbb{E}[X])^{\frac{1}{2}}$ and estimate $\sum_{i=1}^{k} i^2 \leq k^3$ roughly.

**Case 2:** $\nu \neq 1$ Applying Lemma 9 for three times with $r = \|y^{k+1} - x^{k+1}\|$, $\alpha = \tilde{M}_\nu$, $\beta = \frac{\beta_{k+1}}{128\tau_k^2 A_{k+1}}$; $r' = \|y^{k+1} - x^{k+1}\|$, $\alpha' = \Delta_{k+1}^x$, $\beta' = \frac{\beta_{k+1}}{256\tau_k^2 A_{k+1}}$; $r'' = \|y^{k+1} - x^{k+1}\|$, $\alpha'' = \Delta_{k+1}^y$, $\beta'' = \frac{\beta_{k+1}}{256\tau_k^2 A_{k+1}}$, it holds that

$$\frac{\tau_k \eta_k}{2} \leq [\frac{1+\nu}{2}(\frac{1-\nu}{2})^{\frac{1+\nu}{1-\nu}} \tilde{M}_\nu^{\frac{2}{1-\nu}} (\frac{128\tau_k^2 A_{k+1}}{\beta_{k+1}})^{\frac{1+\nu}{1-\nu}} + \frac{1}{2}(\Delta_{k+1}^y + \Delta_{k+1}^x)^2 \frac{256\tau_k^2 A_{k+1}}{\beta_{k+1}}] +$$

$$= \frac{1+\nu}{2}(\frac{1-\nu}{2})^{\frac{1+\nu}{1-\nu}} \tilde{M}_\nu^{\frac{2}{1-\nu}} (\frac{128\tau_k^2 A_{k+1}}{\beta_{k+1}})^{\frac{1+\nu}{1-\nu}} + (\Delta_{k+1}^y + \Delta_{k+1}^x)^2 \frac{128\tau_k^2 A_{k+1}}{\beta_{k+1}},$$

where the last equality is obviously nonnegative.

Similar to the proof of Lemma 8, we have

$$a_{k+1}^2 \leq 4\bar{r}_k A_{k+1},$$

which implies $\tau_k^2 A_{k+1} \leq 4\bar{r}_k$. Consequently,

$$\frac{\tau_k \eta_k}{2} \leq \frac{1+\nu}{2} \tilde{M}_\nu^{\frac{2}{1-\nu}} (\frac{(1-\nu)2^8 \bar{r}_k}{\beta_{k+1}})^{\frac{1+\nu}{1-\nu}} + (\Delta_{k+1}^y + \Delta_{k+1}^x)^2 \frac{2^9 \bar{r}_k}{\beta_{k+1}}$$

$$\beta_{k+1}\bar{r}_k^2 - \beta_k \bar{r}_{k-1}^2 \leq A_{k+1} \tilde{M}_\nu^{\frac{2}{1-\nu}} (\frac{(1-\nu)2^8 \bar{r}_k}{\beta_{k+1}})^{\frac{1+\nu}{1-\nu}} + A_{k+1}(\Delta_{k+1}^y + \Delta_{k+1}^x)^2 \frac{2^{10} \bar{r}_k}{\beta_{k+1}}$$

$$\beta_{k+1}\bar{r}_k^2 - \beta_k \bar{r}_k^2 \leq (k+1)^2 \bar{r}_k \tilde{M}_\nu^{\frac{2}{1-\nu}} (\frac{(1-\nu)2^8 \bar{r}_k}{\beta_{k+1}})^{\frac{1+\nu}{1-\nu}} + (k+1)^2 \bar{r}_k(\Delta_{k+1}^y + \Delta_{k+1}^x)^2 \frac{2^{10} \bar{r}_k}{\beta_{k+1}}$$

$$\beta_{k+1} - \beta_k \leq (k+1)^2 \tilde{M}_\nu^{\frac{2}{1-\nu}} (\frac{(1-\nu)2^8}{\beta_{k+1}})^{\frac{1+\nu}{1-\nu}} \bar{r}_k^{\frac{2\nu}{1-\nu}} + (k+1)^2(\Delta_{k+1}^y + \Delta_{k+1}^x)^2 \frac{2^{10}}{\beta_{k+1}},$$

where we use $A_{k+1} \leq (k+1)^2 \bar{r}_k$ and $\bar{r}_k \geq \bar{r}_{k-1}$. It then follows that

$$\beta_{k+1}(\beta_{k+1} - \beta_k) \leq (k+1)^2 \tilde{M}_\nu^{\frac{2}{1-\nu}} \frac{((1-\nu)2^8)^{\frac{1+\nu}{1-\nu}}}{\beta_{k+1}^{\frac{2\nu}{1-\nu}}} \bar{r}_k^{\frac{2\nu}{1-\nu}} + 2^{10}(k+1)^2 \Delta_{k+1}^2$$

$$\beta_{k+1}^2 - \beta_k^2 \leq 2(k+1)^2 \tilde{M}_\nu^{\frac{2}{1-\nu}} \frac{((1-\nu)2^8)^{\frac{1+\nu}{1-\nu}}}{\beta_{k+1}^{\frac{2\nu}{1-\nu}}} \bar{r}_k^{\frac{2\nu}{1-\nu}} + 2^{11}(k+1)^2 \Delta_{k+1}^2.$$

We apply Lemma 10 with $h_k = \beta_k^2$, $\alpha_k = 2 \times (2^8)^{\frac{1+\nu}{1-\nu}} k^2 \tilde{M}_\nu^{\frac{2}{1-\nu}} (1-\nu)^{\frac{2\nu}{1-\nu}} \bar{r}_{k-1}^{\frac{2\nu}{1-\nu}}$ and $\gamma_k = 2^{11}(k+1)^2 \Delta_k^2$, it holds that

$$\beta_k^2 \leq (\sum_{i=1}^{k} 2 \times (2^8)^{\frac{1+\nu}{1-\nu}} i^2 \tilde{M}_\nu^{\frac{2}{1-\nu}} (1-\nu)^{\frac{2\nu}{1-\nu}} \bar{r}_{i-1}^{\frac{2\nu}{1-\nu}})^{1-\nu} + \sum_{i=1}^{k} 2^{11} i^2 \Delta_k^2$$

$$\leq (2^{9+7\nu}) \tilde{M}_\nu^2 (1-\nu)^{2\nu} \bar{r}_{k-1}^{2\nu} (\sum_{i=1}^{k} i^2)^{1-\nu} + 2^{11} \sum_{i=1}^{k} i^2 \Delta_i^2$$

$$\leq (2^{9+7\nu}) \tilde{M}_\nu^2 (1-\nu)^{2\nu} D^{2\nu} k^{3-3\nu} + 2^{11} \sum_{i=1}^{k} i^2 \Delta_i^2.$$

Here we estimate $\sum_{i=1}^{k} i^2 \leq k^3$ roughly. Applying the inequality $a^2 + b^2 \leq (a+b)^2$ where $a, b \geq 0$, we obtain

$$\beta_k \leq (2^{\frac{9+7\nu}{2}})\tilde{M}_\nu (1-\nu)^\nu D^\nu k^{\frac{3-3\nu}{2}} + 2^{\frac{11}{2}}(\sum_{i=1}^{k} i^2 \Delta_i^2)^{\frac{1}{2}}.$$

Finally, we take the expectation of $\beta_k$:

$$\mathbb{E}[\beta_k] \leq \mathbb{E}[(2^{\frac{9+7\nu}{2}})\tilde{M}_\nu (1-\nu)^\nu D^\nu k^{\frac{3-3\nu}{2}} + 2^{\frac{11}{2}}(\sum_{i=1}^{k} i^2 \Delta_i^2)^{\frac{1}{2}}]$$

$$\leq (2^{\frac{9+7\nu}{2}})\tilde{M}_\nu (1-\nu)^\nu D^\nu k^{\frac{3-3\nu}{2}} + 2^{\frac{11}{2}}(\sum_{i=1}^{k} i^2 \mathbb{E}[\Delta_i^2])^{\frac{1}{2}}$$

$$\leq (2^{\frac{9+7\nu}{2}})\tilde{M}_\nu (1-\nu)^\nu D^\nu k^{\frac{3-3\nu}{2}} + 2^{\frac{11}{2}}\sigma(\sum_{i=1}^{k} i^2)^{\frac{1}{2}}$$

$$\leq (2^{\frac{9+7\nu}{2}})\tilde{M}_\nu (1-\nu)^\nu D^\nu k^{\frac{3-3\nu}{2}} + 2^{\frac{11}{2}} k^{\frac{3}{2}}\sigma,$$

where we use Jensen's inequality again.

In conclusion, we have $\mathbb{E}[\beta_k] \leq 2^{\frac{9+9\nu}{2}}\tilde{M}_\nu D^\nu k^{\frac{3-3\nu}{2}} + 2^{\frac{11}{2}} k^{\frac{3}{2}}\sigma.$ $\qquad\square$

We are now ready to derive the convergence rate of Algorithm 2.

### D.1  Proof of Theorem 3

*Proof.* In view of the balance equation (16) and the fact $[x]_+ \geq x$, it holds that

$$0 \leq \langle \tilde{\nabla} f(x^{k+1}) - \tilde{\nabla} f(y^{k+1}), y^{k+1} - x^{k+1} \rangle + \frac{\beta_{k+1}}{64\tau_k^2 A_{k+1}}\|y^{k+1} - x^{k+1}\|^2 + \frac{\tau_k \eta_k}{2}$$

$$\langle \nabla f(y^{k+1}), y^{k+1} - x^{k+1} \rangle \leq \langle \tilde{\nabla} f(x^{k+1}), y^{k+1} - x^{k+1} \rangle + \frac{\beta_{k+1}}{64\tau_k^2 A_{k+1}}\|y^{k+1} - x^{k+1}\|^2 + \frac{\tau_k \eta_k}{2}$$
$$+ \langle \nabla f(y^{k+1}) - \tilde{\nabla} f(y^{k+1}), y^{k+1} - x^{k+1} \rangle$$

The first order condition of convex function $f(\cdot)$ implies $f(y^{k+1}) - f(x^{k+1}) \leq \langle \nabla f(y^{k+1}), y^{k+1} - x^{k+1} \rangle$, thus

$$f(y^{k+1}) \leq f(x^{k+1}) + \langle \tilde{\nabla} f(x^{k+1}), y^{k+1} - x^{k+1} \rangle + \frac{\beta_{k+1}}{64\tau_k^2 A_{k+1}}\|y^{k+1} - x^{k+1}\|^2 + \frac{\tau_k \eta_k}{2}$$
$$+ \langle \nabla f(y^{k+1}) - \tilde{\nabla} f(y^{k+1}), y^{k+1} - x^{k+1} \rangle.$$

Because $x^{k+1} = \tau_k v^k + (1-\tau_k)y^k$, $y^{k+1} = \tau_k \hat{x}^{k+1} + (1-\tau_k)y^k$ and $\eta_k = \frac{\beta_{k+1}\bar{r}_k^2 - \beta_k \bar{r}_{k-1}^2}{8a_{k+1}}$, it holds that

$$f(y^{k+1}) \leq (1-\tau_k)(f(x^{k+1}) + \langle \tilde{\nabla} f(x^{k+1}), y^k - x^{k+1} \rangle) + \tau_k(f(x^{k+1}) + \langle \tilde{\nabla} f(x^{k+1}), \hat{x}^{k+1} - x^{k+1} \rangle)$$
$$+ \frac{\beta_{k+1}}{64\tau_k^2 A_{k+1}}\tau_k^2\|\hat{x}^{k+1} - v^k\|^2 + \frac{\beta_{k+1} - \beta_k}{16A_{k+1}}\bar{r}_k^2$$
$$+ \langle \nabla f(y^{k+1}) - \tilde{\nabla} f(y^{k+1}), y^{k+1} - x^{k+1} \rangle$$
$$\leq (1-\tau_k)f(y^k) + \tau_k(f(x^{k+1}) + \langle \tilde{\nabla} f(x^{k+1}), \hat{x}^{k+1} - x^{k+1} \rangle)$$
$$+ \frac{\beta_k}{64A_{k+1}}\|\hat{x}^{k+1} - v^k\|^2 + \frac{\beta_{k+1} - \beta_k}{16A_{k+1}}\bar{r}_k^2$$
$$+ (1-\tau_k)\langle \tilde{\nabla} f(x^{k+1}) - \nabla f(x^{k+1}), y^k - x^{k+1} \rangle + \frac{\beta_{k+1} - \beta_k}{64A_{k+1}}\|\hat{x}^{k+1} - v^k\|^2$$
$$+ \langle \nabla f(y^{k+1}) - \tilde{\nabla} f(y^{k+1}), y^{k+1} - x^{k+1} \rangle.$$

Since $a_{k+1}\langle\tilde{\nabla}f(x^{k+1}),\hat{x}^{k+1}\rangle + \frac{\beta_k}{2}\|\hat{x}^{k+1} - v^k\|^2 \le a_{k+1}\langle\tilde{\nabla}f(x^{k+1}),v^{k+1}\rangle + \frac{\beta_k}{2}\|v^{k+1} - v^k\|^2$, we have

$$
\begin{aligned}
f(y^{k+1}) \le &(1-\tau_k)f(y^k) + \tau_k(f(x^{k+1}) + \langle\tilde{\nabla}f(x^{k+1}), v^{k+1} - x^{k+1}\rangle) \\
&+ \frac{\beta_k}{64A_{k+1}}\|v^{k+1} - v^k\|^2 + \frac{\beta_{k+1} - \beta_k}{16A_{k+1}}\bar{r}_k^2 \\
&+ (1-\tau_k)\langle\tilde{\nabla}f(x^{k+1}) - \nabla f(x^{k+1}), y^k - x^{k+1}\rangle + \frac{\beta_{k+1} - \beta_k}{64A_{k+1}}\|\hat{x}^{k+1} - v^k\|^2 \\
&+ \langle\nabla f(y^{k+1}) - \tilde{\nabla}f(y^{k+1}), y^{k+1} - x^{k+1}\rangle \\
\le &(1-\tau_k)f(y^k) + \tau_k(f(x^{k+1}) + \langle\tilde{\nabla}f(x^{k+1}), v^{k+1} - x^{k+1}\rangle) \\
&+ \frac{\beta_k}{64A_{k+1}}\|v^{k+1} - v^k\|^2 + \frac{\beta_{k+1} - \beta_k}{16A_{k+1}}\bar{r}_k^2 \\
&+ (1-\tau_k)\langle\tilde{\nabla}f(x^{k+1}) - \nabla f(x^{k+1}), y^k - x^{k+1}\rangle + \frac{\beta_{k+1} - \beta_k}{16A_{k+1}}\bar{r}_{k+1}^2 \\
&+ \langle\nabla f(y^{k+1}) - \tilde{\nabla}f(y^{k+1}), y^{k+1} - x^{k+1}\rangle.
\end{aligned}
$$

Here, the last inequality is due to $\|\hat{x}^{k+1} - v^k\|^2 \le (d_{k+1} + r_k)^2 \le 4\bar{r}_{k+1}^2$.

We match the two error terms by $\frac{\beta_{k+1}\bar{r}_k^2 - \beta_k\bar{r}_k^2}{16A_{k+1}} + \frac{\beta_{k+1} - \beta_k}{16A_{k+1}}\bar{r}_{k+1}^2 \le \frac{\beta_{k+1} - \beta_k}{8A_{k+1}}\bar{r}_{k+1}^2$. Multiplying both sides by $A_{k+1}$, we have

$$
\begin{aligned}
A_{k+1}f(y^{k+1}) \le &A_k f(y^k) + a_{k+1}(f(x^{k+1}) + \langle\tilde{\nabla}f(x^{k+1}), v^{k+1} - x^{k+1}\rangle) \\
&+ \frac{\beta_k}{64}\|v^{k+1} - v^k\|^2 + \frac{\beta_{k+1} - \beta_k}{8A_{k+1}}\bar{r}_{k+1}^2 \\
&+ A_k\langle\tilde{\nabla}f(x^{k+1}) - \nabla f(x^{k+1}), y^k - x^{k+1}\rangle \\
&+ A_{k+1}\langle\nabla f(y^{k+1}) - \tilde{\nabla}f(y^{k+1}), y^{k+1} - x^{k+1}\rangle.
\end{aligned}
$$

On the other hand, it holds that

$$
g(y^{k+1}) \le (1-\tau_k)g(y^k) + \tau_k g(v^{k+1}). \tag{61}
$$

Combining (31) and (61), we obtain

$$
\begin{aligned}
A_{k+1}\psi(y^{k+1}) \le &A_k\psi(y^k) + a_{k+1}(f(x^{k+1}) + \langle\tilde{\nabla}f(x^{k+1}), v^{k+1} - x^{k+1}\rangle + g(v^{k+1})) \\
&+ \frac{\beta_k}{2}\|v^{k+1} - v^k\|^2 + \frac{\beta_{k+1} - \beta_k}{8A_{k+1}}\bar{r}_{k+1}^2 \\
&+ A_k\langle\tilde{\nabla}f(x^{k+1}) - \nabla f(x^{k+1}), y^k - x^{k+1}\rangle \\
&+ A_{k+1}\langle\nabla f(y^{k+1}) - \tilde{\nabla}f(y^{k+1}), y^{k+1} - x^{k+1}\rangle.
\end{aligned}
$$

For $\frac{\beta_k}{2}\|v^{k+1} - v^k\|^2$, we use Lemma 4, then

$$
\begin{aligned}
A_{k+1}\psi(y^{k+1}) \leq &A_k\psi(y^k) + a_{k+1}(f(x^{k+1}) + \langle\tilde{\nabla}f(x^{k+1}), v^{k+1} - x^{k+1}\rangle + g(v^{k+1})) \\
&+ \sum_{i=1}^{k} a_i(f(x^i) + \langle\tilde{\nabla}f(x^i), v^{k+1} - x^i\rangle + g(v^{k+1})) + \frac{\beta_k}{2}\|x^0 - v^{k+1}\|^2 \\
&- \sum_{i=1}^{k} a_i(f(x^i) + \langle\tilde{\nabla}f(x^i), v^k - x^i\rangle + g(v^k)) - \frac{\beta_k}{2}\|x^0 - v^k\|^2 \\
&+ \frac{\beta_{k+1} - \beta_k}{8A_{k+1}}\bar{r}_{k+1}^2 + A_k\langle\tilde{\nabla}f(x^{k+1}) - \nabla f(x^{k+1}), y^k - x^{k+1}\rangle \\
&+ A_{k+1}\langle\nabla f(y^{k+1}) - \tilde{\nabla}f(y^{k+1}), y^{k+1} - x^{k+1}\rangle \\
\leq &A_k\psi(y^k) + \sum_{i=1}^{k+1} a_i(f(x^i) + \langle\tilde{\nabla}f(x^i), v^{k+1} - x^i\rangle + g(v^{k+1})) + \frac{\beta_{k+1}}{2}\|x^0 - v^{k+1}\|^2 \\
&- \sum_{i=1}^{k} a_i(f(x^i) + \langle\tilde{\nabla}f(x^i), v^k - x^i\rangle + g(v^k))\frac{\beta_k}{2}\|x^0 - v^k\|^2 \\
&+ \frac{\beta_{k+1} - \beta_k}{8A_{k+1}}\bar{r}_{k+1}^2 + A_k\langle\tilde{\nabla}f(x^{k+1}) - \nabla f(x^{k+1}), y^k - x^{k+1}\rangle \\
&+ A_{k+1}\langle\nabla f(y^{k+1}) - \tilde{\nabla}f(y^{k+1}), y^{k+1} - x^{k+1}\rangle.
\end{aligned}
\tag{62}
$$

We can simplify (33) by using the definition of $\phi_k(\cdot)$:

$$
\begin{aligned}
A_{k+1}\psi(y^{k+1}) \leq &A_k\psi(y^k) + \phi_{k+1}(v^{k+1}) - \phi_k(v^k) \\
&+ \frac{\beta_{k+1} - \beta_k}{8A_{k+1}}\bar{r}_{k+1}^2 + A_k\langle\tilde{\nabla}f(x^{k+1}) - \nabla f(x^{k+1}), y^k - x^{k+1}\rangle \\
&+ A_{k+1}\langle\nabla f(y^{k+1}) - \tilde{\nabla}f(y^{k+1}), y^{k+1} - x^{k+1}\rangle.
\end{aligned}
\tag{63}
$$

Applying the upper inequality recursively, it holds that

$$
\begin{aligned}
A_k\psi(y^k) \leq &\phi_k(v^k) - \phi_0(v^0) \\
&+ \sum_{i=0}^{k-1} \frac{\beta_{i+1} - \beta_i}{8}\bar{r}_{i+1}^2 + A_i\langle\tilde{\nabla}f(x^{i+1}) - \nabla f(x^{i+1}), y^i - x^{i+1}\rangle \\
&+ A_{i+1}\langle\nabla f(y^{i+1}) - \tilde{\nabla}f(y^{i+1}), y^{i+1} - x^{i+1}\rangle \\
\leq &\phi_k(v^k) + \frac{\beta_k}{8}\bar{r}_k^2 - \frac{\beta_0}{8}\bar{r}_1^2 + \sum_{i=0}^{k-1} A_i\langle\tilde{\nabla}f(x^{i+1}) - \nabla f(x^{i+1}), y^i - x^{i+1}\rangle \\
&+ A_{i+1}\langle\nabla f(y^{i+1}) - \tilde{\nabla}f(y^{i+1}), y^{i+1} - x^{i+1}\rangle \\
\leq &\phi_k(v^k) + \frac{\beta_k}{8}\bar{r}_k^2 + \sum_{i=0}^{k-1} A_i\langle\tilde{\nabla}f(x^{i+1}) - \nabla f(x^{i+1}), y^i - x^{i+1}\rangle \\
&+ A_{i+1}\langle\nabla f(y^{i+1}) - \tilde{\nabla}f(y^{i+1}), y^{i+1} - x^{i+1}\rangle,
\end{aligned}
$$

where $\phi_0(v^0) = 0$ and $\beta_0 = 0$.

Since $v^k = \arg\min_x \phi_k(x)$, we apply Lemma 4 again and obtain that:

$$A_k\psi(y^k) \leq \phi_k(v^k) + \frac{\beta_k}{8}\bar{r}_k^2 + \sum_{i=0}^{k-1} A_i\langle\tilde{\nabla}f(x^{i+1}) - \nabla f(x^{i+1}), y^i - x^{i+1}\rangle$$

$$+ A_{i+1}\langle\nabla f(y^{i+1}) - \tilde{\nabla}f(y^{i+1}), y^{i+1} - x^{i+1}\rangle$$

$$= \sum_{i=1}^{k} a_i(f(x^i) + \langle\tilde{\nabla}f(x^i), v^k - x^i\rangle + g(v^k)) + \frac{\beta_k}{2}\|x^0 - v^k\|^2 + \frac{\beta_k}{8}\bar{r}_k^2$$

$$+ \sum_{i=0}^{k-1} A_i\langle\tilde{\nabla}f(x^{i+1}) - \nabla f(x^{i+1}), y^i - x^{i+1}\rangle + A_{i+1}\langle\nabla f(y^{i+1}) - \tilde{\nabla}f(y^{i+1}), y^{i+1} - x^{i+1}\rangle$$

$$\leq \sum_{i=1}^{k} a_i(f(x^i) + \langle\tilde{\nabla}f(x^i), x^* - x^i\rangle + g(x^*)) + \frac{\beta_k}{2}\|x^0 - x^*\|^2 - \frac{\beta_k}{2}\|v^{k+1} - x^*\|^2$$

$$+ \frac{\beta_k}{8}\bar{r}_k^2 + \sum_{i=0}^{k-1} A_i\langle\tilde{\nabla}f(x^{i+1}) - \nabla f(x^{i+1}), y^i - x^{i+1}\rangle$$

$$+ A_{i+1}\langle\nabla f(y^{i+1}) - \tilde{\nabla}f(y^{i+1}), y^{i+1} - x^{i+1}\rangle$$

$$\leq \sum_{i=1}^{k} a_i(f(x^i) + \langle\nabla f(x^i), x^* - x^i\rangle + g(x^*)) + \frac{\beta_k}{2}\|x^0 - x^*\|^2 - \frac{\beta_k}{2}\|v^{k+1} - x^*\|^2$$

$$+ \frac{\beta_k}{8}\bar{r}_k^2 + \sum_{i=0}^{k-1} a_{i+1}\langle\tilde{\nabla}f(x^{i+1}) - \nabla f(x^{i+1}), v^i - x^*\rangle$$

$$+ A_{i+1}\langle\nabla f(y^{i+1}) - \tilde{\nabla}f(y^{i+1}), y^{i+1} - x^{i+1}\rangle$$

$$\leq A_k\psi(x^*) + \frac{\beta_k}{2}\|x^0 - x^*\|^2 - \frac{\beta_k}{2}\|v^{k+1} - x^*\|^2 + \frac{\beta_k}{8}\bar{r}_k^2$$

$$+ \sum_{i=0}^{k-1} a_{i+1}\langle\tilde{\nabla}f(x^{i+1}) - \nabla f(x^{i+1}), v^i - x^*\rangle$$

$$+ A_{i+1}\langle\nabla f(y^{i+1}) - \tilde{\nabla}f(y^{i+1}), y^{i+1} - x^{i+1}\rangle.$$

We use $D_0$ and $D_k$ to replace $\|x^0 - x^*\|$ and $\|v^k - x^*\|$. Since $D_0^2 - D_k^2 \leq 2D_0 r_k \leq 2D\bar{r}_k$ , we have

$$A_k\psi(y^k) \leq A_k\psi(x^*) + \frac{\beta_k}{2}D_0^2 - \frac{\beta_k}{2}D_k^2 + \frac{\beta_k}{8}\bar{r}_k^2$$

$$+ \sum_{i=0}^{k-1} a_{i+1}\langle\tilde{\nabla}f(x^{i+1}) - \nabla f(x^{i+1}), v^i - x^*\rangle$$

$$+ A_{i+1}\langle\nabla f(y^{i+1}) - \tilde{\nabla}f(y^{i+1}), y^{i+1} - x^{i+1}\rangle,$$

which implies

$$\begin{aligned}
\psi(y^k) - \psi(x^*) &\leq \frac{\beta_k(D_0^2 - D_k^2)}{2A_k} + \frac{\beta_k\bar{r}_k^2}{8A_k} + \sum_{i=0}^{k-1} a_{i+1}\langle\tilde{\nabla}f(x^{i+1}) - \nabla f(x^{i+1}), v^i - x^*\rangle \\
&\quad + A_{i+1}\langle\nabla f(y^{i+1}) - \tilde{\nabla}f(y^{i+1}), y^{i+1} - x^{i+1}\rangle \\
&\leq \frac{9\beta_k D\bar{r}_k}{8A_k} + \sum_{i=0}^{k-1} a_{i+1}\langle\tilde{\nabla}f(x^{i+1}) - \nabla f(x^{i+1}), v^i - x^*\rangle \\
&\quad + A_{i+1}\langle\nabla f(y^{i+1}) - \tilde{\nabla}f(y^{i+1}), y^{i+1} - x^{i+1}\rangle.
\end{aligned} \tag{64}$$

We take the expectations of both sides and obtain

$$
\mathbb{E}[\psi(y^k)] - E[\psi(x^*)] \leq \mathbb{E}[\frac{9\beta_k D \bar{r}_k}{8A_k}] + \mathbb{E}[\sum_{i=0}^{k-1} a_{i+1}\langle \tilde{\nabla} f(x^{i+1}) - \nabla f(x^{i+1}), v^i - x^* \rangle]
$$
$$
+ \mathbb{E}[\sum_{i=0}^{k-1} A_{i+1}\langle \nabla f(y^{i+1}) - \tilde{\nabla} f(y^{i+1}), y^{i+1} - x^{i+1} \rangle].
$$

Since $\mathbb{E}[\tilde{\nabla} f(x^{i+1}) - \nabla f(x^{i+1})] = 0$ and $v^i$, $x^*$, $\xi_{i+1}$ are independent, we have

$$
\mathbb{E}\left[\sum_{i=0}^{k-1} a_{i+1}\langle \tilde{\nabla} f(x^{i+1}) - \nabla f(x^{i+1}), v^i - x^* \rangle\right]
$$
$$
= \mathbb{E}_{-\xi_{i+1}^x}\left[\mathbb{E}_{\xi_{i+1}^x}\left[\sum_{i=0}^{k-1} a_{i+1}\langle \tilde{\nabla} f(x^{i+1}) - \nabla f(x^{i+1}), v^i - x^* \rangle\right]\right] = 0.
$$

For the same reason, we have

$$
\mathbb{E}\left[\sum_{i=0}^{k-1} A_{i+1}\langle \nabla f(y^{i+1}) - \tilde{\nabla} f(y^{i+1}), y^{i+1} - x^{i+1} \rangle\right]
$$
$$
= \mathbb{E}_{-\xi_{i+1}^y}\left[\mathbb{E}_{\xi_{i+1}^y}\left[\sum_{i=0}^{k-1} A_{i+1}\langle \nabla f(y^{i+1}) - \tilde{\nabla} f(y^{i+1}), y^{i+1} - x^{i+1} \rangle\right]\right] = 0.
$$

Thus

$$
\mathbb{E}[\psi(y^k)] - \psi(x^*) \leq \frac{9}{8}\mathbb{E}\left[\frac{\beta_k D \bar{r}_k}{A_k}\right].
$$

We will apply Lemma 11 and 3 to obtain the final complexity. Note that $k^* = \underset{0 \leq i \leq k}{\arg\min} \frac{\bar{r}_k}{\sum_{i=0}^{k-1} \bar{r}_i}$. Applying Lemma 3, we obtain that:

$$
\frac{\bar{r}_{k^*}^{\frac{1}{2}}}{\sum_{i=0}^{k^*-1} \bar{r}_i^{\frac{1}{2}}} \leq \frac{(\frac{\bar{r}_k}{\bar{r}_0})^{\frac{1}{2} \times \frac{1}{k}} \log e(\frac{\bar{r}_k}{\bar{r}_0})^{\frac{1}{2}}}{k} \leq \frac{(\frac{4D_0}{\bar{r}})^{\frac{1}{2k}} \log e \frac{4D_0}{\bar{r}}}{2k}. \tag{65}
$$

Thus, for $k^*$, combining the inequality (65) and Lemma 11, it holds that

$$
\mathbb{E}[\psi(y^{k^*})] - \psi(x^*)
$$
$$
\leq \frac{9}{8}\mathbb{E}\left[\frac{\beta_{k^*} D \bar{r}_{k^*}}{A_{k^*}}\right]
$$
$$
\leq \frac{9}{8}\mathbb{E}\left[\beta_k D \left(\frac{\bar{r}_{k^*}^{\frac{1}{2}}}{\sum_{i=0}^{k^*-1} \bar{r}_i^{\frac{1}{2}}}\right)^2\right]
$$
$$
\leq \frac{9}{8}\mathbb{E}[(2^{\frac{9+9\nu}{2}} \tilde{M}_\nu D^{1+\nu} k^{\frac{3-3\nu}{2}} + 2^5 k^{\frac{3}{2}} D\sigma)\left(\frac{(\frac{4D}{\bar{r}})^{\frac{1}{2k}} \log e \frac{4D}{\bar{r}}}{2k}\right)^2] \tag{66}
$$
$$
\leq 36(\frac{4D}{\bar{r}})^{\frac{1}{k}} \log^2 e \frac{4D}{\bar{r}} \frac{\tilde{M}_\nu D^{1+\nu} k^{\frac{3-3\nu}{2}} + k^{\frac{3}{2}} D\sigma}{k^2}
$$
$$
\leq 36(\frac{4D}{\bar{r}})^{\frac{1}{k}} \log^2 e \frac{4D}{\bar{r}} (\frac{\tilde{M}_\nu D^{1+\nu}}{k^{\frac{1+3\nu}{2}}} + \frac{D\sigma}{\sqrt{k}}).
$$

where we use the facts $\{\beta_i\}_{i\in\mathbb{N}}$ is nondecreasing and the random variable $B$ is independent of both $k$ and $D$.

Finally, it remains to use $z^k = y^{k^*}$ by definition. $\qquad\square$

The following lemma is used to show that the balance equation admits a closed-form solution.

**Lemma 12.** *Let $\beta,l,d \geq 0$ and $r > 0$. Then the equation*

$$(\beta_+ - \beta)r = [l - \beta_+ d]_+ \tag{67}$$

*has a unique solution given by*

$$\beta_+ = \beta + \frac{[l - \beta d]_+}{r + d}. \tag{68}$$

This auxiliary result has been used in [[34], Lemma E.1]. We give proof for completeness.

*Proof.* First, the equation has a unique solution since the left-hand side increases from zero to infinity monotonically with respect to $\beta_+$ while the right-hand side decreases from a nonnegative number to zero monotonically with respect to $\beta_+$.

We show that the $\beta_+$ in (68) is the very solution of (67).

When $l - \beta d \leq 0$, $\beta_+ = \beta$. $LHS = (\beta_+ - \beta)r = 0$ and $RHS = [l - \beta_+ d]_+ = [l - \beta d]_+ \leq 0$, which implies $RHS = 0$. Therefore, $LHS = RHS$ and $\beta_+$ is the solution.

When $l - \beta d > 0$, $\beta_+ = \beta + \frac{l - \beta d}{r+d}$. then $LHS = (\beta_+ - \beta)r = (\beta + [\frac{l-\beta d}{r+d}]_+ - \beta)r = r\frac{l-\beta d}{r+d}$ and $RHS = [l - \beta_+ d]_+ = [l - \beta d - \frac{l-\beta d}{r+d}d]_+ = [\frac{r}{r+d}(l - \beta d)]_+ = \frac{r}{r+d}(l - \beta d)$. Therefore, $LHS = RHS$ and $\beta_+$ is the solution as well. $\qquad\square$

# E Two approaches for automatic initialization of parameters in Algorithm 1

## E.1 Automatic initialization of $\beta_0$

In the proof of Proposition 1 and Theorem 2, we assume that $\beta_0 \leq 2^7 I_\nu \widehat{M_\nu} \bar{r}^\nu$. This is reasonable since the upper bound of $\beta_k$ increases polynomial in k, it still holds for enough large $k$. Previous works often ignore the choice of a legal $\beta_0$. Nevertheless, we provide a simple method for choosing an admissible $\beta_0$.

---

**Algorithm 3** $\beta_0$ Initialization Method

---

**Input:** $x^0, \bar{r}$, any other point $x' \in \mathbb{R}^d$ that satisfies $\|x' - x^0\| \leq \bar{r}$ and $f(x') - f(x^0) - \langle \nabla f(x^0), x' - x^0 \rangle > 0$;

**Output:** $\bar{\beta} \leq 2^7 I_\nu \widehat{M_\nu} \bar{r}^\nu$;
1: Set $c = \min\{[f(x') - f(x^0) - \langle \nabla f(x^0), x^0 - x' \rangle]/\bar{r}^2, \frac{1}{2}\}$;
  and $M = 2\frac{f(x') - f(x^0) - \langle \nabla f(x^0), x^0 - x' \rangle - c\bar{r}^2/2}{\|x^0 - x'\|^2}$;
2: Let $\bar{\beta} = \bar{r} \max\{8\sqrt{2M}, 128M\} \min\{1, \sqrt{c}\}$;

---

**Proposition 2.** *Suppose $f(\cdot)$ is locally Hölder smooth and $f(\cdot)$ is not a linear function in $\mathrm{dom}\, g$. If $\bar{r}$ is small enough such that $\bar{r} \leq 4D_0$, then Algorithm 3 can generate a $\beta_0$ that satisfies*

$$\beta_0 \leq 2^7 I_\nu \widehat{M_\nu} \bar{r}^v, \tag{69}$$

*and this method can be implemented in one operation.*

*Proof.* Since

$$M = 2\frac{f(x') - f(x^0) - \langle \nabla f(x^0), x^0 - x' \rangle - c\frac{\bar{r}^2}{2}}{\|x^0 - x'\|^2} \geq 0,$$

we have

$$f(x') = f(x^0) + \langle \nabla f(x^0), x^0 - x' \rangle + \frac{M}{2}\|x^0 - x'\|^2 + c\frac{\bar{r}^2}{2}. \tag{70}$$

The equation (70) is tight and $x^0, x' \in \mathcal{B}_{3D_0}(x^*)$, so that we have

$$M \leq \gamma(\widehat{M_\nu}, c\bar{r}^2) = (\frac{1-\nu}{1+\nu}\frac{1}{c\bar{r}^2})^{\frac{1-\nu}{1+\nu}} \widehat{M_\nu}^{\frac{2}{1+\nu}}. \tag{71}$$

$$c^{\frac{1}{2}}\bar{r}\min\{(128\bar{M})^{\frac{1}{2}}, 128\bar{M}\} \leq c^{\frac{1-\nu}{2}}\bar{r}(128\bar{M})^{\frac{1+\nu}{2}} = 128(\frac{1-\nu}{1+\nu})^{\frac{1-\nu}{2}}\widehat{M_\nu}\bar{r}^v. \tag{72}$$

Thus

$$c^{\frac{1}{2}}\bar{r}\min\{(128\bar{M})^{\frac{1}{2}}, 128\bar{M}\} \leq 2^7 I_\nu \widehat{M_\nu}\bar{r}^v. \tag{73}$$

So we can initialize $\beta_0 = c^{\frac{1}{2}}\bar{r}\min\{(128\bar{M})^{\frac{1}{2}}, 128\bar{M}\}$. $\qquad\square$

### E.2 Automatic initialization of $\bar{r}$

As mentioned in the context, Algorithm 1 requires an input $\bar{r}$ as a guess of $4D_0$ that satisfies $\bar{r} \leq 4D_0$. Setting a small enough $\bar{r}$ to meet $\bar{r} \leq 4D_0$ will incur a multiplicative cost of $(\frac{4D_0}{\bar{r}})^{1+\nu}$ in the convergence rate for other algorithms without distance adaptation. In contrast, we reduce the multiplicative cost to $\log^2(\frac{4D_0}{\bar{r}})$. Moreover, we provide a simple method for obtaining an admissible $\bar{r} \leq 4D_0$ in some special cases.

Proposition 3 can handle the cases where we can choose $x^0$ and make sure $f(\cdot) + g(\cdot)$ is weakly smooth on one of its neighborhoods. Then we can modify the problem by setting $f'(x) = f(x) + g(x)$ and $g'(x) = 0$ to get a legal $\bar{r}$.

---

**Algorithm 4** $\bar{r}$ Initialization Method

---

**Input:** $x^0$ and an initial guess $r$ for $4D_0$
**Output:** $\bar{r} \leq 4D_0$
 1: Initialize $i \leftarrow -1$
 2: **repeat**
 3: $\quad i \leftarrow i + 1$
 4: $\quad d_i \leftarrow 2^{-i}r$
 5: $\quad$ Run Algorithm 1 with parameter $d$ by one iteration and collect the point $v_i^1$ and the coefficient $\beta_{1,i}$
 6: $\quad$ Denote $r_{1,i} = \|v_i^1 - x^0\|$
 7: **until** $v_i^1$ is an interior point in $\operatorname{dom} g$ and $r_{1,i} \geq d_i$
 8: Set $\bar{r} = d_i$

---

**Proposition 3.** *For any $x \in \operatorname{dom} g$, $g(x) = 0$, if there exists $\delta > 0$, $S = \mathcal{B}_\delta(x^0) \subset \operatorname{dom} g$, and let $\nu_S$ be the maximal Hölder exponent of $f(\cdot)$ on $S$ with finite local Hölder continuous constant $\hat{M}_{\nu_S} < +\infty$, $\nu_S > 0$, then Algorithm 4 can generate $\bar{r} \leq 4D_0$, and this method can be implemented in a finite number of iterations.*

*Proof.* Without loss of generality, we assume $\delta \leq 3D_0$, since if $\delta > 3D_0$, we can always take a smaller $\delta' \leq 3D_0$ that still satisfies the condition.

Note that Theorem 1 implies that if $\bar{r}_{1,i} = r_{1,i}$, then $\bar{r} \leq r_{1,i} \leq 4D_0$, and this conclusion is independent of the value of $\beta_1$. So we only need to prove that this method completes in a finite number of operations. Note that $x^1 = \tau_0 v^0 + (1 - \tau_0)y^0 = \tau_0 x^0 + (1 - \tau_0)x^0 = x^0$ does not depend on the value of $\tau_0$. The condition of this method ensures that there exists an interior point in the direction of $-\nabla f(x^0)$.

Applying Lemma 7, it holds that

$$\|v_i' - x^0\| \geq \|v_i^1 - x^0\|,$$

where we define

$$
\begin{aligned}
v_i' &= \arg\min_{x \in \operatorname{dom} g} \left(d_i\langle\nabla f(x^1), x - x^1\rangle + \frac{\beta_0\|x - x^0\|^2}{2}\right) \\
&= \arg\min_{x \in \operatorname{dom} g} \langle\nabla f(x^0), x - x^0\rangle + \frac{\beta_0\|x - x^0\|^2}{2d_i}.
\end{aligned}
\tag{74}
$$

Applying Lemma 7 again with $h_i = \frac{\beta_0}{2d_i}$, we have $\|v_i' - x^0\| \to 0$ as $i \to +\infty$. Therefore, there exists large enough $i^*$ such that $\forall i \geq i^*$, it satisfies that

$$\delta \geq \|v_i' - x^0\| \geq \|v_i^1 - x^0\|, \tag{75}$$

**Case 1:** If this method requires at most $i^*$ operations. Then we prove it directly.

**Case 2:** If this method requires more than $i^*$ operations.

Inequality (75) show that $v_i^1 \in \mathcal{B}_\delta(x^0) \subset \operatorname{dom} g$, which means $v_i^1$ is an interior point. Then, applying the first-order optimality condition to the interior point $v_i^1$, we have:

$$d_i \nabla f(x^0) + \beta_{1,i}(v_i^1 - x^0) = 0$$
$$v_i^1 - x^0 = d_i \frac{\nabla f(x^0)}{\beta_{1,i}}. \tag{76}$$

We can adapt the method 3 to obtain a legal $\beta_0$ to produce $\beta_1 \leq c_\nu d_i^\nu$, which is guaranteed by Lemma 8, where $c_\nu = 2^7 I_\nu \widehat{M}_\nu$.

$$r_{1,i} = \|v_i^1 - x^0\| = d_i \frac{\|\nabla f(x^0)\|}{\beta_1} \geq d_i^{1-\nu} \frac{\|\nabla f(x^0)\|}{c_\nu}. \tag{77}$$

Thus $r_{1,i} = d_i^{1-\nu} \frac{\|\nabla f(x^0)\|}{c_\nu} \geq d_i$ when $2^{-i} r = d_i \leq \left(\frac{\|\nabla f(x^0)\|}{c_\nu}\right)^{\frac{1}{\nu}}, \nu \neq 0$ and this method requires at most $-\frac{1}{\nu} \log \left(\frac{\|\nabla f(x^0)\|}{c_\nu}\right) / \log 2$ loops. $\qquad \square$

# F   More experiment details

In this section, we provide more details about the experiments.

**Softmax problem**   We first reexamine the softmax problem with more parameter settings. Specifically, we set $\mu \in \{0.1, 0.01, 0.001\}$. In all the results, we find that our AGDA consistently performs better than the other compared methods.

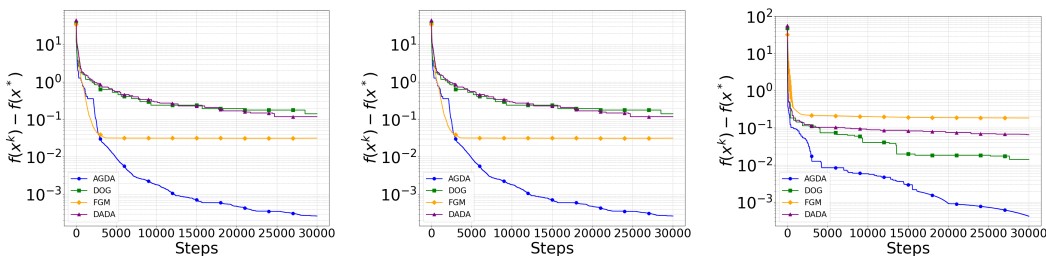

Figure 4: Performance of the compared algorithms on the softmax problem. From left to right: $\mu = 0.1$, $\mu = 0.01$ and $\mu = 0.001$.

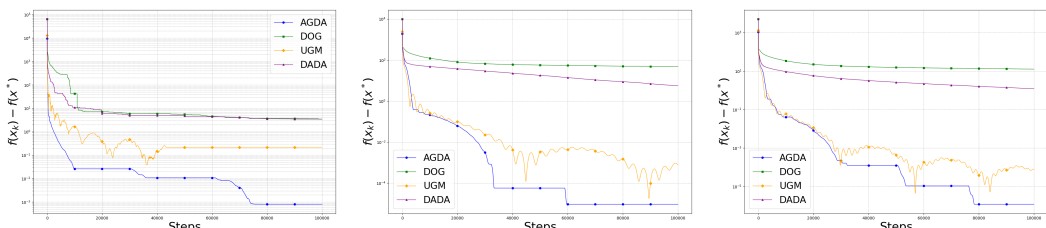

Figure 5: Performance of the compared algorithms on the $L_p$ norm problem. Left: $p = 1$ with `diabetes`. Middle: $p = 1.5$ with `boston`. Right: $p = 2$ with `boston`.

$L_p$ **norm problem** We consider the following problem as an illustrative example, where the smoothness property can be directly adjusted by modifying a parameter $p \in [1, 2]$:

$$\min_{x \in \mathbb{R}^d} f(x) = \|Ax - b\|_p, \tag{78}$$

where $A \in \mathbb{R}^{n \times d}, b \in \mathbb{R}^n$ are taken from real-world datasets in LIBSVM. It is important to note that the smoothness of this problem can be controlled by changing the parameter $p$; as $p$ increases, the degree of smoothness decreases.

We use the same comparison methods as in the softmax problem, adhering to the same parameter settings. In this problem, our algorithm significantly outperforms the other methods, especially when $p$ is small. This suggests that our algorithm is more adaptive in nonsmooth settings and highlights its greater stability.

**Large scale problem** Subsequently, we present the efficacy of our method when applied to large-scale instances. Our main focus is a performance comparison with the UGM algorithm.

### F.1 Line-search bisection method

Here we provide the pseudocode of Line-search bisection method for clarification.

---

**Algorithm 5** Two-Stage Line Search

---

**Input:** $\beta_k$, function $l_k(\cdot)$; tolerance $\epsilon_k^l = \frac{\beta_0}{2k^2}$
**Output:** $\beta_{k+1}$
 1: $i \leftarrow 1$
 2: **while** $l_k(2^{i-1}\beta_k) < 0$ **do**
 3:     $i \leftarrow i + 1$
 4: **end while**                                           $\triangleright$ Stage 1 complete
 5: $i_k' \leftarrow i$
 6: **if** $i_k' = 1$ **then**                                $\triangleright$ Case 1: $l_k(\beta_k) \geq 0$
 7:     $i_k^* \leftarrow 0$
 8:     $\beta_{k+1} \leftarrow \beta_k$
 9: **else**                                      $\triangleright$ Case 2: Need binary search
10:     $a \leftarrow 2^{i_k'-2}\beta_k$
11:     $b \leftarrow 2^{i_k'-1}\beta_k$
12:     **while** $b - a > \epsilon_k^l$ **do**
13:         $m \leftarrow (a + b)/2$
14:         **if** $l_k(m) < 0$ **then**
15:             $a \leftarrow m$
16:         **else**
17:             $b \leftarrow m$
18:         **end if**
19:     **end while**
20:     $\beta_{k+1} \leftarrow b$
21: **end if**

---

