# OpenReview forum: "Accelerated Distance-adaptive Methods for Hölder Smooth and Convex Optimization"
_NeurIPS.cc/2025/Conference — NeurIPS 2025 poster_

### Official Review · Reviewer_7VzW · 2025-07-02

**Clarity:** 4
**Significance:** 3
**Originality:** 3
**Rating:** 4
**Confidence:** 4

**Summary:**

The authors introduce a parameter-free optimization algorithm for convex and Hölder smooth objectives, applicable in both deterministic and stochastic settings. The proposed algorithm removes the need for prior knowledge of key problem parameters such as the distance to the optimal solution or Lipschitz constants. It dynamically adapts its step size to achieve near-optimal convergence rates. Numerical experiments demonstrate robustness to misspecification of initial distance estimates and consistent performance across a wide range of scales.

**Questions:**

How would the algorithm perform if the boundedness assumption (Assumption 1) were relaxed or violated? Could adaptive domain expansion strategies maintain convergence guarantees in unbounded settings?

Can you provide explicit comparisons of computational cost per iteration between your method and DOG/DADAPT?

About the logarithmic factors in your convergence rates: are these fundamental to the distance adaptation approach, or artifacts of the current analysis that could potentially be tightened?

For the stochastic case, how does your balance equation approach compare empirically with variance-reduced methods or adaptive step-size rules like AdaGrad/Adam on unbounded problems?

**Ethical Concerns:**

["NO or VERY MINOR ethics concerns only"]

**Final Justification:**

The authors addressed all the questions that I posed in my review, and I maintain my positive score.

**Limitations:**

The bounded domain assumption for stochastic optimization significantly restricts applicability. Many practical machine learning problems (neural network training, sparse regression) naturally involve unbounded feasible sets.

While claiming to be "parameter-free," the method still requires choosing initial distance estimate $\bar{r}$, and performance depends on this choice being reasonable ($\bar{r}\leq 4D_0$). The line search adds computational overhead per iteration.

The convergence bounds include logarithmic terms whose necessity is not clearly established. Compared to optimal rates, this represents a potentially improvable gap.

Evaluation focuses on convex problems with moderate dimensions. Modern machine learning applications involving high-dimensional, nonconvex, unbounded problems are not addressed. The claimed robustness would be more convincing with large-scale evaluations.

**Paper Formatting Concerns:**

None that caught my attention.

**Quality:**

4

**Strengths And Weaknesses:**

Strengths

1. Quality
The theoretical framework is rigorous and builds upon well-established results in convex optimization. The key innovation lies in combining acceleration with distance adaptation to achieve strong guarntees without specifying the target accuracy in advance.

The extension to stochastic optimization via the balance equation (15) circumvents the line search requirement while maintaining theoretical guarantees, though at the cost of requiring bounded domains.

Empirical evaluation demonstrates practical utility and robustness, especially regarding sensitivity to initialization. However, the experimental scope could be broader in terms of problem scales and domains.

2. Clarity
The paper is well-structured and accessible to readers with background in optimization theory. The motivation behind replacing fixed $\varepsilon$ with the dynamic $\eta_k$ is clearly explained and theoretically justified.

3.  Significance
Removing dependence on unknown problem parameters while maintaining acceleration is an interesting and useful contribution, in particularly in anytime settings, where classical stopping criteria are not easily applied. The stochastic extension fills an important gap.

4. Originality

The authors propose a novel approach to parameter-free optimization by combining acceleration with dynamic distance adaptation. The key insight of using variable target levels instead of a fixed one is new and theoretically sound.

It extends previous work on parameter-free methods (e.g. DOG) in a non-trivial way by incorporating acceleration and Hölder smoothness. The line search procedure and balance equation for stochastic settings represent meaningful technical contributions.


Weaknesses

1. Quality
The assumption of a bounded domain (Assumption 1) in the stochastic case significantly limits applicability. Many machine learning problems are naturally posed on unbounded domains (e.g., unconstrained neural network training). The authors acknowledge this but don't provide clear guidance on relaxation strategies.

The convergence bounds contain logarithmic terms that, while claimed to be "near-optimal," lack clear evidence that tighter analysis is impossible. The dependence on initial distance estimation introduces a practical parameter despite "parameter-free" claims.

The local Hölder smoothness assumption, while more general than global smoothness, still requires the optimization trajectory to remain in the required domain, which may not hold if $\bar{r}$ is poorly chosen initially.

2. Clarity
The notation becomes dense in places (e.g., the interplay between βₖ, r̄ₖ, and Aₖ in Algorithm 1), potentially hindering implementation. The line search termination criteria could be specified more precisely.

The relationship between Algorithms 1 and 2 could be better explained—particularly how the balance equation replaces line search and the computational trade-offs involved.

3. Significance
While theoretically sound, the practical advantages over existing adaptive methods (Adam, AdaGrad variants) in machine learning contexts remain unclear (in what practical scenarios does algorithmic complexity justify the theoretical improvements over simpler adaptive methods?)

The bounded domain requirement for stochastic optimization limits immediate applicability to modern deep learning.

The experimental evaluation, while demonstrating robustness, focuses on relatively small-scale convex problems. Large-scale nonconvex evaluations would strengthen the significance claims.

4. Originality
The technical contributions, while solid, represent incremental advances over DOG/DADAPT rather than fundamental breakthroughs. The core distance adaptation idea predates this work, and the main novelty is in the specific combination with acceleration.

---

> ### Author Rebuttal · Authors · 2025-07-31
>
> Thank you for your insightful comments. Since several questions overlap, we've combined them below to provide a clear and organized response.
>
> ## Weakness
> **Concern about the boundedness assumption in the stochastic case. How does algorithm perform without it? Can you use relax it?**
>
> **A** We agree that the boundedness assumption is a limitation for the stochastic case. We appreciate the opportunity to clarify our reasoning and discuss potential extensions. Our primary goal is to design a fully parameter-free algorithm. For stochastic optimization on an unbounded domain, this ambitious goal is provably impossible. As recently shown by Khaled and Jin [1], any parameter-free method will fail with constant probability for some simple convex problems in this setting, as the iterates' error can diverge. The bounded domain assumption is therefore a necessary precondition for the specific theoretical guarantees we aim to provide. While theoretical result seems unpromising, we did some preliminary experiments without the bounded domain assumption.
>
> **Relaxing boundedness** That being said, we believe this assumption can be relaxed if one is willing to move away from a strictly parameter-free design. A promising direction, inspired by [2], involves using a carefully controlled dynamic step size to ensure the iterates remain bounded with high probability. However, this approach would likely add a logarithmic factor (e.g., $O(\log(T))$) to the convergence rate and hence is suboptimal. Moreover, the proof would require a substantially different analysis based on concentration arguments, which we leave for future work.
>
> **The convergence bounds contain logarithmic terms lack clear evidence that tighter analysis is impossible.**
>
> **A** Thank you for your valuable comment. Compare with the optimally tuned SGD, the convergence rates of AGDA contain a log term $\log(\frac{D_0}{\bar{r}})$. In the distance adapation, this term is generated by the AM-GM inequality, and it is tight in our method.
> To our knowledge, it is unclear at present whether such a logarithmic term is tight or not. We acknowledge this as an interesting open question and will consider investigating it in our future work.
>
> **The dependence on initial distance estimation introduces a practical parameter despite “parameter-free” claims; Trajector may not in the domain if $\bar{r}$ is poorly chosen initially**
>
> **A** The reviewer is correct that our algorithm requires an initial distance estimate $\bar{r}$. In this line of research, the term "parameter-free" or "tuning-free" does not mean an absence of all user inputs. Rather, it refers to algorithms that achieve near-optimal convergence rates (often within logarithmic factors of the best possible rate) with almost no knowledge of problem-dependent constants. The need for an initial, and potentially very loose, estimate for the distance is consistent with the established definition of "parameter-free" used across the recent literature [1, 2, 3]. In practice, our experiments (Fig. 1) demonstrate that the algorithm's performance is quite robust to the choice of this initial estimate.
> From a theoretical standpoint, we discuss in the appendix how a suitable $\bar{r}$ can be determined in a preprocessing step, ensuring this parameter is not a barrier to using the method.
>
> **The local Hölder smoothness requires the trajectory to remain in the required domain, which may not hold if $\bar{r}$ is poorly chosen initially.**
>
> **ANS:** This is a good question. If $\bar{r} > 4D_0$, our analysis shows that the algorithm doesn't fail, but its performance degrades gracefully. Specifically, if $\bar{r}$ is chosen much larger than $D_0$, we can modify the analysis to show that the sequence $\{x^k, y^k, v^k\}\_{k=0,1,\ldots,K}$ remain within an enlarged ball $\mathcal{B}_{D_0+\bar{r}/2}(x^*)$, hence we can still ensure the local Hölder smoothness.
>
> The consequence of a poorly chosen $\bar{r}$ is that the rate is $\mathcal{O}\left(\left(\frac{\bar{r}}{D_0}\right)^{\nu}\frac{D_0^{1+\nu}}{K^{\frac{1+3\nu}{2}}}\right),$ as opposed to $\mathcal{O}\left(\log^2\left(e\frac{D_0}{\bar{r}}\right)\frac{D_0^{1+\nu}}{K^{\frac{1+3\nu}{2}}}\right).$ Thus, while a good initial estimate is needed for optimal performance, the algorithm remains robust and convergent even with a poor one. We will add a remark to the paper to clarify this important trade-off.
>
> **Clarity: The notation is dense, potentially hindering implementation; The line search termination criteria could be specified more precisely.**
>
> **ANS:** Thank you for your feedback. We acknowledge that the notation such as for $\beta_k$, $\bar{r}_k$, and $A_k$.  In the next revision, we will clarify the relationships between these parameters and provide additional explanations.
>
> Thank you for pointing this out. For the function $l_k(\beta)$, we first locate an interval containing the root by repeatedly doubling the search range. We then perform a binary search within this interval until its length is less than a given tolerance. We will include pseudocode in the Appendix to specify the termination criteria more precisely.
>
> **The relationship between AGDA and its LSFM could be clarified by explaining how the balance equation replaces line search, particularly with regard to the computational trade-offs involved.**
>
> **ANS:** Thank you for your suggestion. The line search uses the current smoothness information to update the parameters at each iteration, whereas the balance equation only relies on information from previous iterates. As a result, the balance equation may not accurately estimate the smoothness at the current point. To mitigate the risk of significant inaccuracies, it is necessary to assume that the domain is bounded.
>
> **Large-scale nonconvex evaluations would strengthen the significance claims. Advantage over existing adaptive methods (AdamW, AdaGrad) unclear**
>
> **ANS** Our paper's primary focus is on establishing a rigorous theoretical foundation for a new class of distance-adaptive, parameter-free optimizers. The current experiments are therefore designed to validate these theoretical contributions in controlled settings. We agree that applying these ideas to large-scale non-convex problems is a crucial next step. However, this often requires significant, non-trivial engineering and problem-specific adjustments (e.g., layer-wise DOG learning rate [2]), which represents a substantial project beyond the scope of this theoretical work.
>
> Regarding the comparison with methods like Adam, we believe our work addresses a **complementary, rather than competing**, challenge. Adaptive methods like Adam excel at learning an effective preconditioning metric, while we focus on learning a robust step size schedule adapt to function's unknown smoothness. The success of Adam follow-up work like Muon, Shampoo suggests, that for many deep learning tasks, choosing the right metric is currently more impactful than adapting the step size alone. We hypothesize that the most powerful approach would likely involve integrating both ideas. It would be interesting to integrate our theoretically-grounded adaptive step size schedule with the powerful metric adaptation of methods like Adam.
>
> **The core idea represents no fundamental breakthroughs, and the main novelty is in the specific combination with acceleration.**
>
> **ANS** We thank the reviewer for acknowledging our work's technical solidity. While our approach builds upon the important foundations of distance adaptation and Nesterov's acceleration, we respectfully argue that their synthesis is a significant and non-trivial step, as a naive combination fails.
>
> Our core contribution lies in the new analytical techniques we developed to make this combination successful. Specifically, our analysis introduces a delicate inductive argument to handle acceleration under the practical assumption of only local Hölder smoothness—which requires simultaneously proving iterate boundedness—and eliminates the domain diameter D as a parameter in the stochastic setting. We believe these technical advances are a key contribution and will revise our introduction to better highlight them.
>
> ## Questions:
>
> **Are there specific comparisons of computational cost per iteration between AGDA and DOG/DADAPT?**
>
> **ANS:** The main computational cost of the algorithm comes from computing gradients. Therefore, we evaluate the computational cost by measuring the average number of gradient oracle calls per iteration, as shown in the table below:
>
> | |AGDA|FGM|
> |---|---|---|
> |matrix game|3.06|4.18|
> |softmax|2.7941|3.0024|
>
>
> **For the stochastic case, how does your balance equation approach compare empirically with variance-reduced methods or adaptive step-size rules on unbounded problems?**
>
> **ANS:** In the experiments presented above, we compare AGDA with AdamW. Due to time constraints, we have not yet included comparisons with other adaptive step-size methods or variance-reduced algorithms on large-scale models. We plan to conduct more comprehensive empirical comparisons with a wider range of adaptive and variance-reduced methods on unbounded problems in the future.
>
> **The line search adds computational overhead per iteration.**
>
> **ANS:** Yes, the reason why AGDA only achieves a nearly optimal convergence rate with respect to $K$ is the increasing cost of line search compared to FGM. Up to the $K$-th iteration, AGDA requires about $\mathcal{O}(K\log K)$ calls of oracle, while FGM only need about $\mathcal{O}(K)$ calls. We think this issue could be addressed in future work.
>
> [1] Khaled A, Jin C. Tuning-free stochastic optimization. arXiv:2402.07793, 2024.
> [2] Ivgi, M., Hinder, O. and Carmon, Y., 2023. DoG is SGD’s best friend: A parameter-free dynamic step size schedule. In ICML
> [3] Attia, A. and Koren, T., 2024. How free is parameter-free stochastic optimization?. arXiv:2402.03126
> [4] Bernstein, J. and Newhouse, L., 2024. Old optimizer, new norm: An anthology. arXiv:2409.20325.

---

> > ### Comment · Reviewer_7VzW · 2025-08-05
> >
> > Thank you for your rebuttal. All the questions that I asked in my review have been answered.

---

> > > ### Author Response · Authors · 2025-08-06
> > >
> > > Thank you again for your thoughtful comments and for confirming that our initial rebuttal answered your questions.
> > >
> > >
> > > We wanted to briefly follow up to let you know that we have just added a new set of experiments, which directly address your initial concerns about non-convex and unbounded settings. Due to the short time window, this empirical study required a substantial effort that we've only just been able to complete, and we have added it to our response to Reviewer 4xjD.
> > >
> > > We hope you might have a moment to consider these new results, as we believe they provide stronger evidence for the practical significance of our work.
> > >
> > > Thank you again for your valuable time and consideration.

---

### Official Review · Reviewer_7P9W · 2025-07-02

**Clarity:** 3
**Significance:** 2
**Originality:** 2
**Rating:** 4
**Confidence:** 3

**Summary:**

The paper presents an accelerated gradient method for Hölder-smooth functions that does not require any difficult input parameters, such as the Hölder smoothness constant or the distance to a solution. The algorithm has an almost optimal (up to log factors) any-time convergence rate. The stochastic version of the algorithm relies on the assumption of a bounded domain instead of a line search.

The paper is well written, with clear explanations of how the proof works. However, I did not check the proofs in the appendix.

**Questions:**

### Minor questions/typos

- This paper https://arxiv.org/abs/2402.06271 may be relevant.

- l.16: "where" is missing

- l.104: Why do we introduce the Bregman distance at all?

- l.166: I think error bounds in optimization community have a bit different meaning. To me that condition looks as the "Hölder" descent lemma.

- l. 143: "that" is missing

- l. 159 "smooth smoothness"

- l. 254: "t" is missing

**Ethical Concerns:**

["NO or VERY MINOR ethics concerns only"]

**Final Justification:**

I will keep my score as it is (which is rather positive), as I still find the paper's motivation a bit too narrow for my taste. However, this is, of course, quite subjective.

**Limitations:**

yes

**Paper Formatting Concerns:**

^_^

**Quality:**

3

**Strengths And Weaknesses:**

My main concern is that I don't know which practical functions this method is applicable to. It seems that the authors don't know either. Softmax and least-squares are smooth problems. Matrix games can be solved using primal-dual methods in time complexity of $O(1/T)$ instead of $O(1/\sqrt T)$.

### Experiments.

- **Softmax.** Did authors mean that we first fix $x=0$ and used it to define $a_i$? Otherwise, the function $f$ will become quite non-convex and I don't see why 0 will be its solution.

- **Matrix game.** I find it a bit suspicious.  For the matrix game problem we can estimate easily
all constants we want. Also acceleration is this case ceases to exist. Even good
subgradient algorithm should work equally well on this problem. So to me it
looks as either intentional not-optimal implementation of other algorithms or
some cherry-picking. Or then there must some clear reasons of such start
contrast in the performance.

---

> ### Author Rebuttal · Authors · 2025-07-31
>
> ## Weakness
> **My main concern is that I don't know which practical functions this method is applicable to.**
>
> **ANS:** The primary strength of our algorithm lies in its ability to automatically adapt to the effective smoothness of a problem, which is valuable in scenarios where the optimal smoothness parameters for an optimizer are unknown or difficult to determine.
> This is particularly relevant in several practical settings: 1) Black-Box Optimization where the function's analytical form is hidden; 2) ill-Conditioned Smooth Problems, which may be theoretically smooth, but can behave like a non-smooth function from an optimization perspective; 3) locally Smooth Problems (e.g. piece-wise smooth function): A globally non-smooth function may possess some local smooth structure that our algorithm can exploit to accelerate convergence.
>
> Our experiment with the softmax function provides a concrete example. As shown in Figure 1 (left), when the problem is made ill-conditioned ($\mu=0.0005$), our adaptive algorithm significantly outperforms specialized methods, including standard non-smooth (DOG), non-accelerated smooth (DADA), and even accelerated smooth (FGM) algorithms. This result highlights our method's ability to find a more effective optimization path by adapting to the problem's structure.
>
>
> **Did authors mean that we first fix $x = 0$ and used it to define $a_i$? Otherwise, the function $f$ will become quite non-convex and I don't see why 0 will be its solution.**
>
> **ANS:** Yes, the correct way to define $a_{i}$ is $a_{i} = \hat{a}_{i} - \nabla \hat{f}(0)$. There is indeed a typo in the paper regarding this definition. Thank you for pointing it out; we have fixed this issue.
>
> **Matrix games can be solved using primal-dual methods in $O(1/T)$ instead of $O(1/\sqrt{T})$. Result looks a bit suspicious. Subgradient method should work equally well**
>
> **ANS:**
> The reviewer is correct that specialized primal-dual methods can solve matrix games with a faster rate of $O(1/T)$. Our goal with this experiment was not to claim state-of-the-art performance for this specific problem class, but rather to demonstrate the generality and robustness of our parameter-free framework on a standard non-smooth problem, a testbed also used in [1].
>
> Regarding the comparison to the subgradient method, our algorithm's superior performance stems from its ability to discover and exploit the problem's "hidden smoothness." While the matrix game problem is non-smooth in the worst case, our adaptive method (AGDA) identifies a much smaller local smoothness constant. This allows it to take more effective steps and converge faster. A standard subgradient method cannot adapt to this structure and thus performs more poorly. This finding is consistent with the results in [1], where Nesterov also reported that a subgradient-based method (dual averaging) was outperformed by his FGM on this same problem. The table below provides direct evidence for this mechanism, showing that the effective Lipschitz parameter computed by AGDA is orders of magnitude smaller than that found by the FGM.
>
> | Steps      | 200        | 400        | 600        | 800        | 1000       | 1200       | 1400       | 1600       |
> |:---------:|:----------:|:----------:|:----------:|:----------:|:----------:|:----------:|:----------:|:----------:|
> | **FGM**   | 14459.5507 | 43499.1062 | 211329.6208| 236944.2313| 262557.3429| 288169.2348| 620206.7242| 656420.8525|
> | **AGDA**  | 35.8016    | 216.1367   | 474.8734   | 798.8322   | 1213.5683  | 1668.9429  | 1913.2760  | 1837.6467  |
>
> (Table: The computed Lipschitz parameters in FGM and AGDA. The results of FGM are nearly the same as those reported by Nesterov's paper.)
>
>
>
>
> ## Minor questions
>
> **This paper [2] may be relevants**
>
> **ANS**: Thank you for bringing [2] to our attention. First, this paper considers the same setting as ours and proposes a novel step size adaptation algorithm. The main difference between our method and this paper is that we use acceleration, which allows us to handle the general case, and we also provide an analysis for the stochastic setting. We will cite [2] in the next revision.
>
> **Why do we introduce the Bregman distance at all?**
>
> **Ans**: We apologize for the confusion. We will remove the introduction of the general Bregman distance in the next revision.
>
> **There are some typos in the paper.**
>
> **ANS:** Thank you! We have fixed these typos.
>
> [1] Nesterov Y. Universal gradient methods for convex optimization problems[J]. Mathematical Programming, 2015, 152(1): 381-404.
>
> [2] Oikonomidis K A, Laude E, Latafat P, et al. Adaptive proximal gradient methods are universal without approximation[J]. arXiv preprint arXiv:2402.06271, 2024.

---

> > ### Comment · Reviewer_7P9W · 2025-08-04
> > **Response**
> >
> > Thanks for the response!
> >
> > I have no further comments. I will keep my score as it is (which is rather positive), as I still find the paper's motivation a bit too narrow for my taste. However, this is, of course, quite subjective.

---

> > > ### Author Response · Authors · 2025-08-06
> > >
> > > Thank you again for your feedback and for confirming that our rebuttal addressed your questions!

---

### Official Review · Reviewer_4xjD · 2025-07-03

**Clarity:** 2
**Significance:** 3
**Originality:** 3
**Rating:** 5
**Confidence:** 3

**Summary:**

This paper propose a novel parameter-free first-order method (called AGDA) for solving Hölder smooth composite convex optimization problems. It is proven that AGDA achieves a near-optimal convergence rate, up to logarithmic factors, without the need to specify the target accuracy in advance. Additionally, the paper eliminates the necessity of line-search procedures in AGDA and introduces AGDA-LSFM. Under bounded domain assumption, a convergence result for AGDA-LSFM is established. Preliminary experiments demonstrate the effectiveness of both AGDA and AGDA-LSFM.

**Questions:**

Q1. Why introduce the general Bregman distance at the beginning of Section 2 but not use it later?

Q2. Why is $\eta_k$ chosen in the special form, differing from the earlier approach? What is the underlying insight?

Q3. What is the price of the automatic initialization strategy for $\bar{r}$ discussed in Remark 3 and Appendix?

Q4. In Fig. 1 (Middle and Right), why does FGM with $\epsilon=1$ perform better than FGM with $\epsilon=0.01$?

Q5. Why are standard statistical results not reported for the experiment in the stochastic setting?

Typos:

- Line 16: if $g$ is real-valued, then its domain is the whole space.

- Eq. (8): $k$ should start from $0$.

- Should $\bar{r}$ be introduced after Eq. (4)?

- In Algorithm 1, should $z^0$ be $v^0$?

- Line 8 in Algorithm 1 and Line 147: There are two different sums in the definition of $v^{k+1}$ ? By the way, $v_{k+1}$ should be $v^{k+1}$.

- Proposition 2: It seems that $D_k$ has no definition.

- Remark 2: A reference is missing for the lower bound.

- Line 4 of Algorithm 2: $\nabla f$ should be a stochastic gradient.

- What is $f$ for the matrix game?

**Ethical Concerns:**

["NO or VERY MINOR ethics concerns only"]

**Final Justification:**

The rebuttal addressed all of my concerns thoroughly.

**Limitations:**

Yes

**Quality:**

3

**Strengths And Weaknesses:**

**Strengths:**

1）The proposed AGDA is novel, combining Nesterov’s dual averaging technique with a distance-adaptive approach.

2）AGDA is shown to achieve a near-optimal convergence rate for solving Hölder-smooth composite convex optimization problems without requiring prior knowledge of smoothness parameters or explicit parameter tuning.

3）The effectiveness of AGDA and AGDA-LSFM is validated through numerical experiments.

**Weaknesses:**

1）The line search procedure used in AGDA introduces additional computational overhead per iteration, which may reduce the algorithm’s overall efficiency.

2）In Line 56, the paper claims, ``In addition, we propose a line-search-free accelerated method that achieves optimal convergence rates ... " but it lacks clarity following Theorem 3.

3）The paper does not sufficiently discuss the proposed methods in comparison to the restarted variant of SGD, such as the recent results in Attia, A. and Koren, T. (How Free is Parameter-Free Stochastic Optimization?) and Khaled, A. and Jin, C. (Tuning-Free Stochastic Optimization).

4）Although the numerical results are interesting, the paper lacks further experimental results on large-scale model training, which would provide valuable insights.

---

> ### Author Rebuttal · Authors · 2025-07-31
>
> ## Weakness
> **1The line search procedure introduces additional computational overhead per iteration, which may reduce the algorithm’s overall efficiency.**
>
> **Ans**: Yes, the reason why AGDA only achieves a nearly optimal convergence rate with respect to $K$ is the increasing cost of line search compared to FGM. Up to the $K$-th iteration, AGDA requires about $\mathcal{O}(K\log K)$ calls of oracle, while FGM only need about $\mathcal{O}(K)$ calls. We think this issue could be addressed in future work.
>
>
> **2 In Line 56, what the paper claimed lacks clarity following Theorem 3.**
>
> **Ans**: Thank you for pointing this out. Theorem 3 establishes the optimal convergence rate with respect to $K$ for Hölder-smooth and stochastic problems. However, in our analysis, the bound on $D_0$ is relaxed to $D$, which may become large if $D_0$ is not on the same order as $D$. We will clarify this point in the next revision.
>
> **3 The paper does not sufficiently discuss the proposed methods in comparison to the restarted variant of SGD, such as the recent results in [1] and [2].**
>
> **Ans**:  Thank you for pointing out this omission. We acknowledge that our paper does not sufficiently compare our proposed methods with restarted variants of SGD, as discussed in [1] and [2].
> [1] investigates parameter-free methods and notes that most existing approaches still require certain problem-specific information, such as the Lipschitz constant and $D_0$. In contrast, our algorithms only require an estimate of a lower bound for $D_0$, and this incurs only a minor additional cost. [2] further shows that it is impossible to develop a truly tuning-free algorithm for smooth or nonsmooth stochastic convex optimization when the domain is unbounded. We will include a more thorough discussion of these works in the literature review section of the next revision.
>
>
> **4 The paper lacks further experimental results on large-scale model training, which would provide valuable insights.**
> **Ans**: See our response to 7VzW.
>
> ## Question
>
> **1 Why introduce the general Bregman distance at the beginning of Section 2 but not use it later?**
>
> **Ans**: We apologize for the confusion. We will remove the introduction of the general Bregman distance in the next revision.
>
>
> **2 What is the underlying insight of choosing $\eta_k$ in the special form, differing from the earlier approach?**
>
> **Ans**: FGM, as proposed by Nesterov, achieves a convergence rate of $\mathcal{O}(D_0^\nu M_\nu / k^{\frac{1+3\nu}{2}} + \delta/2)$, where $\delta$ is set as the target accuracy $\epsilon$. Under this setting, FGM cannot guarantee convergence to the optimal solution as $k$ approaches infinity. Our key insight is that by adaptively setting $\delta$ to $\mathcal{O}(D_0^\nu M_\nu / k^{\frac{1+3\nu}{2}})$, it is possible to balance the two terms in the convergence rate expression.
>
>
> **3 What is the price of the automatic initialization strategy for $\bar{r}$ discussed in Remark 3 and Appendix?**
>
> **Ans**: The automatic initialization strategy for $\bar{r}$ provide a legal $\bar{r}$ satisfied $\bar{r}\leq4D_0$ in some special cases. At the end of the detailed proof of this strategy, we claim that we need implement the first step of AGDA at most $-\frac{1}{\nu}\log\left(\frac{\|\nabla f(x^0)\|}{c_\nu }\right)/\log2$ times. Moreover, the first step of AGDA will call at most a constant number of oracle.
>
> **4 In Fig. 1 (Middle and Right), why does FGM with $\epsilon = 1$ perform better than FGM with $\epsilon = 0.01$?**
>
> **Ans**: $\epsilon = 0.01$ corresponds to a large Lipschitz smooth constant for FGM. Consequently, FGM struggles to make progress despite its acceleration. This is precisely the issue our proposed method, AGDA, is designed to overcome. By adaptively tuning $\delta$, AGDA maintains a well-conditioned trajectory, enjoying the benefits of a small effective error tolerance without suffering from the associated ill-conditioning. This experiment effectively demonstrates the practical advantage of our adaptive mechanism over the fixed-parameter approach of the universal FGM.
>
> **5 Why are standard statistical results not reported for the experiment in the stochastic setting?**
>
> **Ans**: We apologize for not reporting the standard statistical results. We provide some statistical results below and will include them in the next revision.
>
> AGDA-LSFM:
>
> |Epoch|Mean|Std|Var|95%-CI|
> |---|---|---|---|---|
> | 100   | 4.5176e+03 | 1.3818e+02| 1.9093e+04  | (4.4187e+03, 4.6164e+03)  |
> | 200   | 3.1232e+03 | 1.6090e+02| 2.5889e+04  | (3.0081e+03, 3.2383e+03)  |
> | 300   | 2.1278e+03 | 1.8061e+02| 3.2620e+04  | (1.9986e+03, 2.2570e+03)  |
> | 400   | 1.3908e+03 | 5.9527e+01| 3.5435e+03  | (1.3482e+03, 1.4334e+03)  |
> | 500   | 9.6490e+02 | 1.0554e+02| 1.1139e+04  | (8.8940e+02, 1.0404e+03)  |
> | 600   | 6.0236e+02 | 1.5445e+01| 2.3854e+02  | (5.9131e+02, 6.1341e+02)  |
> | 700      | 3.9653e+02 | 2.7015e+01| 7.2980e+02  | (3.7721e+02, 4.1586e+02)  |
>
> Adam:
>
> |Epoch|Mean|Std|Var|95%-CI|
> |---|---|---|---|---|
> | 100   | 6.0804e+03 | 6.1950e+01| 3.8378e+03  | (6.0361e+03, 6.1247e+03)  |
> | 200   | 4.2011e+03 | 7.8974e+01| 6.2370e+03  | (4.1446e+03, 4.2576e+03)  |
> | 300   | 3.1595e+03 | 4.8552e+01| 2.3573e+03  | (3.1247e+03, 3.1942e+03)  |
> | 400   | 2.5412e+03 | 1.8405e+02| 3.3874e+04  | (2.4096e+03, 2.6729e+03)  |
> | 500   | 1.9329e+03 | 1.2730e+02| 1.6206e+04  | (1.8418e+03, 2.0240e+03)  |
> | 600   | 1.4822e+03 | 4.7201e+01| 2.2279e+03  | (1.4484e+03, 1.5160e+03)  |
> | 700   | 1.1632e+03 | 9.7067e+01| 9.4220e+03  | (1.0938e+03, 1.2326e+03)  |
>
> USFGM:
>
> |Epoch|Mean|Std|Var|95%-CI|
> |---|---|---|---|---|
> | 100   | 1.1473e+04 | 7.6193e+02| 5.8054e+05  | (1.0928e+04, 1.2018e+04)  |
> | 200   | 1.0233e+04 | 3.0175e+02| 9.1050e+04  | (1.0017e+04, 1.0449e+04)  |
> | 300   | 9.3534e+03 | 1.8890e+02| 3.5684e+04  | (9.2182e+03, 9.4885e+03)  |
> | 400   | 8.7421e+03 | 1.6653e+02| 2.7732e+04  | (8.6229e+03, 8.8612e+03)  |
> | 500   | 8.2098e+03 | 1.6352e+02| 2.6739e+04  | (8.0928e+03, 8.3267e+03)  |
> | 600   | 7.7925e+03 | 1.5409e+02| 2.3745e+04  | (7.6823e+03, 7.9027e+03)  |
> | 700   | 7.4379e+03 | 1.5542e+02| 2.4156e+04  | (7.3267e+03, 7.5490e+03)  |
>
> As shown in the table, our method demonstrates faster convergence and greater stability.
>
>
> **There are some typos in the paper.
> Typos:**
> 1) Line 16: If $g$ is real-valued, then its domain is the whole space.
> 2) Eq.(8): $k$ should start from 0.
> 3) Should $\bar{r}$ be introduced after Eq. (4)?
> 4) In Algorithm 1, should $z^0$ be $v^0$?
> 5) Line 8 in Algorithm 1 and Line 147: There are two different sums in the definition of $v^{k+1}$? By the way, $v_{k+1}$ should be $v^{k+1}$.
> 6) Proposition 2: It seems that $D_k$ has no definition.
> 7) Remark 2: A reference is missing for the lower bound.
> 8) Line 4 of Algorithm~2: $\nabla f$ should be a stochastic gradient.
> 9) What is $f$ for the matrix game?**
>
> **Ans**:
> 1) Yes.
> 2) Yes.
> 3) Yes, we will do it.
> 4) Yes.
> 5) The definition of $v^{k+1}$ in Algorithm 1 is correct and yes, $v_{k+1}$ should be $v^{k+1}$.
> 6) Yes, $D_k:=\|v^k-x^*\|$. We will add the defination before Proposition 2.
> 7) We will add the reference in Remark 2.
> 8) Yes, the notation should be $\tilde{\nabla} f$ instead of $\nabla f$.
> 9) For the matrix game, we consider $\min_{x \in \Delta_n} \max_{y \in \Delta_m}\langle x, A y \rangle = \min_{x \in \Delta_n}  \psi_p(x) = \max_{y \in \Delta_m}  \psi_d(y),$ where $\psi_p(x)\stackrel{\text{def}}{=} \max_{1 \leq j \leq m} \langle x, A e_j \rangle $ and $ \psi_d(y) \stackrel{\text{def}}{=} \max_{1 \leq j \leq m} \langle x, A e_j \rangle$
>
> Then, this problem can be posed as a minimization problem:
> $$
> \min_{x \in \Delta_n, y \in \Delta_m} \{ \psi_{pd}(x,y) = \psi_p(x) - \psi_d(y) \}.
> $$
>
> And the optimal value of this problem is zero.
>
> We appreciate your feedback and have fixed these typos.
>
> [1] Attia A, Koren T. How free is parameter-free stochastic optimization?[J]. arXiv preprint arXiv:2402.03126, 2024.
> [2] Khaled A, Jin C. Tuning-free stochastic optimization[J]. arXiv preprint arXiv:2402.07793, 2024.
> [3] Dahl G E, Schneider F, Nado Z, et al. Benchmarking neural network training algorithms[J]. arXiv preprint arXiv:2306.07179, 2023.

---

> > ### Comment · Reviewer_4xjD · 2025-08-04
> >
> > Thanks for the rebuttal and for addressing all raised points.
> >
> > > 4 The paper lacks further experimental results on large-scale model training, which would provide valuable insights. Ans: See our response to 7VzW.
> >
> > Are you referring to the experimental results mentioned in the response to Reviewer 7P9W?

---

> > > ### Author Response · Authors · 2025-08-06
> > >
> > > Thank you for the follow-up message. We apologize for the confusion. We intended to direct you to Reviewer 7VzW’s final comment in the *limitation* section, where we were asked for evaluation on modern machine learning applications involving high-dimensional,nonconvex,unbounded problems.
> > >
> > > Your request and Reviewer 7VzW’s are closely aligned, as both call for large-scale experiments to demonstrate practical relevance. We agree with this assessment and, in response, have added larger experiments than those in the original submission. While our main focus is still on providing new theoretical analysis, we hope these new experiments (including nonconvex problems) can provide the valuable insights, as you suggested.
> > >
> > > ### 1.  Convex problems
> > > For convex problems, with consider the Lp norm SVM:
> > > $$\min_{x\in\mathbb{R}^d}\frac{1}{n}\sum_{i=1}^{n}\frac{1}{p}[1-b_i\langle a_i,x \rangle]_+^{p}+\lambda \|\|x\|\|_1.$$
> > > We conducted a comparison experiment between FGM and AGDA on the different dataset with different parameter $p$.
> > >
> > >
> > > First, we use the HIGGS dataset, which is a binary classification benchmark from LibSVM. Due to time constraint, we only use 10% data for the experiments. The results are shown in the following tables.（Set p=1.5 and $\lambda$=0.001.）
> > >
> > > #### Table 1: Performance on HIGGS Dataset
> > >
> > > **AGDA:**
> > > |Epoch|1|20|40|60|80|100|120|140|160|180|
> > > |---|---|---|---|---|---|---|---|---|---|---|
> > > |Loss|0.66585279|0.58963430|0.58821201|0.58815455|0.58814651| 0.58814526| 0.58814484| 0.58814484| 0.58814490| 0.58814484|
> > > | Train accuracy| 51.1278% | 63.0706% | 63.9517% | 63.7186% | 63.8078% | 63.7656% | 63.7808% | 63.7820% | 63.7778% | 63.7836% |
> > > | Test accuracy| 51.2135% | 63.2115% | 63.9785% | 63.8035% | 63.8800% | 63.8545% | 63.8620% | 63.8600% | 63.8605% | 63.8585% |
> > >
> > > **FGM:**
> > > |Epoch|1|20|40|60|80|100|120|140|160|180|
> > > |---|---|---|---|---|---|---|---|---|---|---|
> > > |Loss|0.64150435|0.58818716|0.58815223|0.58817464|0.58815044| 0.58819669|0.58814931|0.58815795|0.58822095|0.58814859  |
> > > | Train accuracy|58.9240% | 63.8908% | 63.7422% | 63.7963% | 63.7750% | 63.8015% | 63.7803% | 63.7863% | 63.7966% | 63.7811% |
> > > | Test accuracy| 58.8495% | 63.9325% | 63.8325% | 63.8680% | 63.8510% | 63.8480% | 63.8490% | 63.8495% | 63.8425% | 63.8485% |
> > >
> > > Furthermore, another one is on the News20.binary dataset. The results are as follows:(Set p=1 and $\lambda$ = 0.001; The values in each row represent the gap between the current loss and the optimal loss.)
> > >
> > > #### Table 2: Performance on News20.binary
> > > |Iteration|500|1000|1500|2000|2500|3000|3500|4000|4500|5000|
> > > |---|---|---|---|---|---|---|---|---|---|---|
> > > |AGDA|2.2761e-02|2.1232e-03|9.2168e-06|7.5934e-07|4.1175e-07|3.6585e-08|5.7613e-08|2.0129e-08|1.4268e-08|8.5435e-09|
> > > |FGM|3.0000e-03|2.4903e-06|1.2692e-06|1.1495e-06|6.2296e-07|5.9503e-07|5.8089e-07|5.1022e-07|2.9895e-07|2.9490e-07|
> > >
> > > **In both experiments, we observe that AGDA demonstrates a significantly faster convergence rate, this observation is consistent with the earlier small-scale and synthetic datasets in our draft.**
> > >
> > >
> > > ### 2. Nonconvex Problems
> > > For non-convex optimization, we trained a ResNet18 on CIFAR-10 and compared AGDA with AdamW and DoG. For AdamW, we use a learning rate of 1e-3 and weight decay of 0.01. For DOG, we use reps_rel=1e-3, which is the same as our method. The results are shown below.
> > > #### Table 3: Performance on neural network
> > >
> > > **AdamW**
> > > |Epoch|1|10|20|30|40|50|
> > > |---|---|---|---|---|---|---|
> > > | loss| 1.84| 0.46| 0.22| 0.13| 0.08| 0.06|
> > > | vali loss| 0.16| 0.07| 0.05| 0.06| 0.04| 0.04|
> > > | accuracy| 8.60| 82.18| 86.38| 89.06| 89.46| 90.22|
> > >
> > > **AGDA_LSFM**
> > > |Epoch|1|10|20|30|40|50|
> > > |---|---|---|---|---|---|---|
> > > | loss| 3.43| 0.34| 0.16| 0.10| 0.07| 0.05|
> > > | vali loss| 0.17| 0.05| 0.04| 0.04| 0.04| 0.04|
> > > | accuracy| 8.60| 84.36| 88.64| 89.88| 89.08| 88.92|
> > >
> > > **DoG**
> > > |Epoch|1|10|20|30|40|50|
> > > |---|---|---|---|---|---|---|
> > > | loss| 1.91| 0.78| 0.44| 0.29| 0.20| 0.15|
> > > | vali loss| 0.18| 0.10| 0.07| 0.05| 0.05| 0.07|
> > > | accuracy| 8.60| 69.48| 79.70| 84.02| 84.72| 86.88|
> > >
> > >
> > > In this experiment we observe that AGDA-LSFM tracks the AdamW baseline and outperforms DoG. We note that these experiments are preliminary due to the limited time in the rebuttal period. For example, we used a standard learning rate for AdamW without further tuning, so these results do not claim that our method is superior. However, they clearly show that our method is competitive with SOTA parameter-free method, even in a deep learning setting where our theory's boundedness and convexity assumption may not hold.
> > >
> > > As noted in Ivgi et al., extension to really large-scale models like LLMs requires substantial improvement, for example, like layer-wise DOG stepsize, and careful setting $\bar{r}$. We leave this, as well as theoretical analysis for the non-convex and unbounded setting, as exciting directions for future work.
> > >
> > > We hope these additional experiments address your question. Please let us know if you have any further concerns.

---

> > > > ### Comment · Reviewer_4xjD · 2025-08-07
> > > >
> > > > Thank you for the rebuttal. I have no further questions and will raise my rating to 5, although I do not observe a significantly faster convergence rate for AGDA in Table 1.

---

> > > > > ### Author Response · Authors · 2025-08-08
> > > > >
> > > > > Thank you for updating your rating and for the candid comment!
> > > > >
> > > > > Your observation is correct. We speculate that this is because the Higgs dataset is low-dimensional (28 features) and highly overdetermined (we use about 1 million samples), making it relatively well-conditioned; in such regimes, FGM already converges quickly, so the benefit of our distance-adaptive step size is understandably modest. By contrast, on higher-dimensional or more ill-conditioned problems (e.g., News20) AGDA’s advantage is clearer. In the revision, we will test AGDA on more challenging problems to further evaluate its effectiveness.
> > > > >
> > > > > Thank you again for your time and constructive feedback.

---

### Official Review · Reviewer_MdZu · 2025-07-10

**Clarity:** 3
**Significance:** 2
**Originality:** 2
**Rating:** 4
**Confidence:** 4

**Summary:**

This paper introduces new parameter-free optimization algorithms.
The paper then analyzes the convergence rate for the objective function, which is convex and exhibits Hölder smoothness.

**Questions:**

# Questions
1. Lemma 7 shows that $x^i,y^i,v^i$ remain inside the ball around $x^*$. Can't Theorem 1 be proven directly from Lemma 7 instead of being proven using Proposition 2?

# Suggestions
1. In my opinion, the paper should explain more clearly why the algorithm only needs that the objective function is locally Hölder smooth. This is related to question 1.
2. In Theorem 2, instead of using $z^K$, which is an arg-min, it would probably be better to define $k^\*$ as in Theorem 3 and use $y^{k^\*}$.
3. Missing citation of [1], which introduced a parameter-free algorithm that achieves near-optimal results for both the nonsmooth and smooth cases.
4. There are some typos in the paper:
    1. In equation (4), you forgot to write the Hölder smoothness constant.
    2. In line 184, you accidentally wrote $f$ inside the bound.
    3.  In Algorithm 2, line 2, you wrote $z$ instead of $v$.
    4. In Algorithm 2, line 4, you accidentally wrote $\nabla$ instead of $\tilde{\nabla}$.
    5. In equation (17), you wrote $z^{k^\*}$ instead of $y^{k^\*}$.

[1] Itai Kreisler, Maor Ivgi, Oliver Hinder, and Yair Carmon. Accelerated parameter-free stochastic
optimization. In Conference on Learning Theory (COLT), 2024.

**Ethical Concerns:**

["NO or VERY MINOR ethics concerns only"]

**Final Justification:**

The weaknesses that I mentioned still hold after the author's rebuttal; however, in my opinion, none of the weaknesses were major.
A slight exception is Weakness 1, where the author said they will clarify the theorem; however, the theorem still requires that $K\ge \Omega(\text{log}(D/\bar{r}))$ in order to be optimal.

**Limitations:**

yes, except what I mentioned in weakness 1.1.

**Quality:**

3

**Strengths And Weaknesses:**

# Strengths
1. The paper introduces the AGDA algorithm and analyzes the convergence rate bound of this algorithm for the deterministic case. The resulting convergence rate bound is near optimal.
2. To avoid the problems that the line search in the AGDA algorithm causes for the stochastic case, the paper introduces the AGDA LSFM algorithm, which does not contain a line search. The paper then analyzes the convergence rate bound of this algorithm for the stochastic case and achieves an in-expectation bound. In the case that the domain is bounded, has a diameter $D$, and that $\\| x_0-x^* \\|=\Theta(D)$, then the in-expectation bound is near optimal.

# Weaknesses
1. The convergence rate bounds require that the number of steps $K$ will be $K\ge\Omega(\text{log}(D/\bar{r}))$.
    1. In this regard, the paper only mentions this constraint in the proofs that are written in the appendix. The paper should mention this clearly in the theorems themself.
2. As mentioned in the paper, the AGDA algorithm contains a line search and thus is inappropriate for the stochastic case.
3. The AGDA LSFM algorithm requires a bounded domain with diameter $D$. The convergence rate bound is dependent on $D$ instead of $\\| x_0-x^* \\|$, and thus can be very large in the case that $\\| x_0-x^* \\|\neq\Theta(D)$.

---

> ### Author Rebuttal · Authors · 2025-07-31
>
> ## Weakness
> **1)The convergence rates of the algorithms hold under the condition that $k \geq \Omega(\log(D_0/\bar{r}))$. 2) This condition is in appendix and should be mentioned clearly in the theorems.**
>
> **Ans**: We thank the reviewer for the careful reading and agree completely.  Our original proof used the condition $k \geq \Omega(\log(D_0/\bar{r}))$ to ensure the term $(\frac{4D_0}{\bar{r}})^{\frac{1}{k}}$ in (51) is almost a constant, which simplified the final expression of the rate. While this is a mild condition, its role should be transparent.
>
> To address this, we have revised the manuscript to make the result more precise. We now state the more general bound directly in Theorem 2, which removes the hidden assumption. The updated rate is:
> \begin{equation}
> \mathcal{O}\left(\frac{\widehat{M}_{\nu}D_0^{1+\nu}(\frac{4D_0}{\bar{r}})^{\frac{1}{k}}\log^2 e\frac{D_0}{\bar{r}}}{k^{\frac{1+3\nu}{2}}}\right),
> \end{equation}
>
> We have added a remark immediately following the theorem. This remark explains that the term $(\frac{4D_0}{\bar{r}})^{\frac{1}{k}}$ approaches 1 as k increases and is bounded by a small constant under the mild condition $k \geq \Omega(\log(D_0/\bar{r}))$. If $\bar{r}$ is sufficiently large,  this condition is much weaker than the polynomial dependency on $D_0$ typically required for nontrivial rate in gradient descent. We thank the reviewer for helping us improve the theorem's clarity and precision.
>
> **AGDA contains a line search and thus is inappropriate for the stochastic case.**
>
> **ANS**: Certainly, line search is inappropriate for the stochastic case without stronger assumptions on the stochasticity of the gradients. To address this, we developed AGDA-LSFM, which extends AGDA to the stochastic setting. However, this extension requires the additional assumption that the domain is bounded.
>
> **In the stochastic case, the convergence rate depends on $D$ instead of $\|\|x_0-x^\*\|\|$, and thus can be very large when $\|\|x_0-x^\*\|\|\neq \Theta(D)$.**
>
> **ANS**: Yes, it is a limitation of AGDA-LSFM. We have some ideas about removing the assumption of bounded domain and we think it is a future work. However, you can check our proof of AGDA-LSFM and find out the main error term is $\frac{\beta_k (D_0^2 - D_k^2)}{2A_k} + \frac{\beta_k \bar{r}_k^2}{8A_k}$ and the right term can be bounded using $D_0$ but not $D$. So thisthe bound can be partially relaxed comparing to USFGM which dosen't have this property.
>
> ## Question
> **Can't Theorem 1 be proven directly from Lemma 7 instead of being proven using Proposition 2?**
>
> **Ans:** We thank the reviewer for this excellent suggestion. The reviewer is correct that Theorem 1 can be proven directly from Lemma 7 using an inductive argument.
>
> The core of our analysis is contained in Lemmas 6 and 7. These lemmas establish the crucial results for a single iteration of our algorithm (AGDA), including the success of the line search, the one-step convergence error, and the boundedness of the next iterate, given that all previous iterates are well-defined. These findings provide the essential components for the main inductive proof in Theorem 1.
>
> To improve the paper's clarity and logical flow, we decided to remove Proposition 2 and integrate its content in Theorem 1. We now prove Theorem 1 directly from the foundational lemmas. Recognizing the importance of those lemmas, we will state Lemmas 6 and 7 in the main body of the paper. Moreover, in the revision, we will give an outline to the structure of our proof and clarify the connections between the key lemmas and theorems.
>
>
>
>
> **Why the algorithm only needs the objective function to be locally Hölder smooth?**
>
> **ANS:** This is a great question that gets to the heart of our paper's technical contribution. For local Hölder smooth function, smooth parameter will depend on the variables, and hence it would be potentially be unbounded if we can not guarantee boundedness of the iterates $\{x^k,y^k,v^k\}$.
> Therefore, our proof must achieve two intertwined goals simultaneously:
> - Prove Boundedness: Show that all iterates generated by the algorithm remain within the domain where the function is well-behaved.
> - Establish Convergence: Prove the algorithm's convergence rate.
>
> This dual objective makes the analysis more complex than standard proofs for globally smooth functions, which only need to focus on the second goal. Our inductive proof is designed specifically to handle this challenge. We understand this point could have been clearer. As promised in our previous response, we will better illustrate this key technical intuition and the structure of our proof. Thank you for helping us improve the paper's clarity.
>
>
>
> **In Theorem 2, instead of using $z^K$, it would probably be better to define $k^\*$ as in Theorem 3 and use  $y^{k^\*}$.**
>
> **ANS:** We thank the reviewer for this constructive suggestion. We agree that using the same notation for the output iterate in both theorems improves the consistency and readability of our results. In the revised manuscript, we have updated Theorem 2 and state the bound in terms of $\psi(y^{k^\*})-\psi(x^\*)\in\mathcal{O}\left(\frac{\widehat{M}\_{\nu}D_0^{1+\nu}\log^2 e\frac{D_0}{\bar{r}}}{K^{\frac{1+3\nu}{2}}}\right)$, where $k^\*=\mathop{\arg\min}\_{0\le i\le k}\frac{\bar{r}\_k^{{1}/{2}}}{\sum_{i=0}^{k-1}\bar{r}_i^{1/2}}$.
>
> **Missing citation of [1], which introduced a parameter-free algorithm that achieves near-optimal results for both the nonsmooth and smooth cases.**
>
> **ANS:** We will cite and discuss [1] in our revision—thank you for the suggestion.  Our paper focuses on achieving near-optimal accelerated rates across the full spectrum of Hölder smooth functions.  Although U-DOG achieves a near-optimal rate in the smooth setting, it does not have a theoretical guarantee for nonsmooth or weakly smooth problems. A key strength of their work is providing guarantees for unbounded domains, albeit with a suboptimal rate in that setting. We plan to continue our research in this direction in future work.
>
> **There are some typos in the paper.**
>
> **ANS**: Thank you! We have fixed these typos in the revision.
>
>
> [1] Itai Kreissler, Maor Ivgi, Oliver Hinder, and Yair Carmon. Accelerated parameter-free stochastic optimization. In Conference on Learning Theory (COLT), 2024.

---

> > ### Comment · Reviewer_MdZu · 2025-08-05
> >
> > I want to thank the authors for their response.
> > For now, I will keep my score as it is.

---

> > > ### Author Response · Authors · 2025-08-06
> > >
> > > Thank you very much for your critical comments and patient replies!

---

### Decision · Program_Chairs · 2025-09-17

**Decision:**

Accept (poster)

**Comment:**

The paper studies the problem of minimizing a convex function that is Holder smooth, both in the deterministic and stochastic settings. The main contributions of the paper are parameter-free algorithms that attain nearly-optimal convergence guarantees under some assumptions such as bounded diameter in the stochastic setting. In contrast, prior work requires knowledge of problem parameters such as the smoothness. The paper evaluates the empirical performance of the proposed algorithms on small scale instances.

The reviewers generally appreciated the theoretical contribution and were positive about the paper. The reviewers raised several concerns, including significant limitations of the experimental evaluation, the overhead introduced by the approach, the bounded domain assumption for the stochastic algorithm, and the extent of the technical novelty. The author response provided additional experiments with larger-scale datasets. Following the discussion, the reviewers remained positive about the paper and there was consensus that the paper meets the threshold for acceptance.